# Transitions in dynamical regime and neural mode during perceptual decisions

Thomas Zhihao Luo[1,3,8 ✉], Timothy Doyeon Kim[1,4,5,8 ✉], Diksha Gupta[1,6], Adrian G. Bondy[1], Charles D. Kopec[1], Verity A. Elliott[1], Brian DePasquale[1,7] & Carlos D. Brody[1,2 ✉]

Perceptual decision-making is thought to be mediated by neuronal networks with attractor dynamics[1,2]. However, the dynamics underlying the complex neuronal responses during decision-making remain unclear. Here we use simultaneous recordings of hundreds of neurons, combined with an unsupervised, deep-learning-based method, to discover decision-related neural dynamics in the rat frontal cortex and striatum as animals accumulate pulsatile auditory evidence. We found that trajectories evolved along two sequential regimes: an initial phase dominated by sensory inputs, followed by a phase dominated by autonomous dynamics, with the flow direction (that is, neural mode) largely orthogonal to that in the first regime. We propose that this transition marks the moment of decision commitment, that is, the time when the animal makes up its mind. To test this, we developed a simplified model of the dynamics to estimate a putative neurally inferred time of commitment (nTc) for each trial. This model captures diverse single-neuron temporal profiles, such as ramping and stepping[3,4]. The estimated nTc values were not time locked to stimulus or response timing but instead varied broadly across trials. If nTc marks commitment, evidence before this point should affect the decision, whereas evidence afterwards should not. Behavioural analysis aligned to nTc confirmed this prediction. Our findings show that decision commitment involves a rapid, coordinated transition in dynamical regime and neural mode and suggest that nTc offers a useful neural marker for studying rapid changes in internal brain state.

Theories of attractor dynamics have been successful at capturing several brain functions[5], including motor planning[6] and neural representations of space[7,8]. Attractors are a set of states towards which a system tends to evolve from a variety of starting positions. In these theories, computations of a brain function are carried out using the temporal evolution or the dynamics of the system. Experimental findings support the idea that the brain uses systems with attractor states for computations underlying working memory[6] and navigation[7]. These theories often focus on the low-dimensional nature of neural population activity[2,9,10] and account for responses across a large number of neurons using a dynamical system model in which the variable has only a few dimensions[7,11–13].

Attractor network models have also been proposed to underlie perceptual decision-making: the process by which noisy sensory stimuli are categorized to select an action or mental proposition. In these hypotheses, the network dynamics carry out the computations needed in decision formation[1,2,14–16], such as accumulating sensory evidence and committing to a choice. Although some experimental evidence favours a role of attractors in perceptual decisions[2,16,17], the actual population-level dynamics underlying decision-making have not

been directly estimated. Knowledge of these dynamics would directly test the current prevailing attractor hypotheses, provide fundamental constraints on neural circuit models and account for the often complex temporal profiles of neural activities.

A separate line of work involves tools, sometimes based on deep learning, for discovering the low-dimensional component of neural activity in a data-driven manner[10,18,19]. In this approach, the spike trains of many simultaneously recorded neurons are modelled as being a function of a few latent variables that are shared across neurons.

To combine both lines of work, we used an innovative method[20] that estimates, from the spike trains of simultaneously recorded neurons, the dynamics of a low-dimensional variable $\mathbf{z}$, given by:

$$\dot{\mathbf{z}} = F(\mathbf{z}, \mathbf{u}) + \boldsymbol{\eta}, \qquad (1)$$

where $\mathbf{u}$ are external inputs, $\boldsymbol{\eta}$ is noise and, when applied to perceptual decisions, $\mathbf{z}$ represents the dynamical state of the decision process of the brain at a given time (Fig. 1a–c). The instantaneous change of the decision variable or its dynamics is given by $\dot{\mathbf{z}}$, which depends on $\mathbf{z}$

[1]Princeton Neuroscience Institute, Princeton University, Princeton, NJ, USA. [2]Howard Hughes Medical Institute, Princeton University, Princeton, NJ, USA. [3]Present address: School of Biological Sciences, University of Utah, Salt Lake City, UT, USA. [4]Present address: University of Washington, Seattle, WA, USA. [5]Present address: Allen Institute, Seattle, WA, USA. [6]Present address: Sainsbury Wellcome Centre, University College London, London, UK. [7]Present address: Department of Biomedical Engineering, Boston University, Boston, MA, USA. [8]These authors contributed equally: Thomas Zhihao Luo, Timothy Doyeon Kim. ✉e-mail: luo@utah.edu; timkimd@uw.edu; brody@princeton.edu

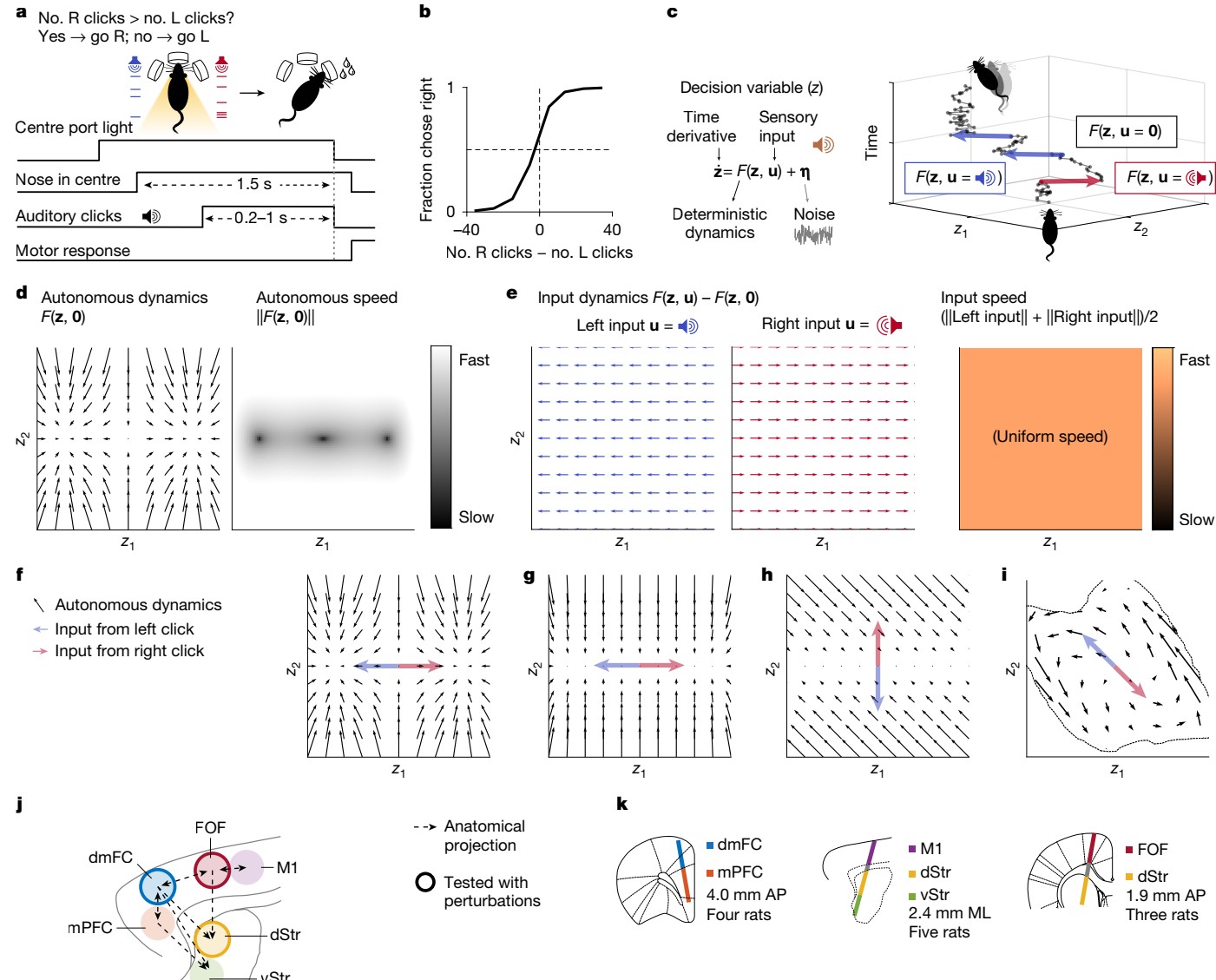

**Fig. 1 | Attractor models of decision-making were tested by recording from the rat frontal cortex and striatum. a**, Rats were trained to accumulate auditory pulsatile evidence over time. While keeping its head stationary, the rat listened to randomly timed clicks played from loudspeakers on its left (L) and right (R). At the end of the stimulus, the rat received a water reward for turning to the side with more clicks. The earliest time when a rat could respond was fixed at 1.5 s relative to the moment of inserting its nose in the centre port (that is, not a reaction time paradigm). **b**, Behavioural performance in an example recording session. Dashed reference lines at abscissa = 0 and ordinate = 0.5. **c**, The decision process is modelled as a dynamical system. Right, the blue and red arrows represent the change in the decision variable in the presence of a left or right, respectively. $z_1$, $\mathbf{z}$ dimension 1; $z_2$, $\mathbf{z}$ dimension 2. **d**, Autonomous dynamics illustrated using the bistable attractor hypothesis. In the velocity vector field (that is, flow field; left), the arrow at each value of the decision variable $\mathbf{z}$ indicates how the instantaneous change depends on $\mathbf{z}$ itself. The orientation of the arrow represents the direction of the change, and its size represents the speed, also quantified using a heat map (right). **e**, Changes in $\mathbf{z}$ driven solely by external sensory inputs (example of bistable attractors). **f**, Bistable attractor hypothesis of decision-making, with directions of the input dynamics (based on ref. 1). **g**, A hypothesis supposing a line attractor in the autonomous dynamics on the basis of the DDM of decision behaviour (based on ref. 23). **h**, Recurrent neural networks can be trained to make perceptual decisions using a line attractor that is not aligned to the input dynamics (non-normal; based on ref. 2). **i**, Unsupervised discovery (this study) of dynamics that have not been previously considered. **j**, Six interconnected frontal cortical and striatal regions are examined here. vStr, ventral striatum. **k**, Neuropixels recordings (318 ± 147 neurons per session per probe, mean ± s.d.) from 12 rats in total (two to three regions per rat). AP, anteroposterior; ML, mediolateral.

itself, and $\mathbf{u}$ and $\boldsymbol{\eta}$. This approach aims to estimate the function $F$ and, through it, capture the nature of decision-making neural dynamics.

## Differentiating dynamical hypotheses

The function $F$ is useful for distinguishing among hypotheses of decision-making. $F$ can be dissected into two components: autonomous dynamics and input-driven dynamics. Autonomous dynamics are dynamics in the absence of sensory inputs $\mathbf{u}$ (that is, $F(\mathbf{z}, \mathbf{0})$;

Fig. 1d and Extended Data Fig. 1a,b). Input dynamics are changes in $\mathbf{z}$ driven by $\mathbf{u}$, which can be distinguished from autonomous dynamics as $F(\mathbf{z}, \mathbf{u}) - F(\mathbf{z}, \mathbf{0})$. Input dynamics can depend on $\mathbf{z}$ (Fig. 1e and Extended Data Fig. 1c–e).

Many of the prevailing neural attractor hypotheses have been inspired by a classic and successful behavioural-level model, the drift diffusion model (DDM)[21,22]. In the behavioural DDM, a scalar (that is, one-dimensional) decision variable $z$ is driven by sensory evidence inputs (Extended Data Fig. 6a,b). For example, for decisions between

go right versus go left, momentary evidence for right (left) might drive $z$ in a positive (negative) direction. Through these inputs, the momentary evidence accumulates over time in $z$ until the value of $z$ reaches an absorbing bound, a moment thought to correspond to decision commitment and after which inputs no longer affect $z$. Different bounds correspond to different choice options: a positive (or negative) bound would correspond to the decision to go right (or go left). A straightforward implementation of the DDM in neural population dynamics, which we refer to as the DDM line attractor, would posit a line attractor in neural space, with the position of the neural state $\mathbf{z}$ along that line representing the value of $\mathbf{z}$ and two point attractors at the ends of the line representing the decision commitment bounds[23] (Fig. 1g). Another hypothesis approximates the DDM process using bistable attractors[1], with each of the two attractors representing each of the decision bounds and, in between the two attractors, a one-dimensional stable manifold of slow autonomous dynamics that corresponds to the evidence accumulation regime (Fig. 1f). In both the DDM line attractor and bistable attractor hypotheses, evidence inputs are aligned with the slow dynamics manifold and the attractors at its end points. A third hypothesis, inspired by trained recurrent neural networks, also posits a line attractor (Fig. 1h) but allows for evidence inputs that are not aligned with the line attractor and that accumulate over time through non-normal autonomous dynamics[2]. In all three hypotheses, the one-dimensional line attractor and/or slow manifold is stable, meaning that autonomous dynamics flow towards it (Fig. 1f–h). Because these three hypotheses were each designed to explain a particular set of the phenomena observed in decision-making experiments, a broader range of experimental observations could suggest dynamics that have not been previously considered. As but one example, autonomous dynamics may contain discrete attractors that do not lie at the end points of a one-dimensional slow dynamics manifold; many other arrangements are possible. In the data-driven approach we describe below, $F$ is estimated purely from the spiking data and the timing of sensory input pulses, without incorporating any assumptions from the behavioural DDM or other existing hypotheses.

Dissociating between autonomous and input dynamics requires neural recordings during a decision unfolding over a time period that includes intervals both with and without momentary evidence inputs. We trained rats to perform a task in which they listened to randomly timed auditory pulses played from their left and their right and reported the side on which more pulses were played[24] (Fig. 1a). The stochastic pulse trains allow us to sample neural responses time locked to pulses, which are useful for inferring input-driven dynamics, and also the neural activity in the intervals between pulses, which is useful for inferring autonomous dynamics. Expert rats are highly sensitive to small differences in auditory pulse number (Fig. 1b and Extended Data Fig. 2a), and the behavioural strategy of rats in this task is typically well captured by gradual accumulation of evidence, which is at the core of the DDM[24–26].

While the rats performed this task, we recorded six frontal cortical and striatal regions with chronically implanted Neuropixels probes (Fig. 1j,k and Extended Data Fig. 2b). The frontal orienting fields (FOF) and the anterior dorsal striatum (dStr) are known to be causally necessary for this task and are interconnected[27–29]. The dorsomedial frontal cortex (dmFC) is a major anatomical input to the dStr[30], as confirmed by our retrograde tracing (Extended Data Fig. 2c), and is also causally necessary for the task (Extended Data Fig. 2d). The dmFC is interconnected with the medial prefrontal cortex (mPFC) and, less densely, the FOF, the primary motor cortex (M1)[31] and the anterior ventral striatum[30].

## Unsupervised discovery of dynamics

To test the attractor hypotheses and allow discovery of dynamics not previously considered, a flexible yet interpretable method was needed. We used an innovative deep learning method (flow field inference from neural data using deep recurrent networks; FINDR[20]) that

infers the low-dimensional stochastic dynamics that best account for population spiking data. The low dimensionality of the description is critical for interpretability. Prominent alternative deep-learning-based approaches for inferring neural latent dynamics involve models in which these latent dynamics have hundreds of dimensions and are deterministic[18]. By contrast, FINDR infers latent dynamics that are low dimensional and stochastic. The stochasticity in the latent dynamics accounts for noise in the decision process that contributes to errors. FINDR approximates the decision-relevant dynamics $F$ with a gated multilayer perceptron network[32] and noise $\boldsymbol{\eta}$ as a Gaussian with diagonal covariance (equation (1) and Fig. 2a). The firing rate of each neuron at each time point is modelled as a weighted sum of the $\mathbf{z}$ variables, followed by a softplus nonlinearity, which can be thought of as approximating neuronal current–frequency curves[6] (Fig. 2b). The weighting for each neuron (vector $\mathbf{w}_n$ for neuron $n$, comprising the $n$th row of a weight matrix $W$; Fig. 2b) is fit to the data. To aid the interpretability of $\mathbf{z}$, we transform $W$ after training such that its columns are orthonormal and it therefore acts as a rotation. As a result, angles and distances in $\mathbf{z}$ are preserved in $W\mathbf{z}$ (neural space before softplus). Before learning $F$ and $W$, we separately account for the decision-irrelevant, deterministic but time-varying baseline firing rate for each neuron (baseline in Fig. 2b) so that FINDR can focus on the choice formation process.

We first confirmed that, in synthetic data, the velocity vector fields (flow fields) inferred by FINDR can distinguish between existing attractor hypotheses (Extended Data Fig. 1f–h). Next, we turned to the recorded spiking data and confirmed that FINDR provides a good fit to the heterogeneous single-trial firing rates of individual neurons and to the complex dynamics in their peristimulus time histograms (PSTHs) conditioned on the sign of the evidence (Extended Data Fig. 3a–d). We found that two latent dimensions suffice to capture our data well (Extended Data Fig. 3e–i). For models with more than two latent dimensions, the latent dynamics are still mostly confined to two dimensions, and this two-dimensional manifold is approximately an attractor (Extended Data Fig. 3h–k).

Figure 2c–h shows a representative recording session from the dmFC and the mPFC. We found that, generally, two-dimensional input-driven dynamics and autonomous dynamics inferred by FINDR were not described well by the existing hypotheses: in all three hypotheses illustrated in Fig. 1d–h, there is a one-dimensional stable manifold that either is or approximates a line attractor. By contrast, even though, over the first 330 ms, the average trajectories evolve along an approximately straight line (Fig. 2h), the line is not a one-dimensional attractor, and individual trials diverge from it. Furthermore, in all three hypotheses in Fig. 1d–h and in all other hypotheses we are aware of, autonomous dynamics play an important part throughout the entire decision-making process. For example, autonomous dynamics are what enforce the stability of the one-dimensional slow manifolds in Fig. 1d–h. By contrast, at least in the space of the latent variable $\mathbf{z}$, FINDR-inferred dynamics suggest that, initially, motion in neural space is dominated and driven by inputs to decision-making regions (that is, by the input-dependent dynamics), not the autonomous dynamics, which are slow in both dimensions (Fig. 2c–h), not only one. Later in the decision-making process, the balance between autonomous versus input-driven dynamics inverts, and it is the autonomous dynamics that become dominant. Plots in Fig. 2g show the difference in magnitude between autonomous and input-driven dynamics (indicated with the colour scale) on the $z$ plane. The initial dominance of the input-driven dynamics can be seen in the zone near the $(0, 0)$ origin at the negative end of the colour scale. The later dominance of autonomous dynamics can be seen in the right and left edges of the sampled region, reached later in the decision-making process, at the opposite end of the colour scale. Moreover, the direction of instantaneous change driven by the inputs (slightly clockwise from horizontal in Fig. 2e) is not aligned with the direction of the strongest autonomous dynamics in the left and right edges of the sampled region (slightly anticlockwise from

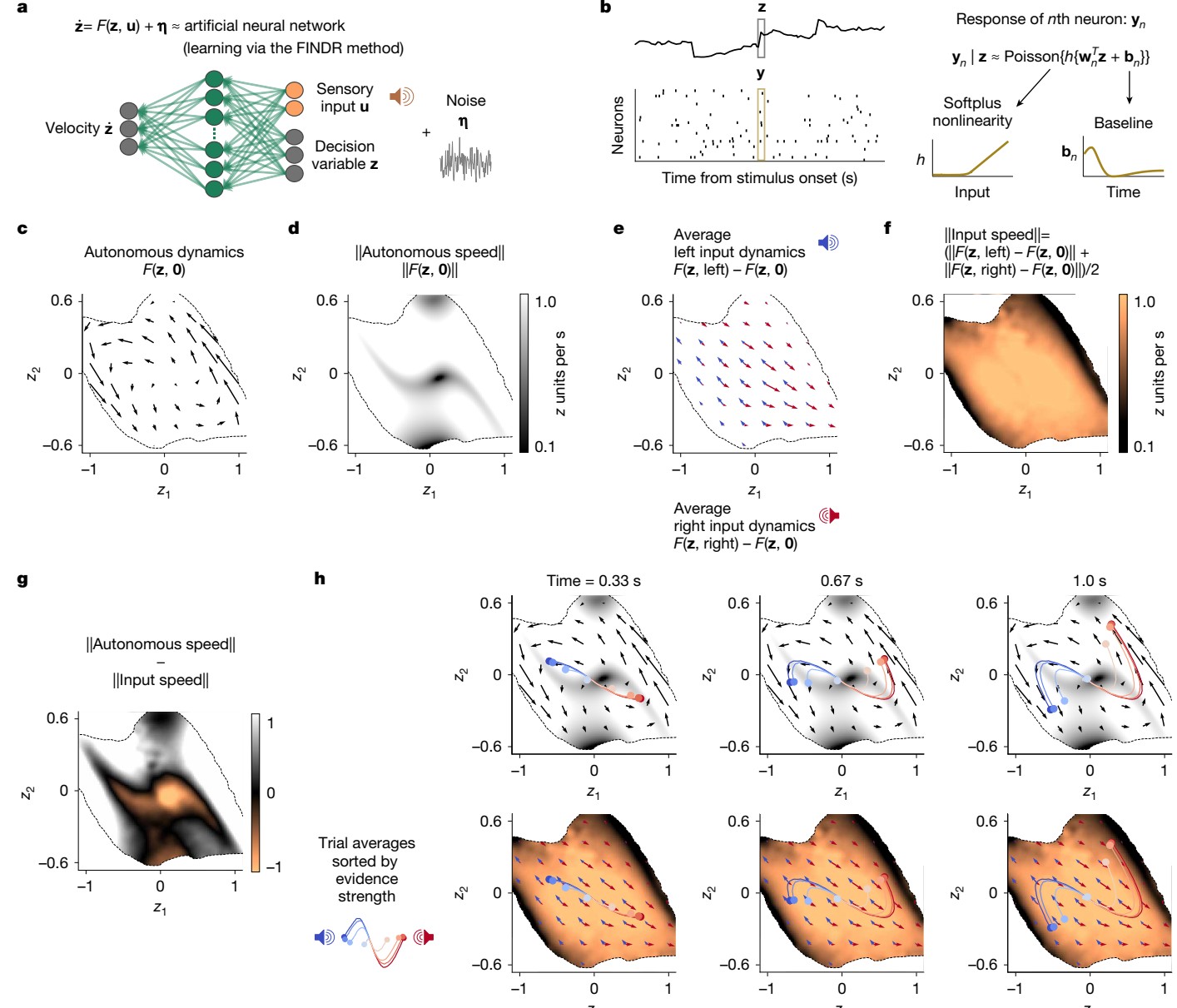

**Fig. 2 | Unsupervised discovery shows transitions in dynamical regime and neural mode underlying the shift from evidence accumulation to decision commitment. a**, Decision-relevant dynamics are inferred using FINDR[20]. **b**, FINDR learns the decision variable **z** that best captures neural spiking activity. Each neuronal spike count at a given time step is modelled as a Poisson random variable with the rate given by an affine transformation of **z** at that time step, followed by the softplus nonlinearity. The grey box indicates the decision variable **z** at an example time step, and the yellow box indicates the spike counts at that time step. A time-varying baseline is learnt for each neuron to capture the decision-irrelevant component of its activity. **c–h**, Vector field inferred from 96 simultaneously recorded choice-selective neurons in the dmFC and the mPFC from a representative session. Only the portion of the state space visited by at least 50 of 5,000 simulated 1-s trajectories (sample zone) is shown.

**c**, Autonomous dynamics. **d**, Speed of autonomous dynamics. **e**, Input dynamics for left and right clicks. If **u** = [1;0] indicates a left click input, $F(\mathbf{z}, [1;0]) - F(\mathbf{z}, \mathbf{0})$ is the input dynamics given a left click. However, the average left input dynamics depend on the frequency of left clicks, given by $p(\mathbf{u} = [1;0]|\mathbf{z})$. Therefore, we compute the average left input dynamics $F(\mathbf{z}, \text{left}) - F(\mathbf{z}, \mathbf{0})$ as $p(\mathbf{u} = [1;0]|\mathbf{z})(F(\mathbf{z}, [1;0]) - F(\mathbf{z}, \mathbf{0}))$. We compute the average right input dynamics similarly, with **u** = [0;1]. **f**, Speed of input dynamics. **g**, Difference in speed between autonomous and input dynamics. **h**, Initially, **z** is strongly driven by inputs, and its trajectories develop along the evidence accumulation axis aligned with the direction of input dynamics. At a later time, the trajectories become largely insensitive to the inputs and are instead driven by autonomous dynamics to evolve along the decision commitment axis aligned with the direction of autonomous dynamics.

vertical in Fig. 2c). The curved trial-averaged trajectories of **z** emerge from this non-alignment in the input direction and the autonomous direction later in the decision-making process. The change from an input-dominated to an autonomous-dominated dynamical regime and the sharp turn in the direction of the neural trajectories in Fig. 2c–h were observed consistently across rats and behavioural sessions (Fig. 3a–d). These observations were robust to several different initializations of

the neural networks in FINDR, the order of minibatches during training and how datasets were split into training and test sets (Extended Data Fig. 4). They are therefore a consistent finding of the analysis.

To perform a head-to-head comparison with the three hypotheses in Fig. 1d–h, we constructed a variant of FINDR in which the network parametrizing *F* was replaced by a parametrization of the dynamics constrained to describe those three hypotheses (Fig. 3e,f and Extended

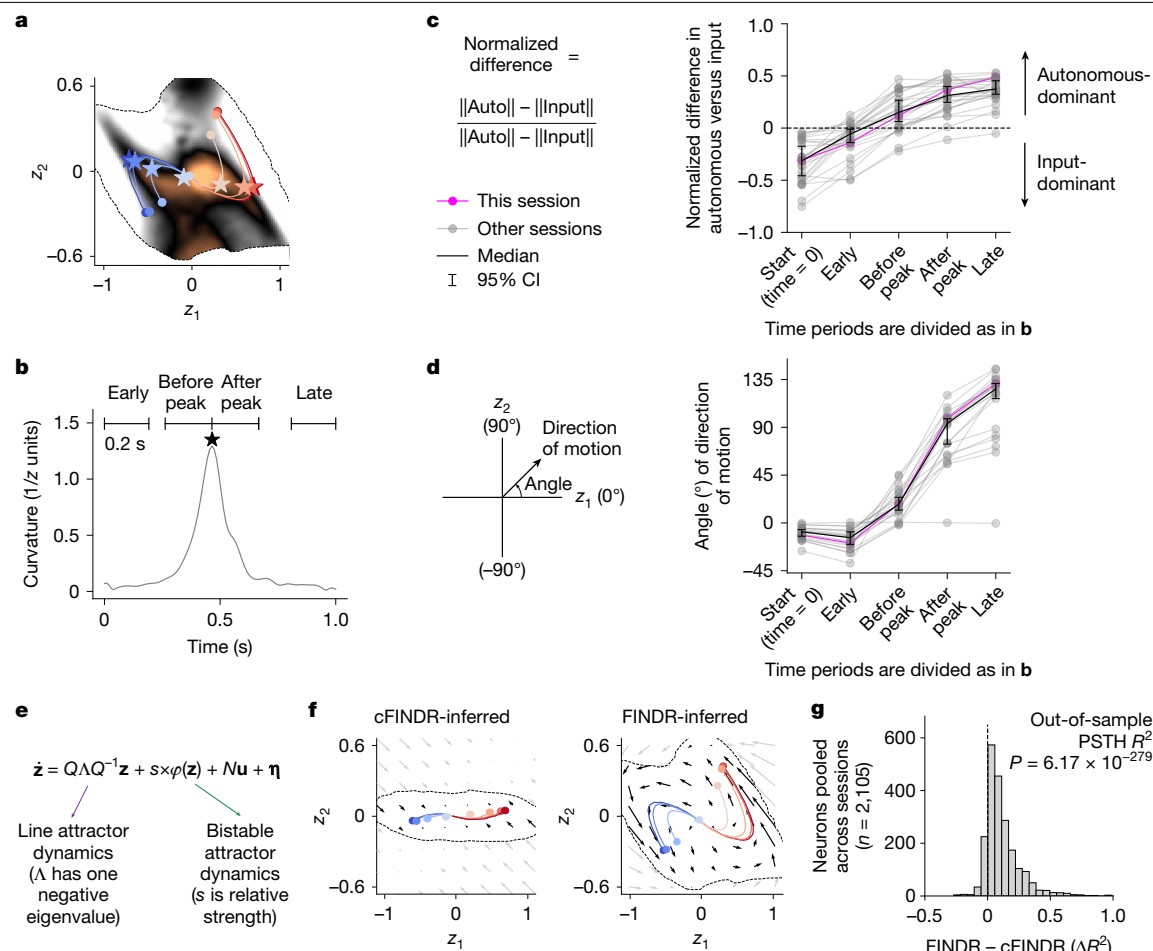

**Fig. 3 | FINDR shows transitions in dynamical regime and neural mode consistently across sessions and better captures the data than a constrained model based on previous hypotheses. a**, To quantify how speed difference between autonomous and input dynamics evolves over a trial, we identify the time point when the latent trajectories curve (stars) and compute the speed difference in Fig. 2g before and after this point. The latent trajectories are trial-averaged, sorted by evidence strength. The trial-averaged trajectories and stars are coloured as in Fig. 2h. **b**, The peak is defined as the time of maximum curvature in the trial-averaged trajectory. Time periods are defined relative to this peak (before peak and after peak) and to trial start and end (early and late) for **c**,**d**. Black star symbol represents the peak of average trajectory curvature. **c**, We compute the normalized difference in speed between autonomous and input dynamics for five different time periods (start (time = 0 s), early, before peak, after peak and late) from vector fields inferred from sessions with more than 30 recorded neurons, over 400 trials during which the animal performed with more than 80% accuracy ($n = 27$ sessions). The dashed line indicates

normalized difference of 0. CI, confidence interval. **d**, For sessions in which FINDR with the two-dimensional decision variable **z** fit significantly better than FINDR with one-dimensional $z$ ($n = 21$ of 27; Extended Data Fig. 3), we measured the direction of motion of the trial-averaged trajectories and its angle with respect to the $z_1$ axis for different time periods (curving of trial-averaged trajectories across 21 sessions). **e**, cFINDR captures previous hypotheses and replaces the neural network parametrizing $F$ with a combination of line attractor dynamics (specified by $Q\Lambda Q^{-1}$, with the diagonal matrix $\Lambda$ having one zero and one negative eigenvalue) and bistable attractor dynamics (specified by a nonlinear function $\varphi$; Methods). **f**, Autonomous dynamics inferred by cFINDR and FINDR are shown for a representative session, with vector field outside the sample zone in grey. **g**, The coefficient of determination ($R^2$) of the evidence–sign conditioned PSTH computed using fits of FINDR is significantly greater than those computed using fits of cFINDR (across 27 sessions, two-sided Wilcoxon signed-rank test).

Data Fig. 5). If the data were described well by one of these hypotheses, we would expect this variant (which we refer to as cFINDR, for constrained FINDR) to fit the data well, particularly out of sample, because it has far fewer parameters than FINDR. However, unconstrained FINDR consistently fit the data better than cFINDR, confirming that previous hypotheses do not adequately capture the data. Although one of the hypotheses (Fig. 1h, suggesting non-normal dynamics with a line attractor) can generate curved trial-averaged trajectories apparently similar to those we see in the data (Fig. 3e,f and Extended Data Fig. 5g), there is a key difference, which is that, in this particular hypothesis, the turn from the initial flow direction induced by the inputs happens early, because the autonomous dynamics causing it are strong the moment the latent state departs from the line attractor. However, our data suggest that there is a more prolonged initial phase of flow along

the input directions before the turn, with the stronger autonomous dynamics happening much later in the decision-making process. We believe that this underlies the much better fits to the data for FINDR than those for cFINDR.

A recent study[33] described neural trajectories that were described well by non-normal dynamics[34,35]. Consistent with this, the two-dimensional FINDR-inferred autonomous dynamics around the origin are also non-normal (Extended Data Fig. 10b,c), although with a key difference with respect to refs. 33–35, which is that here the origin is unstable (Extended Data Fig. 10a,e).

Unsupervised inference of dynamics underlying decision-making, based only on spiking activity and sensory evidence inputs, thus suggests that the process unfolds in two separate sequential regimes. In the initial regime, dynamics are largely determined by the inputs, with

autonomous dynamics playing a minor role. The sensory evidence inputs (right and left clicks) drive the decision variable to evolve along an axis, parallel to the directions of the input dynamics, that we will term the evidence accumulation axis. In the second, later regime, these characteristics reverse; the trajectories representing the evolution of the decision variable become largely independent of the inputs and are instead mostly determined by autonomous dynamics. We will term the straight line along the direction of the autonomous dynamics in the later regime the decision commitment axis. Of note, the evidence accumulation axis and the decision commitment axis are not aligned with each other. During the regime transition, the trajectories in $z$ veer from evolving along the evidence accumulation axis to developing along the decision commitment axis. In neural space, this will equate to a transition from evolving along one mode (that is, a direction in neural space), corresponding to evidence accumulation, to another mode that, as explained below, we believe may correspond to decision commitment.

Although derived entirely from unsupervised analysis of neural spiking activity and auditory click times, these two regimes are reminiscent of the two regimes of the behavioural DDM: namely, an initial regime in which momentary sensory inputs drive changes in the state of a scalar decision variable $z$ and a later regime, after $z$ reaches a bound, in which the state becomes independent of sensory inputs (Extended Data Fig. 2e,f). The correspondence between the two regimes inferred from spiking activity and the behavioural DDM suggests that the transition between regimes may correspond to the moment of decision commitment. It further suggests that a modified neural implementation of the DDM, focusing on key aspects of the two regimes, could be a simple model that captures many aspects of the neural data, although having far fewer parameters than FINDR and thus greater statistical power. We next develop this model and show that it can be used to precisely infer the regime transition time in each trial and test the proposal that this transition corresponds to decision commitment.

## Simplified model of decision dynamics

FINDR-inferred vector fields show a rapid shift from strongly input-driven to autonomous-dominant dynamics, analogous to the transition from evidence accumulation to decision commitment in the behavioural DDM (Fig. 4a,b). The DDM captures behaviour in a wide range of decision-making tasks, including tasks in which the stimulus duration is determined by the environment[24,25,28,36,37], as used here. This suggests that the FINDR-inferred dynamics may be approximated by a simplified model in which the decision variable evolves as in the behavioural DDM.

The regime transition coincides with rapid reorganization in the neuronal population representation of the decision process. To quantify this reorganization, we treat the activity of each neuron as a dimension in neural space, with axes in this space as neural modes. Seen in this way, the shift from evidence accumulation to decision commitment is coordinated with a fast transition in the neural mode, analogous to the rapid change in neural modes from motor preparation to motor execution[38]. This motivates whether a simplified model based on a rapid, coordinated transition in both dynamical regime and neural mode can capture the key features of FINDR-inferred dynamics and broader experimental observations.

In what we will call the multimode or minimally modified DDM (MMDDM), a scalar decision variable $z$ evolves just as in the behavioural DDM, governed by three parameters (Fig. 4b, Extended Data Fig. 6a,b and the Methods). The key addition is that neurons encode $z$ differently before and after the decision commitment bound is reached. Each neuron has two weights: $w_{EA}$ for the evidence accumulation phase and $w_{DC}$ for the decision commitment phase. When $w_{EA}$ and $w_{DC}$ are constrained to be the same, the MMDDM reduces to a standard DDM with a single neural mode. In the DDM line attractor hypothesis in Fig. 1g, if the

autonomous dynamics towards the line attractor are strong relative to the noise, trajectories will be largely one dimensional, which are approximated well by a single-mode DDM. Because neurons multiplex both decision-relevant and decision-irrelevant signals[39,40], MMDDM includes terms for spike history and, similar to FINDR, decision-irrelevant baseline changes (Extended Data Fig. 6c–f). All parameters are fit jointly for each session using both neural activity and behavioural choices.

MMDDM can account for a broader range of neuronal profiles (Fig. 4c–g) than the single-mode DDM, which captures only ramp-like neuronal temporal profiles (Extended Data Fig. 2e–l). In the vast majority of recording sessions, the data are better fit by MMDDM than by the single-mode DDM (cross-validated; Fig. 4h,i). The model also accurately captures the choice data (Fig. 4j and Extended Data Fig. 6g) and reproduces vector fields that closely resemble those inferred from real spike trains (Extended Data Fig. 6h). Additional validations are shown in Extended Data Fig. 6i–n. Finally, because the end of the stimulus was fixed across trials relative to fixation onset, stimulus offset was not included as an input in MMDDM, consistent with the lack of abrupt neural changes at stimulus offset (Extended Data Fig. 9).

## nTc

In MMDDM, the transition from evidence accumulation to decision commitment and a consequent switch from $w_{EA}$ to $w_{DC}$ directly implement a change in neural mode between the two phases of the trial, which was previously suggested[9,41]. However, it remains unclear whether this neural mode change corresponds to the animal making up its mind, in part because no method has been developed previously to precisely estimate its timing in single trials. The behavioural DDM, without neural data, can provide a rough estimate of the moment of commitment (Fig. 5a, dashed grey line). But on the basis of the hypothesis that the time of the neural mode change corresponds to the time of commitment and, using data from many simultaneously recorded neurons, MMDDM allows a far more precise estimate per trial (Fig. 5a, orange line). We refer to this moment as nTc. Surprisingly, nTc varied widely across trials. It was not time locked to stimulus onset (Fig. 5b), stimulus offset (Extended Data Fig. 7n) or the onset of the decision-reporting motor response (Fig. 5c). Instead, nTc seemed to be an internally timed event. nTcs also occurred well after the onset of perimovement kernels inferred from generalized linear models of single-neuron spike trains[40] (Extended Data Fig. 8), indicating that nTcs do not reflect the initiation of action plan encoding.

A core prediction of the hypothesis that nTc marks the time of internal decision commitment is that, after nTc, auditory click stimuli should stop influencing the behavioural choice, because the animal will have already made up its mind. The single-trial estimates of nTc that MMDDM provides can be used to test this prediction: we time align the sensory stimulus data of each trial to the neurally estimated nTc and then behaviourally measure the weight with which stimulus fluctuations at each time point affect choice (that is, the psychophysical kernel[42]; Methods). Remarkably, as predicted, we found that the psychophysical weight of stimulus fluctuations on the choice of the animal diminished abruptly to zero after nTc (Fig. 5d and Extended Data Fig. 7). Because these commitment times varied widely across trials (Fig. 5b,c), the abrupt drop in psychophysical weight cannot be observed without the single-trial nTc estimates. If we instead align trials to the stimulus onset, we obtain a smooth psychophysical kernel (Extended Data Fig. 7e–h), as observed in previous studies lacking access to nTc[24].

nTc showed further hallmarks of being a marker of commitment: First, for a given evidence strength, trials without commitment are predicted to be more likely to involve noise acting against the sensory evidence, leading to lower accuracy. Consistent with this prediction, accuracy was lower in trials when nTc could not be identified (Fig. 5e). Second, commitment should occur more often when evidence is stronger, and, accordingly, nTc was more frequently detected in

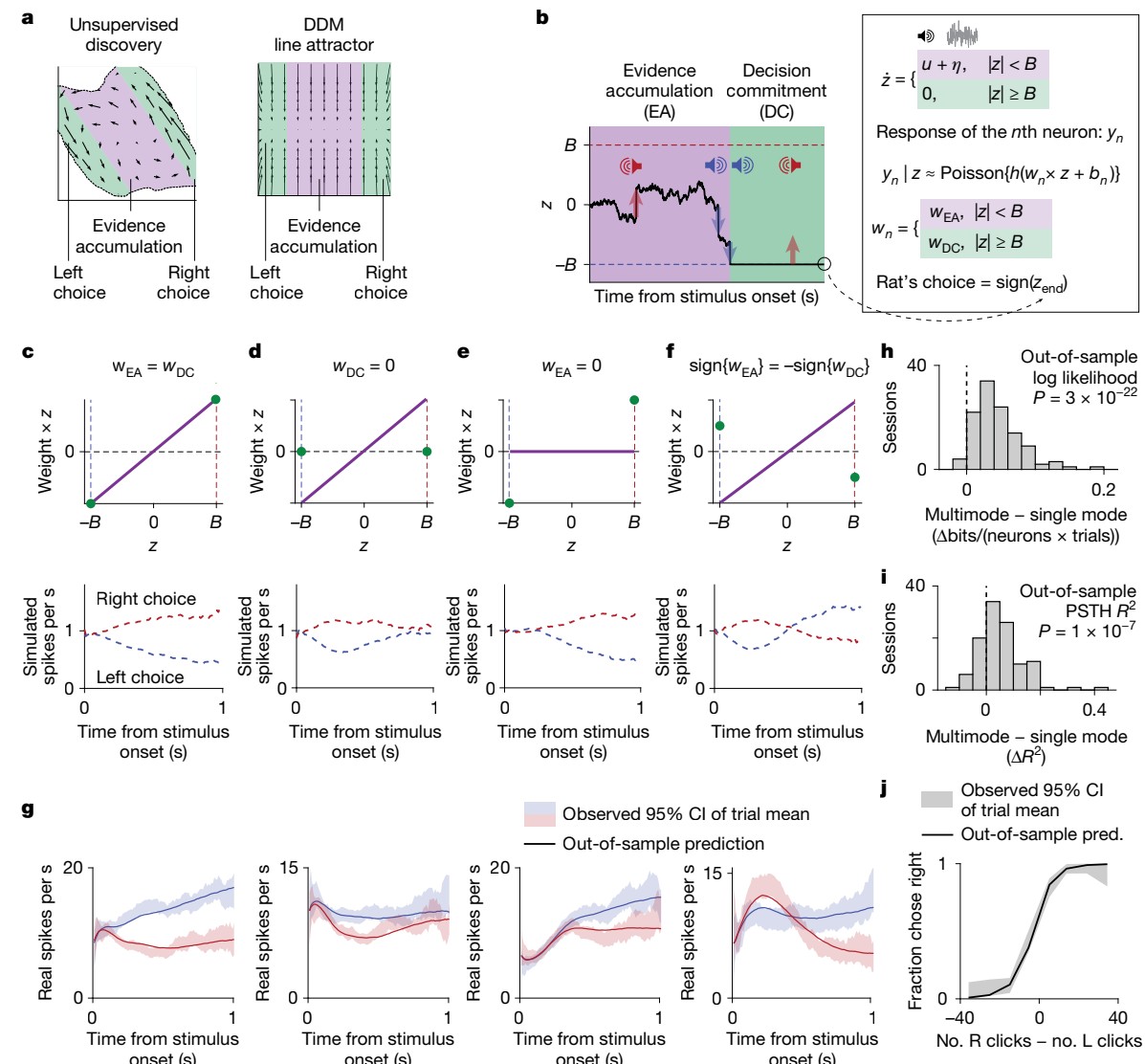

**Fig. 4 | A simplified model captures discovered dynamics and diverse neuronal profiles. a**, The velocity vector field of both the discovered dynamics and the DDM line attractor can be partitioned into evidence accumulation (EA) and decision commitment (DC) regimes. **b**, The MMDDM, a simplified model of the discovered dynamics. As in the behavioural DDM, momentary evidence ($u$) and noise ($\eta$) accumulate over time in the decision variable ($z$) until $z$ reaches either the left ($-B$) or right ($+B$) bound. At this moment, the animal commits to a decision: $z$ becomes fixed and unresponsive to further input. Also at this moment, the encoding weight ($w$) of each neuron shifts from $w_{EA}$ to $w_{DC}$, changing how $z$ maps to the predicted Poisson firing rate $y$ through softplus nonlinearity $h$ and baseline $b$. **c**, MMDDM captures heterogeneous single-neuron profiles. A ramp PSTH arises when $w_{EA}$ and $w_{DC}$ are equal. **d**, A decay

profile emerges when $w_{DC}$ is zero because, over time, more trials reach the bound where encoding of $z$ vanishes. **e**, A delay profile results from setting $w_{EA}$ to zero because, early in the trial, it is unlikely to have reached the bound. **f**, 'Flip' is produced by setting $w_{EA}$ and $w_{DC}$ to have opposite signs. **g**, MMDDM captures heterogeneity in single-neuron temporal profiles. Shading represents 95% bootstrap CI of the mean; the solid line is the model prediction. **h**, MMDDM has a higher out-of-sample likelihood than a one-dimensional DDM without a neural mode switch. **i**, MMDDM achieves a higher goodness-of-fit $R^2$ value of the choice-conditioned PSTHs. **h,i**, P values were computed using two-sided sign tests. **j**, Model prediction (pred.) and observed psychometric function for one example session. The shaded areas are the 95% bootstrap CI of the mean; the solid line is the model prediction.

trials with stronger evidence (Fig. 5f). Additional hallmarks are shown in Extended Data Fig. 7i–q. Together, these results offer behavioural support for an internally timed commitment event, after which sensory inputs are ignored, and the timing of which can be inferred from spiking data using nTc.

## Abrupt and gradual shifts at commitment

Perceptual decision-making involves a diversity in the temporal profiles of choice-selective neurons, with some showing a ramp-to-bound profile, others exhibiting a step-like profile and some falling in between a ramp and a step[3,4]. We found that the continuum of ramping and stepping profiles can be captured by a rapid reorganization in population

activity at the time of decision commitment, as described by MMDDM. We grouped neurons by whether they were estimated to be more, less or similarly engaged in evidence accumulation relative to decision commitment ($|w_{EA}| > |w_{DC}|$, $|w_{EA}| \approx |w_{DC}|$ and $|w_{EA}| < |w_{DC}|$, respectively, in MMDDM fits). We then computed the pericommitment neural response time histogram (PCTH) of each neuron (Methods and Fig. 6a,b). For neurons similarly engaged in accumulation and commitment, the PCTH had a ramp-to-bound profile, whereas, for neurons more engaged in commitment, the PCTH resembled a step. For neurons more engaged in accumulation, the PCTH had a ramp-and-decline profile. Even without grouping neurons, we found that the first three principal components (PCs) of the PCTHs correspond to the ramp-to-bound, step and ramp-and-decline profiles.

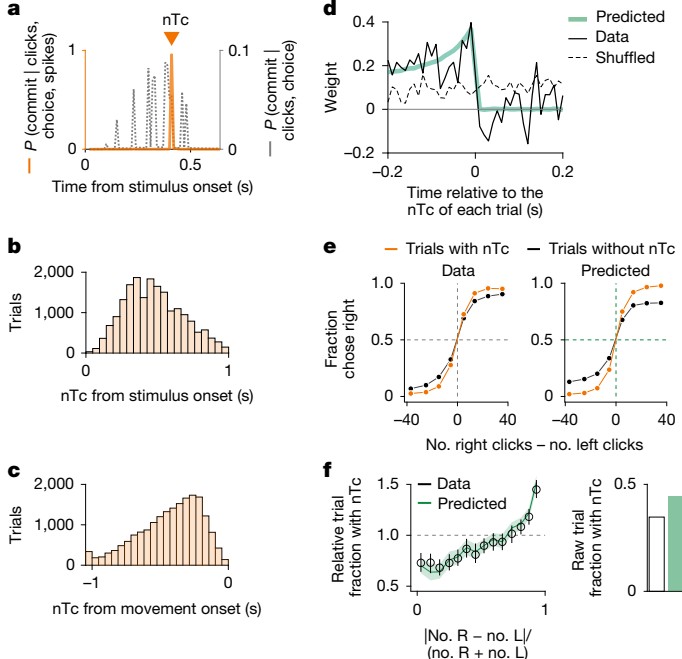

**Fig. 5 | nTc marks the moment of internal decision commitment. a**, Example trial. The inferred time of commitment is far more precise when neural activity is used (nTc, orange line) than when it is inferred solely from sensory stimulus timing and choice behaviour (dashed grey line). **b**, Distribution of estimated nTc values relative to stimulus onset. Among the 34.7% of trials in which commitment times could be detected, nTc varied widely relative to the onset of auditory click trains. The decline in nTc frequency over time reflects randomized stimulus durations (0.2–1.0 s). **c**, Distribution of nTc values relative to movement onset to report the decision of the animal (exiting centre fixation port). As in **b**, nTc timing also varies widely across trials. The leftmost bin includes trials in which the nTc occurred more than 1 s before movement. **d**, Supporting the interpretation of nTc as a decision commitment and, despite the highly variable timing of nTc, sensory evidence presented before nTc affects the decision of the animal but evidence presented after nTc does not (weight of clicks on choice inferred using logistic regression). Trials for which the estimated time of commitment occurred at least 0.2 s before stimulus offset and 0.2 s or more after stimulus onset were included for this analysis (9,397 of 55,057 trials across 115 sessions with 12 rats). The green line is the prediction from the MMDDM model fit to the data. **e**, Behavioural accuracy was lower in trials in which nTc could not be identified. Predictions were made by fitting MMDDM to the data, simulating trials from the fitted models and applying the same nTc detection procedure as that used for real data. Dashed reference lines at abscissa = 0 and ordinate = 0.5. **f**, nTc was more likely to be identified in trials with stronger evidence. For each evidence strength bin, the fraction of trials with an identified nTc was divided by the overall trial fraction across all bins, which was lower in the data than in the model predictions. Black circles and green lines indicate the mean across sessions. Black error bars and green shading indicate the 95% bootstrap confidence of the mean. Dashed reference lines at ordinate = 1.0.

The abrupt changes at decision commitment seem inconsistent with smoothly curved trial-averaged trajectories in low-dimensional neural state space often observed in decision-making studies[2,9]. Similar phenomena are observed in our data: the trial-averaged trajectories for left and right choices do not separate from each other along a straight line but rather along curved arcs (Fig. 6c). These smoothly curving arcs may result from averaging over trajectories with an abrupt turn aligned to decision commitment, which occurs at different times across trials (Fig. 5b–d). Consistent with this account, the smooth curves in low-dimensional neural state space can be captured well by the out-of-sample predictions of MMDDM but not by a one-dimensional DDM without a neural mode switch (Fig. 6c). These results indicate that

the MMDDM, a simplified model of the discovered dynamics, can capture the widespread observation well of smoothly curved trial-averaged trajectories.

## Mode transitions across regions

Although we generally observed dynamics with a neural mode transition across several frontal cortical and striatal areas, quantitative differences could be observed across these regions. The choice selectivity (a measure, ranging from −1 to 1, of the difference in firing rates for right-versus-left-choice trials; Extended Data Fig. 2m) averaged across neurons had different temporal profiles across brain regions (Fig. 6e). Although mPFC neurons were most choice selective near the beginning, FOF neurons were most choice selective towards the end. We found that the difference in latencies to peak choice selectivity was linked to differences in relative neuron engagement in evidence accumulation and decision commitment. Neurons that were more strongly engaged in evidence accumulation ($w_{EA} > w_{DC}$) tended to have a shorter latency to peak selectivity than neurons that were more strongly engaged in decision commitment ($w_{DC} > w_{EA}$). This result indicates that differences in choice-related encoding across frontal cortical and striatal regions can be understood in terms of relative participation in evidence accumulation versus decision commitment (Fig. 6f,g).

## Discussion

How neural dynamics govern the formation of a perceptual choice has been long debated[1,2,5]. Here we suggest that, for decisions on the timescale of hundreds of milliseconds to seconds, an initial input-driven regime mediates evidence accumulation and a subsequent autonomous-dominant regime subserves decision commitment. This regime transition is coupled to a rapid change in the representation of the decision process by the neural population: the initial neural mode (that is, direction in neural space) representing evidence accumulation is largely orthogonal to the subsequent mode representing decision commitment. In this sense, it is reminiscent of other covert cognitive operations, such as attentional selection, that also involve a change in neural mode[43].

If this coupled transition in dynamical regime and neural mode indeed corresponds to the time of decision commitment, sensory evidence presented after the transition would have minimal impact on the decision of the animal, because the animal would have already committed to a particular choice. Behavioural analysis confirmed this prediction in the experimental data (Fig. 5d), leading us to conclude that the transition is indeed a signal for covert decision commitment. We refer to the estimate of the presence and timing of such a transition in each trial, which is based on the sensory stimulus and firing rates of simultaneously recorded neurons, as nTc.

We wondered how decisions end. In reaction time paradigms of perceptual decision-making, animals are trained to respond as soon as they make a decision. The moment the animal initiates its response is then used to operationally define when it commits to a choice[44,45]. In these paradigms, decision commitment is overt, as it is closely linked to the onset of the movement animals make to report their choice[45]. Here, by contrast, using an experimenter-controlled duration paradigm, we found a decision commitment signal (nTc) that is covert in the sense of occurring at a time highly variable with respect to the timing of the external motor action used to report the decision, which it can precede by as much a second or more (Fig. 5c). It is also highly variable with respect to stimulus onset (Fig. 5b) or offset (Extended Data Fig. 7n). It is thus an internal signal, largely defined by coordination across neurons, not by its timing with respect to external events. The pericommitment neural responses observed here contrast sharply with the ramp-and-burst neural responses observed in animals trained

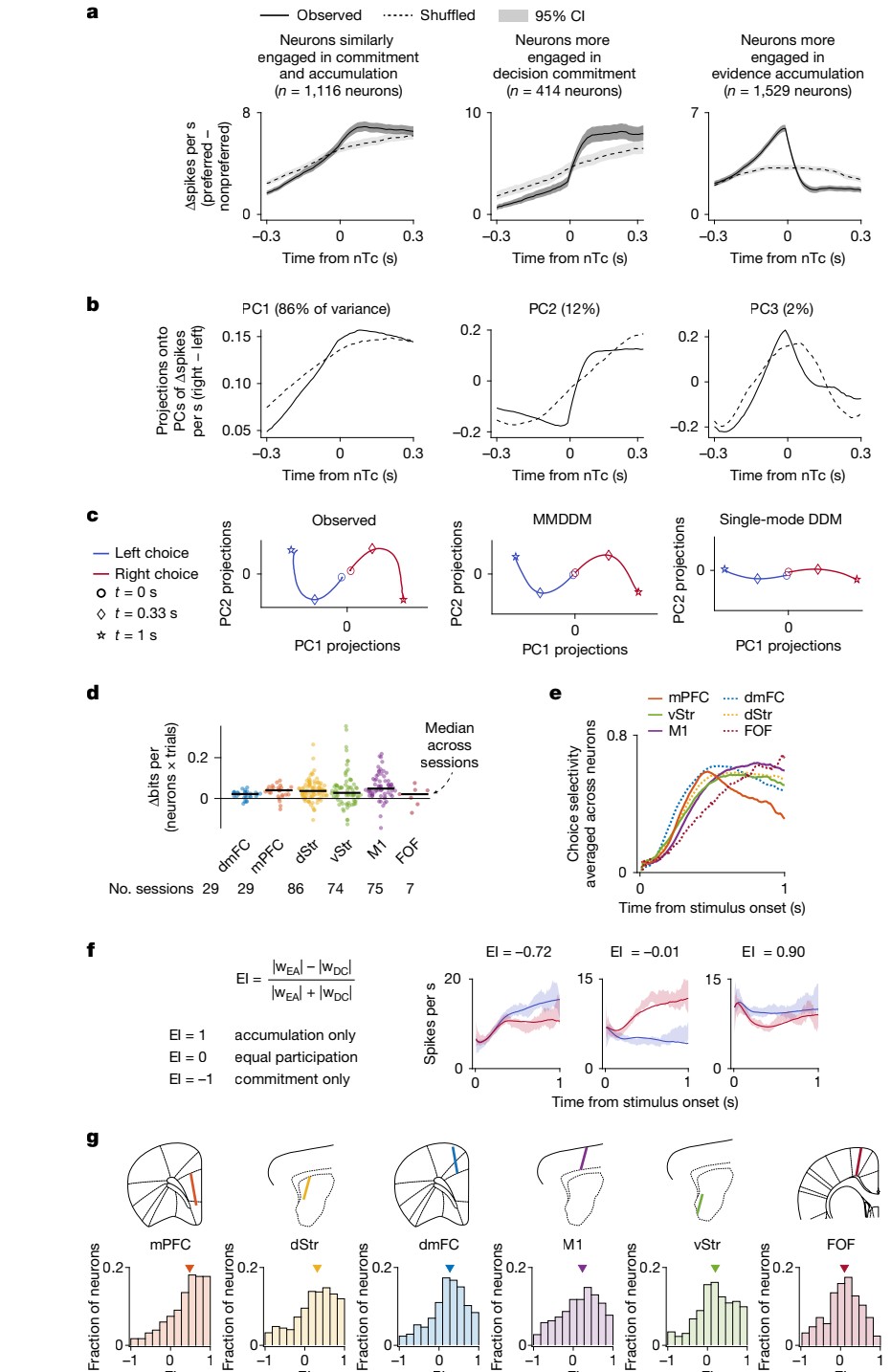

**Fig. 6 | Simplified model captures heterogeneous single-neuron temporal profiles, such as ramping and stepping, and shows functional distinctions between brain regions. a**, PCTHs for neurons grouped by relative engagement (defined in **f**). Neurons similarly engaged in evidence accumulation and decision commitment have ramp-to-bound profiles (centre). Neurons more engaged in decision commitment have step-like profiles (right), whereas those more engaged in evidence accumulation have ramp-and-decline profiles (left). 'Preferred' indicates the choice eliciting higher firing. Data are the mean across neurons and the 95% CI. **b**, First three PCs of PCTH differences (preferred − nonpreferred choice) across all neurons, capturing ramp-to-bound (PC1), step (PC2) and ramp-and-decline (PC3) profiles. **c**, Observed curved trial-averaged trajectories (projected onto the first two PCs) are captured by the MMDDM (centre) but not the single-mode DDM (right). Time from stimulus onset. Proj., projection. **d**, MMDDM better captures the data than the single-

mode DDM (out-of-sample log likelihood: MMDDM − single-mode DDM). **e**, The neuron-averaged choice selectivity has different temporal profiles across brain regions: mPFC neurons are most choice selective near the beginning, whereas FOF neurons are most choice selective towards the end. **f**, Engagement index (EI) quantifying relative neuronal engagement in evidence accumulation versus decision commitment. PSTHs are shown for three example neurons. Shading is the 95% CI of the mean; line indicates model prediction. **g**, A gradient across brain regions in the strength of neural mode transitions from stronger engagement in accumulation (for example, mPFC) to more balanced engagement (for example, FOF). Marker indicates median. Overall differences in engagement index across regions were assessed using the Kruskal–Wallis test ($P = 1 \times 10^{-44}$). Post hoc pairwise comparisons using the Tukey–Kramer test yielded $P < 0.001$ for mPFC versus dStr, dmFC, M1 and ventral striatum; dStr versus M1 and FOF; and dmFC versus FOF (exact $P$ values are in the Supplementary Notes (section 2.1).

to couple their decision commitment with response initiation[45] in a reaction time task.

Although the timing of the nTc signal reported here makes it very distinct from motor execution, the signal is also distinct from action preparation or planning. The beginning of action planning carries no implication as to whether sensory evidence presented subsequently will or will not be ignored. Indeed, in perceptual decision-making tasks, preliminary action preparation, driven by choice biases induced by previous trials, is often observed to begin even before the sensory stimulus, as reported previously[40] and found in our own data (Extended Data Fig. 8). By contrast, commitment to a decision suggests that evidence presented subsequently to the commitment will no longer affect the choice of the animal. Here we found that nTc corresponds to such a decision commitment moment. This was the case both at the neural level, in which it correlates with a substantial decrease in the effect of sensory inputs on neural responses in the regions we recorded (Fig. 2), and at the whole-organism behavioural level, in the sense that sensory evidence before nTc affects the choices of the animal but sensory evidence after nTc does not (Fig. 5d).

Although the behavioural DDM is a widely used model of decision-making, other frameworks are also prevalent, such as the linear ballistic accumulator[46] or urgency gating[47]. It is notable that the dynamics inferred by FINDR, obtained in a data-driven, unsupervised manner from spike times and auditory click times alone, resulted in regimes that match the characteristics of the behavioural DDM but not those of the alternatives. This match led us to explore a simplified model, the MMDDM, in which a scalar latent decision variable evolves as in the DDM but is represented in different neural modes before versus after decision commitment. The neural mode change indicates that a downstream decoder of the categorical choice can improve its accuracy by selectively reading out from neurons with post-commitment weights large in magnitude. A possible mechanism for the neural mode change is an input from ascending midbrain neurons, which is suggested by a recent finding in a working memory task that midbrain neurons, in response to an external auditory cue, trigger rapid reorganization of motor cortex activity to switch from planning-related activity to a motor command that initiates movement in mice[48].

We found that the MMDDM provides a parsimonious explanation of a variety of experimental findings from several species: across primates and rodents, sensory inputs and choice are represented in separate neural dimensions[2,9,40] across time, and neither sensory responses nor the neural dimensions for optimal decoding of the choice are fixed[9]. These phenomena, along with other observations including diversity in single-neuron dynamics[39,40], curved average trajectories[9], choice behaviour[24] and some vigorously debated phenomena such as a variety of single-neuron ramping versus stepping temporal profiles[3,4], are all captured by the MMDDM. However, we do not see MMDDM as a unique or a unified model of perceptual decision-making. Rather, we see it as a simple yet useful approximation, a minimally modified DDM, and a stepping stone towards a unified model of decision-making.

Single-trial trajectories, in sum, filled out the two-dimensional latent space inferred by FINDR. But when averaged over trials of a given evidence strength (Fig. 2h), they evolved along a one-dimensional curved trajectory. Looking exclusively along this one-dimensional manifold, the dynamics resemble those of the bistable attractor hypothesis[1] (Fig. 1f) in the sense of a one-dimensional unstable point at the origin, with autonomous dynamics growing stronger the farther the system is from the origin. However, the bistable attractor hypothesis and the other two hypotheses in Fig. 1g,h posit a one-dimensional manifold of slow autonomous dynamics, along which evidence accumulation evolves and towards which other states are attracted[1,23]. By contrast, the FINDR-inferred dynamics (which are inferred from single trials, not averaged trials) suggest an initial two-dimensional manifold of slow autonomous dynamics. Sensory evidence inputs drive evidence accumulation along one of these slow dimensions. The other slow

dimension corresponds to the decision commitment axis, along which autonomous dynamics will become dominant later in the process. We wondered why there would be slow autonomous dynamics along this second dimension. We speculate that, during initial evidence accumulation, slow autonomous dynamics along the decision commitment axis provide a mechanism for inputs driven by non-sensory factors such as trial history[49] to influence choice independent of the accumulating sensory evidence.

The authors of one recently proposed method to infer autonomous dynamics, applied to data from a task that did not require accumulating evidence over time, proposed that variety across the tuning curves of individual neurons could lead to curved one-dimensional decision manifolds[14]. However, the authors' method cannot yet infer input dynamics, and thus data from tasks with evidence that arrives gradually over time cannot yet be analysed; such an extension would have to be realized before we can assess whether the curvature their approach could infer would correspond to the curvature we described here for accumulation of evidence. Importantly, inferring input dynamics in addition to autonomous dynamics was critical to our observation that a change in dynamical regime, from input dominated to autonomous dominated, seemed to coincide with the change in neural mode (Fig. 2). This observation was key for our hypothesis that this event (nTc) could correspond to decision commitment, for development of the MMDDM simplified model to estimate nTc and for experimental confirmation that nTc is indeed the moment when sensory evidence ceases to affect the decision of the animal (Fig. 5d).

Finally, our approach expands the classic repertoire of techniques used to study perceptual decision-making. We inferred decision dynamics directly from neural data rather than assuming a specific hypothesis, and we took steps to enhance the human interpretability of the discovered dynamics: the unsupervised method (FINDR) focuses on low-dimensional rather than high-dimensional decision dynamics, and the mapping from latent to neural space (before the activation function of each neuron) preserves angles and distances. On the basis of key features of the inferred latent dynamics, we developed a highly simplified, tractable model (MMDDM) that is directly relatable to the well-known DDM framework. We found that the MMDDM, despite its simplicity, could describe a broad variety of previously observed phenomena and allowed us to infer the internal decision commitment times of the animal in each trial. Pairing deep-learning-based unsupervised discovery with simplified, parsimonious models may be a promising approach for studying not only perceptual decision-making but also other complex phenomena.

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

# Methods

## Experiments

**Animals.** The animal procedures described in this study were approved by the Princeton University Institutional Animal Care and Use Committee and were carried out according to the standards of the National Institutes of Health (NIH). Animals consisted of 16 adult, 6–24-month-old, male Long–Evans rats (*Rattus norvegicus*, Hilltop Lab Animals, Taconic) that were housed in Technoplast cages in pairs with a 12-h reversed light–dark cycle. All training and testing procedures were performed during the dark cycle. The rats had free access to food, but they had restricted access to water. The amount of water that the rats obtained daily was at least 3% of their body weight. Sample sizes were chosen on the basis of previous electrophysiological studies in rats[28,29]. No blinding or randomization was performed.

**Behavioural task.** Rats performed the behavioural task in custommade training enclosures (Island Motion) placed inside sound- and light-attenuated chambers (IAC Acoustics). Each enclosure consisted of three straight walls and one curved wall in which three nose ports were embedded (one in the centre and one on each side). Each nose port also contained one light-emitting diode that was used to deliver visual stimuli, and the front of the nose port was equipped with an infra-red beam to detect the entrance of the nose of the rat into the port. A loudspeaker was mounted above each of the side ports and used to present auditory stimuli. Each of the side ports also contained a silicone tube that was used for water reward delivery, with the amount of water controlled by valve-opening time.

Rats performed an auditory discrimination task in which optimal performance required the gradual accumulation of auditory clicks[24]. At the start of each trial, rats inserted their nose in the central port and maintained this placement for 1.5 s (fixation period). After a variable delay of 0.5–1.3 s, two trains of randomly timed auditory clicks were presented simultaneously: one from the left speaker and one from the right speaker. At the beginning of each click train, a click was played simultaneously from the left and right speakers (stereoclick). Regardless of onset time, the click trains ended at the end of the fixation period, resulting in stimuli ranging from 0.2 s to 1 s. The train of clicks from each speaker was generated by an underlying Poisson process, with different click rates for each side. The combined mean click rate was fixed at 40 Hz, and trial difficulty was manipulated by varying the ratio of the generative click rate between the two sides. The generative click rate ratio varied from 39:1 (easiest) to 26:14 (most difficult) clicks per s. At the end of the fixation period, the rats could orient towards the nose port on the side where more clicks were played and obtain a water reward.

Psychometric functions were calculated by grouping the trials into eight bins of similar size according to the difference in the total number of right and left clicks and, for each group, computing the fraction of trials ending in a right choice. The CI of the fraction of right responses was computed using the Clopper–Pearson method.

**Electrophysiological recording.** Neurons were recorded using chronically implanted Neuropixels 1.0 probes that are recoverable after the experiment[50]. In four animals, a probe was implanted at 4.0 mm anterior to the bregma and 1.0 mm lateral, for a distance of 4.2 mm, and at an angle of 10° relative to the sagittal plane that intersects the insertion site (the probe tip was more medial than the probe base). In five other animals, a probe was implanted to target M1, the dStr and the ventral striatum at the site 1.0 mm anterior and 2.4 mm lateral, for a distance of 8.4 mm, and at an angle of 15° relative to the coronal plane intersecting the insertion site (the probe tip was more anterior than the probe base). In a final set of three rats,

a probe was implanted to target the FOF and anterior dStr at 1.9 mm anterior and 1.3 mm lateral, for a distance of 7.4 mm, and at an angle of −10° relative to the sagittal plane intersecting the insertion site (the probe tip was more lateral than the probe base). Spikes were sorted into clusters using Kilosort2 (ref. 51), and clusters were manually curated.

**Muscimol inactivation.** Infusion cannulas (Invivo1) were implanted bilaterally over the dmFC (4.0 mm AP, 1.2 mm ML) in three rats. After the animal recovered from surgery, the animal was anaesthetized, and, on alternate days, a 600-nl solution of either only saline or muscimol (up to 150 ng) was infused in each hemisphere. Half an hour after the animal woke up from anaesthesia, the animal was allowed to perform the behavioural task.

**Retrograde tracing.** To characterize anatomical inputs into the dStr, 50 nl of cholera toxin subunit B conjugate (Thermo Fisher Scientific) was injected into the dStr at 1.9 mm AP, 2.4 ML and 3.5 mm below the cortical surface. The animal was perfused 7 days after surgery.

**Histology.** The rat was fully anaesthetized with 0.4 ml ketamine (100 mg ml⁻¹) and 0.2 ml xylazine (100 mg ml⁻¹) intraperitoneally, followed by transcardial perfusion of 100 ml saline (0.9% NaCl, 0.3× PBS, pH 7.0 and 0.05 ml heparin at 10,000 USP units per ml) and finally transcardial perfusion of 250 ml of 10% formalin neutral buffered solution (Sigma, HT501128). The brain was removed and postfixed in 10% formalin solution for a minimum of 7 days. Sections (100 μm) were prepared on a Leica VT1200 S vibratome and mounted on Superfrost Plus glass slides (Fisher) with Fluoromount-G (SouthernBiotech) mounting solution and glass coverslips. Images were acquired on a Hamamatsu NanoZoomer under ×4 magnification.

## Autonomous and input dynamics

The class of dynamical systems we study here is specified by

$$\dot{\mathbf{z}} = F(\mathbf{z}, \mathbf{u}) \tag{2}$$

for some generic function $F$, with $\mathbf{z}$ the latent decision variable and $\mathbf{u}$ the external input to the system from the auditory clicks in the behavioural task. At each moment, there may be no click, a click from the left or a click from the right. When time is discretized to sufficiently short steps, $\mathbf{u}$ is one of three values:

$$\mathbf{u} = \begin{cases} [0; 0] = \mathbf{0} & \text{representing when there is no click,} \\ [1; 0] & \text{representing when there is a left click or} \\ [0; 1] & \text{representing when there is a right click.} \end{cases} \tag{3}$$

We define the autonomous dynamics of the system as

$$\dot{\mathbf{z}}_{\text{autonomous}} = F(\mathbf{z}, \mathbf{0}) \tag{4}$$

and the average input dynamics as

$$\dot{\mathbf{z}}_{\overline{\text{input}}} = p(\mathbf{u}|\mathbf{z})(F(\mathbf{z}, \mathbf{u}) - F(\mathbf{z}, \mathbf{0})) \tag{5}$$

and, specifically, the average left and right input dynamics as

$$\begin{aligned} \dot{\mathbf{z}}_{\overline{\text{left}}} &= p(\mathbf{u} = [1; 0]|\mathbf{z})(F(\mathbf{z}, [1; 0]) - F(\mathbf{z}, \mathbf{0})), \\ \dot{\mathbf{z}}_{\overline{\text{right}}} &= p(\mathbf{u} = [0; 1]|\mathbf{z})(F(\mathbf{z}, [0; 1]) - F(\mathbf{z}, \mathbf{0})). \end{aligned} \tag{6}$$

The sum of autonomous dynamics and average input dynamics is equal to the expected value of $\dot{\mathbf{z}}$ computed over the distribution $p(\mathbf{u}|\mathbf{z})$:

$$\mathbb{E}[\dot{\mathbf{z}}] = \sum_{\mathbf{u}} p(\mathbf{u}|\mathbf{z})F(\mathbf{z}, \mathbf{u})$$

$$= p(\mathbf{u} = \mathbf{0}|\mathbf{z})F(\mathbf{z}, \mathbf{0}) + p(\mathbf{u} = [1; 0]|\mathbf{z})F(\mathbf{z}, [1; 0])$$
$$\quad + p(\mathbf{u} = [0; 1]|\mathbf{z})F(\mathbf{z}, [0; 1])$$
$$= (1 - p(\mathbf{u} = [1; 0]|\mathbf{z}) - p(\mathbf{u} = [0; 1]|\mathbf{z}))F(\mathbf{z}, \mathbf{0})$$
$$\quad + p(\mathbf{u} = [1; 0]|\mathbf{z})F(\mathbf{z}, [1; 0]) + p(\mathbf{u} = [0; 1]|\mathbf{z})F(\mathbf{z}, [0; 1]) \quad (7)$$
$$= F(\mathbf{z}, \mathbf{0}) + p(\mathbf{u} = [1; 0]|\mathbf{z})(F(\mathbf{z}, [1; 0]) - F(\mathbf{z}, \mathbf{0}))$$
$$\quad + p(\mathbf{u} = [0; 1]|\mathbf{z})(F(\mathbf{z}, [0; 1]) - F(\mathbf{z}, \mathbf{0}))$$
$$= \dot{\mathbf{z}}_{\text{autonomous}} + \dot{\mathbf{z}}_{\overline{\text{left}}} + \dot{\mathbf{z}}_{\overline{\text{right}}}.$$

Figure 2c shows a plot of $\dot{\mathbf{z}}_{\text{autonomous}}$, and Fig. 2e shows a plot of $\dot{\mathbf{z}}_{\overline{\text{left}}}$ and $\dot{\mathbf{z}}_{\overline{\text{right}}}$. $F(\mathbf{z}, \text{left})$ is defined as $p(\mathbf{u} = [1; 0]|\mathbf{z})F(\mathbf{z}, [1; 0]) + (1 - p(\mathbf{u} = [1; 0]|\mathbf{z}))F(\mathbf{z}, \mathbf{0})$, and $F(\mathbf{z}, \text{right})$ is defined as $p(\mathbf{u} = [0; 1]|\mathbf{z})F(\mathbf{z}, [0; 1]) + (1 - p(\mathbf{u} = [0; 1]|\mathbf{z}))(F(\mathbf{z}, \mathbf{0}))$.

Because $p(\mathbf{u}|\mathbf{z}) = p(\mathbf{z}|\mathbf{u})p(\mathbf{u})/p(\mathbf{z})$ and $p(\mathbf{z})$ in general do not have an analytical form, we estimate $p(\mathbf{u}|\mathbf{z})$ numerically. To do this, we train FINDR[20] to learn $F$ and generate click trains for 5,000 trials in a way that is similar to how clicks are generated for the task performed by our rats. Next, we simulate 5,000 latent trajectories from the learnt $F$ and the generated click trains. We then bin the state space of $\mathbf{z}$ and ask, for a single bin, how many times the latent trajectories cross that bin in total and how many of the latent trajectories when crossing that bin had $\mathbf{u} = [1;0]$ (or $\mathbf{u} = [0;1]$). That is, we estimate $p(\mathbf{u} = [1;0]|\mathbf{z})$ with $\frac{\text{No. latent states with } \mathbf{u} = [1; 0] \text{ in the bin that covers } \mathbf{z}}{\text{No. latent states in the bin that covers } \mathbf{z}}$. For Fig. 2, because $\mathbf{z}$ is two dimensional, we use bins of eight-by-eight that cover the state space traversed by the 5,000 latent trajectories and weigh the flow arrows of the input dynamics with the estimated $p(\mathbf{u}|\mathbf{z})$. Similarly, for the background shading that quantifies the speed of input dynamics in Fig. 2, we use bins of 100-by-100 to estimate $p(\mathbf{u}|\mathbf{z})$ and apply a Gaussian filter with $\sigma = 2$ (in the units of the grid) to smooth the histogram. A similar procedure was performed for Extended Data Figs. 1 and 4 to estimate $p(\mathbf{u}|\mathbf{z})$ numerically.

**Speed of autonomous and input dynamics.** To compute the normalized difference in the speed of autonomous and input dynamics in Fig. 3c, similar to previous sections, we first generated latent trajectories from the learnt $F$ for 5,000 different trials with generative click rate ratios used in our experiments with rats. Next, we computed the magnitude of the autonomous dynamics $\|\dot{\mathbf{z}}_{\text{autonomous}}\|$ and the magnitude of the average input dynamics $(\|\dot{\mathbf{z}}_{\overline{\text{left}}}\| + \|\dot{\mathbf{z}}_{\overline{\text{right}}}\|)/2$ for each time point for each of the 5,000 trajectories and then averaged across the trajectories and across time periods defined in Fig. 3b to obtain Fig. 3c.

## FINDR

Detailed descriptions are provided in ref. 20. Briefly, to infer velocity vector fields (or flow fields) from the neural population spike trains, we used a sequential variational autoencoder called FINDR.

FINDR minimizes a linear combination of two losses: one for neural activity reconstruction ($\mathcal{L}_1$) and the other for vector field inference ($\mathcal{L}_2$). To reconstruct neural activity, FINDR uses a deep neural network $G$ that takes the spike trains of $N$ simultaneously recorded neurons $\mathbf{y}$ and the sensory click inputs $\mathbf{u}$ in a given trial to obtain the time derivative of the $d$-dimensional latent decision variable $\mathbf{z}$:

$$\mathbf{z}_{t+1} = \mathbf{z}_t + \Delta t G(\mathbf{z}_t, \mathbf{u}_{1:T}, \mathbf{y}_{1:T}) + \mathbf{\eta}_t, \; t = 1, 2, 3, \dots. \quad (8)$$

Here, $T$ is the number of time steps in a given trial, $\mathbf{u}_t$ is a two-dimensional vector representing the number of left and right clicks played in a time step ($\Delta t = 0.01$ s), $\mathbf{y}_t$ is an $N$-dimensional vector of the spike counts in a time step and $\mathbf{\eta}_t$ is noise drawn from $N(\mathbf{0}, \Delta t\Sigma)$ in each time step. $\Sigma$ is a $d$-dimensional diagonal matrix in which the diagonal elements need not be equal to each other. For each time step, FINDR infers the firing rates of $N$ simultaneously recorded neurons $\mathbf{r}_t$ from $\mathbf{z}_t$ with

$$\mathbf{r}_t = \text{softplus}(W\mathbf{z}_t + \mathbf{b}_t), \quad (9)$$

where softplus is a function approximating the firing rate–synaptic current relationship ($f$–$I$ curve) of neurons, $W$ is an $N \times d$ matrix representing the encoding weights and $\mathbf{b}_t$ is an $N$-dimensional vector representing the putatively decision-irrelevant baseline input. The baseline $\mathbf{b}_t$ is learnt before fitting FINDR using the procedure described in Baseline and in detail in the Supplementary Methods, section 1.2. The reconstruction loss is given by

$$\mathcal{L}_1 = -\sum_{t=1}^{T} \log \text{Poisson}(\mathbf{y}_t|\mathbf{r}_t). \quad (10)$$

For vector field inference, we parametrize the vector field $F$ with a gated feedforward neural network[20,32]:

$$\dot{\mathbf{z}} \approx \frac{\mathbf{z}_t - \mathbf{z}_{t-\Delta t}}{\Delta t} = F(\mathbf{z}_{t-\Delta t}, \mathbf{u}_t). \quad (11)$$

$F$ gives the discretized time derivative of $\mathbf{z}$. We find the vector field $F$ that captures the latent trajectories $\mathbf{z}$ inferred from $G$ in equation (8) by minimizing

$$\mathcal{L}_2 = \sum_{t=1}^{T} (F(\mathbf{z}_t, \mathbf{u}_t) - G(\mathbf{z}_t, \mathbf{u}_{1:T}, \mathbf{y}_{1:T}))^\top \Sigma^{-1}(F(\mathbf{z}_t, \mathbf{u}_t) \\ - G(\mathbf{z}_t, \mathbf{u}_{1:T}, \mathbf{y}_{1:T})). \quad (12)$$

The total loss that is minimized by FINDR is

$$\mathcal{L} = \mathcal{L}_1 + c\mathcal{L}_2, \quad (13)$$

where $c = 0.1$ is a fixed hyperparameter ($c = 0.0125$ in Extended Data Fig. 1g). FINDR minimizes $\mathcal{L}$ by using stochastic gradient descent to learn $W, \Sigma$, the parameters of the neural network representing $F$ and the parameters of the neural network $G$. It can be shown that $\mathcal{L}$ is an approximate upper bound on the marginal log likelihood of the data and that training FINDR this way is equivalent to performing inference and learning with a sequential auto-encoding variational Bayes algorithm that straightforwardly extends the standard auto-encoding variational Bayes algorithm[52].

After training, we plot the vector field (that is, a grid of $\dot{\mathbf{z}}$) using the learnt $F$ and generate FINDR-predicted neural responses using equation (9) and

$$\mathbf{z}_t = \mathbf{z}_{t-\Delta t} + \Delta t F(\mathbf{z}_{t-\Delta t}, \mathbf{u}_t) + \mathbf{\eta}_t. \quad (14)$$

Equation (14) is an Euler-discretized gated neural stochastic differential equation[20,32].

**Parameters.** The total number of free parameters $P$ of the FINDR model is given by

$$P = P_W + P_\Sigma + P_F + P_G,$$
$$P_W = N \times d,$$
$$P_\Sigma = d,$$
$$P_F \in \{90 + (64 + d)d, 150 + (104 + d)d, 300 + (204 + d)d\}, \quad (15)$$
$$P_G \in \{15, 900 + 300N + 100x + P_F, 61, 800 + 600N + 200x \\ + P_F, 243, 600 + 1, 200N + 400x + P_F\}.$$

$P_W$ is the number of parameters in the encoding weight matrix $W$, the dimensions of which are the number of neurons $N$ and latent dimensionality $d$. $P_\Sigma$ is the parameter count in the diagonal covariance $\Sigma$ of the additive Gaussian noise of the latent $\mathbf{z}$. The number of parameters in the neural networks parametrizing $F(P_F)$ and $G(P_G)$ are separate hyperparameters. Here, $x = \frac{P_F - d + d^2}{2d + 3}$.

**Hyperparameters.** The hyperparameters that were optimized ($P_F$, $P_G$ and $\alpha$) include the number of parameters of the network $F(P_F)$, the number of parameters of the network $G(P_G)$ and the learning rate $\alpha \in \{10^{-2}, 10^{-1.625}, 10^{-1.25}, 10^{-0.875}, 10^{-0.5}\}$. We identify the optimal values for these hyperparameters in a $3 \times 3 \times 5 = 45$ grid search. The grid search was performed separately for each set of training data for each of five crossvalidation folds. In each training set, three-quarters of the trials were used to the optimize the parameters under a given set of hyperparameters and the remaining one-quarter was held out to evaluate the model performance for that set of hyperparameters. Test data were never used in the grid search.

**Latent space transformation.** Because the encoding weight matrix $W$ is not constrained to semi-orthogonality and can take only any real values, different combinations of $W$ and $\mathbf{z}_t$ can give rise to the same firing rate vector $\mathbf{r}_t$, even when baseline $\mathbf{b}_t$ is fixed. To uniquely identify the latent trajectories (except for redundancy from rotations and reflections), after optimization, we linearly transformed the latent space $\mathbf{z}$ to $\tilde{\mathbf{z}}$:

$$\tilde{\mathbf{z}}_t = SV^{\top}\mathbf{z}_t, \tag{16}$$

where $S$ is a $d \times d$ diagonal matrix containing the singular values of $W$ and $V$ is a $d \times d$ matrix containing the right singular vectors

$$W = USV^{\top}. \tag{17}$$

$U$ is an $N \times d$ matrix containing the left singular vectors of $W$ (where $N$ is the number of neurons). In the space of $\tilde{\mathbf{z}}$, the encoding weight matrix is a linear transformation that preserves angles and distances because $U$ is semi-orthogonal and can only give rise to an isometry such as rotation and reflection.

$$\begin{aligned} W\mathbf{z} &= USV^{\top}\mathbf{z} \\ &= U\tilde{\mathbf{z}} \end{aligned} \tag{18}$$

To obtain meaningful axes for the transformed latent space $\tilde{\mathbf{z}}$, we generate 5,000 different trajectories of $\tilde{\mathbf{z}}$ in generative mode (that is, using $F$ and $\Sigma$ in equation (14) but not $G$) and perform PC analysis on the trajectories. The PCs were used to define the axes of the decision variable $\tilde{\mathbf{z}}$. In the main text, the PC1 axis of $\tilde{\mathbf{z}}$ was denoted as $z_1$ and the PC2 axis of $\tilde{\mathbf{z}}$ was denoted as $z_2$. In all our analyses, the latent trajectories and vector fields inferred by FINDR are shown in the transformed latent space of $\tilde{\mathbf{z}}$ and scaled such that the latent trajectories along PC1 lie between $-1$ and $1$.

**Sample zone.** In Figs. 2 and 3, to focus on the portion of the inferred vector field that is used by the single-trial trajectories, we show only the well-sampled subregion of the state space, which is the portion occupied by at least 50 of 5,000 simulated single-trial latent trajectories of 1 s. With this definition, the sample zone is the same across time points in Fig. 2h.

**Model evaluation.** The goodness of fit of the PSTH was quantified using the coefficient of determination ($R^2$) of the evidence–sign conditioned PSTH as defined in equation (34) using fivefold cross-validation. We used three-fifths of the trials in a session as the training dataset, one-fifth of the trials as the validation dataset to optimize the hyperparameters of FINDR and one-fifth of the trials as the test (that is, out-of-sample) dataset to evaluate performance of FINDR. Therefore, when we compute the goodness of fit, we also obtain five different vector fields inferred by FINDR for each fold, which we confirm are consistent across folds (Extended Data Fig. 4).

**Curvature of trial-averaged trajectories.** To compute the curvature of trial-averaged trajectories in Fig. 3b, as before, we first generate

latent trajectories from FINDR for 5,000 different trials with generative click rate ratios used in our experiments with rats. Next, we separate the trials on the basis of whether the generative click ratio in a given trial favours a leftward choice or a rightward choice. We take the average of the latent trajectories over the left-favouring trials and then convolve the trial-averaged trajectory with a Gaussian filter with $\sigma = 3$ (in units of the time step $\Delta t = 0.01$ s). We take this smoothed trajectory to numerically compute the planar curvature. We do the same for the right-favouring trials and take the average between the curvature obtained from left-favouring trials and the curvature obtained from right-favouring trials to generate the plot in Fig. 3b.

**cFINDR.** The cFINDR model replaces the neural network parametrizing $F$ in FINDR with a linear combination of affine dynamics, specified by $M$ and $N$, and bistable attractor dynamics specified by $\varphi$. The dynamics are furthermore constrained to be two dimensional.

$$\begin{aligned} \dot{\mathbf{z}} &\approx \frac{\mathbf{z}_t - \mathbf{z}_{t-\Delta t}}{\Delta t} = F(\mathbf{z}_{t-\Delta t}, \mathbf{u}_t) = M\mathbf{z}_{t-\Delta t} + N\mathbf{u}_t + s \times \varphi(\mathbf{z}_{t-\Delta t}), \\ M &= Q\Lambda Q^{-1}, \\ Q &= \begin{bmatrix} 1 & \sin(\theta) \\ 0 & \cos(\theta) \end{bmatrix}, \\ \Lambda &= \begin{bmatrix} 0 & 0 \\ 0 & -r \end{bmatrix}, \\ \varphi(\mathbf{z}_t) &= -\exp(-(\mathbf{z}_t - \mathbf{x})^2/\rho) \odot (\mathbf{z}_t - \mathbf{x}) \\ &\quad -\exp(-(\mathbf{z}_t + \mathbf{x})^2/\rho) \odot (\mathbf{z}_t + \mathbf{x}). \end{aligned} \tag{19}$$

The matrix $M$ implements a line attractor located at $z_2 = 0$. The inputs $\mathbf{u}_t$ are the same as those in FINDR and represent the auditory clicks. The two discrete attractors are constrained such that $x_2 = 0$ and implemented through the function $\varphi$. The shape of the basin of attraction corresponding to each point attractor is specified by the parameter $\rho$. The relative contribution of the discrete attractors and the line attractor to the overall dynamics is specified by the scalar $s$.

The DDM line attractor hypothesis can be implemented in cFINDR by setting $\theta = 0$. Non-normal dynamics with a line attractor[2] can be implemented by setting $\theta \neq 0$. The bistable attractor hypothesis can be implemented by increasing $\rho$.

As in FINDR, cFINDR learns $W$, $\Sigma$ and parameters of $G$. Instead of the neural networks parametrizing $F$, cFINDR, learns $s$, $\theta$, $r$, $\mathbf{x}$, $\rho$ and the $2 \times 2$ matrix $N$ to approximate $F$, which has nine parameters. The same objective function and optimization procedure were used in cFINDR. After optimization, as in FINDR, the latent space $\mathbf{z}$ is linearly transformed to uniquely identify the dynamics (except for arbitrary rotations or reflections). As in the analysis of results from FINDR, the latent trajectories and vector fields inferred by cFINDR are in the transformed latent space $\tilde{\mathbf{z}}$.

When we fit cFINDR to the data, we experimented with the different constraints $r > 0$ and $r > 3$. The fits using $r > 0$ were superior to those using $r > 3$ and were therefore used in the comparison between cFINDR and FINDR for the data presented in Fig. 3e,f. We were motivated to try both $r > 0$ and $r > 3$ because we found that, in synthetic data, cFINDR under the constraint $r > 0$ could not recover the dynamics generated under the DDM line attractor hypothesis ($r = 10$). For this reason, Extended Data Fig. 5f shows results from synthetic data using $r > 3$. When fit to data, FINDR outperforms cFINDR using either $r > 0$ or $r > 3$.

**FINDR models with more than two latent dimensions.** For Extended Data Fig. 3j,k, we evaluated FINDR models with more than two latent dimensions to assess whether the two-dimensional manifold we found is approximately an attractor. To show that the sample zone was an approximate attractor manifold, we perturbed the latent states on the manifold along the third PC direction. When the latent states were

perturbed (but not so far that the latent states went outside the range along the PC3 axis covered by the sample zone), the latent states flowed towards the manifold. To obtain the flow directions along PC3, we first generated 5,000 latent trajectories (similar to Fig. 2 for computing the sample zone). We then divided the PC1 × PC2 space into an eight-by-eight grid (the grid used for the vector field arrows in Extended Data Fig. 3i). For each cell in the grid, we identified the latent states from the 5,000 trajectories that were inside the cell and identified the highest (lowest) PC3 value $z_3^{\mathrm{up}}(z_3^{\mathrm{dn}})$. This was to ensure that the perturbation along the PC3 axis was not too large. Next, we computed the flow vector using a 100-by-100 grid on the PC1 × PC2 space, assuming that PC3 = $z_3^{\mathrm{up}}(z_3^{\mathrm{dn}})$ and PC4 = 0. The space covered by each cell of the grid is coloured on the basis of the direction of the flow vector along PC3: if flowing upwards, green; if flowing downwards, pink. A Gaussian filter was applied to this heat map with $\sigma = 2$ (in units of the 100-by-100 grid), similar to the heat map for input dynamics in Fig. 2f. The resulting plot is shown on the left (right) panel. Results were similar without the Gaussian filter.

**Choice decoding from FINDR.** FINDR does not use the choice of the animal for reconstructing neural activity. However, after training, we can fit a logistic regression model that predicts the choice of the animal from the decision variable $\mathbf{z}$ at the final time step $T$. When we fit an $\ell_2$-regularized logistic regression model using $\mathbf{z}_T$ from the trained network $G$ and the choice of the animal in the representative session in Fig. 2c–h, we found that the logistic choice decoder achieves 89.7% accuracy in predicting choice in the out-of-sample dataset. We can generate choices from this decoder by generating latent trajectories using $F$ and $\Sigma$ in equation (14) as in previous sections and by supplying $\mathbf{z}_T$ to the trained decoder. A total of 5,000 latent trajectories and choices generated from $F$ and the choice decoder were used for the analysis in Extended Data Fig. 4l. We used a separate logistic regression model for predicting choice from the latent trajectories truncated at time = 0.33 s projected onto PC2. Optimization of the logistic regression models was carried out using L-BFGS[53].

## MMDDM

The MMDDM is a state-space model, comprising a dynamic model that governs the time evolution of the probability distributions of latent (that is, hidden) states and measurement models that define the conditional distributions of observations (that is, emissions) given the latent state. Additional information is provided in the Supplementary Methods, section 1.3.

**Dynamic model.** The latent variable $z$ is one dimensional (that is, a scalar), and its time evolution is governed by a piecewise linear function:

$$z(t+1) = \begin{cases} z(t) + u(t) + \eta, & -B < z(t) < B \\ B \cdot \mathrm{sign}(z(t)), & \text{otherwise.} \end{cases} \tag{20}$$

When the absolute value of $z$ is less than the bound height $B$ (free parameter), its time evolution depends on momentary external input $u$ and i.i.d. (independent and identically distributed) Gaussian noise $\eta$.

$$\eta \sim \mathcal{N}(0, \Delta t), \tag{21}$$

where $\Delta t$ is the time step and set to 0.01 s. Here, $\sim$ means 'distributed as'. When $z$ is either less than $-B$ or greater than $B$, it becomes fixed at the bound. The initial probability distribution of $z$ is given by

$$z(t=1) \sim \mathcal{N}(\mu_0, 1), \tag{22}$$

where the mean $\mu_0$ is a free parameter. In time step $t$, the input $u(t)$ is the total difference in the per-click input $v$ between the right and left clicks that occurred in the time interval $(t - \Delta t, t)$:

$$u(t) = \sum_{\tau \in \mathrm{R}} v(\tau; t) - \sum_{\tau \in \mathrm{L}} v(\tau; t), \tag{23}$$

where L(R) is the set of the left (right) click times and $v(\tau; t)$ is the per-click input of a click occurring at time $\tau$ and time step $t$. Note that $\tau \in \mathbb{R}$ indicates continuous time, whereas $t \in \mathbb{N}$ indexes a time step. The per-click input is given by

$$v(\tau; t) = D(\tau; t) \cdot C(\tau) \cdot \zeta, \tag{24}$$

where $D(\tau; t)$ indicates the integral over the interval $[t - \Delta t, t)$ of the Dirac delta function $\delta$ delayed by $\tau$:

$$D(\tau; t) = \int_{t-\Delta t}^{t-\varepsilon} \delta(x - \tau) dx = \begin{cases} 1, & \tau \in [t - \Delta t, t) \\ 0, & \text{otherwise,} \end{cases} \tag{25}$$

where $\varepsilon$ is the machine epsilon. To account for sensory adaptation, the per-click input is depressed by preceding clicks by a time-varying scaling factor given by the function $C(\tau)$, implemented according to previous work[24] (Supplementary Methods, section 1.3.1). The per-click input is corrupted by i.i.d. multiplicative Gaussian noise $\zeta$:

$$\zeta \sim \mathcal{N}(1, \sigma_s^2). \tag{26}$$

The free parameter $\sigma_s^2$ is the variance of the per-click noise. Variability in the dynamic model is fit to the data through the per-click noise $\zeta$ rather than per-time step noise $\eta$ on the basis of previous findings[24]; our results are similar if we set the variance of $\eta$ rather than the variance of $\zeta$ as a free parameter.

The dynamic model has three free parameters: bound height $B$, variance $\sigma_s^2$ of the per-click noise and mean $\mu_0$ of the initial state. These parameters are learnt simultaneously with the parameters of the measurement models.

**Measurement model of behavioural choices.** In each trial, the binary behavioural choice $c$ (1, right; 0, left) is the sign of $z$ in the last time step $T$ of the trial (the earlier of 1 s after the onset of the clicks or immediately before the animal leaves the fixation port):

$$c|z(T) = \mathrm{sign}(z(T)). \tag{27}$$

**Measurement model of spike counts.** In each time step $t$, given the value of $z$, the spike count $y$ of neuron $n$ is a Poisson random variable

$$y^{(n)}(t)|z(t) \sim \mathrm{Poisson}(\lambda^{(n)}(t)\Delta t). \tag{28}$$

The firing rate $\lambda$ is given by

$$\lambda^{(n)}(t)|z(t) = h(w^{(n)} \cdot z(t) + b(t)), \tag{29}$$

where $h(\cdot)$ is the softplus function used to approximate the neuronal frequency–current curve of a neuron:

$$h(x) = \log(1 + \exp(x)). \tag{30}$$

The encoding weight $w$ depends on $z$ itself:

$$w^{(n)} = \begin{cases} w_{\mathrm{EA}}^{(n)}, & -B < z < B \\ w_{\mathrm{DC}}^{(n)}, & z \in \{-B, B\}. \end{cases} \tag{31}$$

Each neuron has two scalar weights, $w_{\mathrm{EA}}$ and $w_{\mathrm{DC}}$, that specify the encoding of the latent variable during the evidence accumulation regime and the decision commitment regime, respectively. When the latent variable has not yet reached the bound ($-B$ or $B$), all simultaneously recorded neurons are in the evidence accumulation regime and

encode the latent variable through their own private $w_{EA}$. When the bound is reached, all neurons transition to the decision commitment regime and encode $z$ through their own $w_{DC}$.

The bias $b$ accounts for factors that are putatively independent of the decision, including a component that varies only across trials and another component that varies both across and within trials:

$$b^{(n)}(m, t) = b^{(n)}_{\text{cross}}(m) + b^{(n)}_{\text{within}}(m, t). \tag{32}$$

The cross-trial trial component $b^{(n)}_{\text{cross}}$ is a function of time $m$ from the first trial of the session, whereas $t$ indicates time in each trial relative to the stimulus onset of that trial. The within-trial component consists of time-varying influence from spike history, post-stimulus (stim) onset and pre-movement (move) onset.

$$b_{\text{within}}(m, t) = \tau^{(m)}_{\text{stim}}(k_{\text{stim}} * \delta)(t) + \tau^{(m)}_{\text{move}}(k_{\text{move}} * \delta)(t) \\ + \sum_i \tau^{(m,i)}_{\text{spike}}(k_{\text{spike}} * \delta)(t), \tag{33}$$

where the symbol $*$ indicates convolution, $\tau_x$ indicates translation $\tau_x k(t) = k(t - \tau_x)$ by the time of event $x$ and $\delta$ is the Dirac delta function. The functions $b^{(n)}_{\text{cross}}, k_{\text{stim}}, k_{\text{move}}, k_{\text{spike}}$ are learnt, and each is parametrized as a linear combination of radial basis functions[40,54] (Supplementary Methods, section 1.5). The measurement model of each neuron of the spike train has 19 parameters that are learnt simultaneously with the parameters of the dynamic model (that is, the model of the latent variable).

**Parameter learning.** All parameters, including the three parameters of the latent variable and the 19 parameters private to each neuron, are learnt simultaneously by jointly fitting to all spike trains and choices using maximum a posteriori estimation. Gaussian priors were placed on the model parameters to ensure that the optimization reached a critical point and confirmed to not change the results in separate optimizations using maximum likelihood estimation (that is, optimization without Gaussian priors). Out-of-sample predictions were computed using fivefold cross-validation.

**nTc.** The time step when decision commitment occurred is selected to be when the posterior probability of the latent variable at either the left bound or the right bound, given the click times, spike trains and behavioural choice, is greater than 0.8. Results were similar for other thresholds, and the threshold of 0.8 was chosen to balance between prediction accuracy and the number of trials for which commitment was predicted to have occurred. Using this definition, commitment occurred in 34.6% of trials.

**Engagement index.** The engagement index was computed for each neuron to quantify its involvement in evidence accumulation and decision commitment. The index was defined using $w_{EA}$ and $w_{DC}$ of the neuron: $EI \equiv (|w_{EA}| - |w_{DC}|)/(|w_{EA}| + |w_{DC}|)$. It ranges from −1 to 1. A neuron with an engagement index of −1 encodes the latent variable only during decision commitment, an engagement index of 1 indicates involvement only during evidence accumulation, and an engagement index of 0 represents a similar strength of encoding the latent variable during evidence accumulation and decision commitment.

## Analyses
**Neuronal selection.** Only neurons that meet a preselected threshold for being reliably choice selective were included for analysis. For each neuron, reliable choice selectivity was measured using the area under the receiver operating characteristic curve (auROC) indexing how well an ideal observer can classify between a left-choice trial and a right-choice trial on the basis of neuronal spike counts. Spikes were counted in four non-overlapping time windows (0.01–0.21 s, 0.21–0.4 s, 0.41–0.6 s and 0.61–0.9 s after stimulus onset), and an auROC was

computed for each time window. A neuron with an auROC < 0.42 or an auROC > 0.58 for any of these windows was considered choice selective and included for other analyses. Moreover, neurons must have had an average firing rate of at least two spikes per s. Across sessions, the median fraction of neurons included under this criterion was 10.4%.

**PSTH.** Spike times were binned at 0.01 s and were included up to 1 s after the onset of the auditory stimulus (click trains) until 1 s after the stimulus onset or until the animal removed its nose from the central port, whichever came first. The time-varying firing rate of each neuron in each group of trials (that is, task condition) was estimated with a PSTH, which was computed by convolving the spike train on each trial with a causal Gaussian linear filter with a standard deviation of 0.1 s and a width of 0.3 s and averaging across trials. The CI of a PSTH was computed by bootstrapping across trials.

The goodness of fit of the model predictions of the PSTH was quantified using the coefficient of determination ($R^2$), computed using fivefold crossvalidation. $R^2$ was computed by conditioning the PSTH on either the sign of the evidence (that is, whether the generative click ratio in a given trial favoured a leftward choice or a rightward choice) or the choice of the animal:

$$R^2 = 1 - \frac{\text{SS}_{\text{res}}}{\text{SS}_{\text{tot}}}$$
$$\text{SS}_{\text{res}} = \sum_t ((\text{PSTH}^R_{\text{obs}}(t) - \text{PSTH}^R_{\text{pred}}(t))^2 \\ + (\text{PSTH}^L_{\text{obs}}(t) - \text{PSTH}^L_{\text{pred}}(t))^2)$$
$$\text{SS}_{\text{tot}} = \sum_t ((\text{PSTH}^R_{\text{obs}}(t) - \mathbb{E}_t[\text{PSTH}^R_{\text{obs}}(t)])^2 \\ + (\text{PSTH}^L_{\text{obs}}(t) - \mathbb{E}_t[\text{PSTH}^L_{\text{obs}}(t)])^2), \tag{34}$$

where $t$ is time in a trial that goes from 0 s to 1 s, with 0 s being the stimulus onset. The superscripts '$R$' and '$L$' indicate either the sign of the difference in the total number of right and left clicks or the choice of the animal. The subscripts '$_{\text{obs}}$' and '$_{\text{pred}}$' indicate whether the PSTH was computed using observed neural activity or model-predicted neural activity. $\text{SS}_{\text{res}}$ is the residual sum of squares, and $\text{SS}_{\text{tot}}$ is the total sum of squares.

A normalized PSTH was computed by dividing the PSTH by the mean firing rate of the corresponding neuron across all time steps across all trials. When PSTHs were separated by 'preferred' and 'null', the preferred task condition was defined as the group of trials with the behavioural choice when the neuron responded more strongly and a null task condition was defined as the trials associated with the other choice.

**Choice selectivity.** In Fig. 6 and Extended Data Fig. 2m, for each neuron and for each time step $t$ aligned to the onset of the auditory click trains, we computed choice selectivity $c(t)$:

$$c(t) \equiv \frac{r(t) - l(t)}{r(t^*) - l(t^*)}, \tag{35}$$

where $r$ and $l$ are the PSTHs computed from trials ending in a right choice and a left choice, respectively. The time step $t^*$ is the time of the maximum absolute difference:

$$t^* \equiv \text{argmax}_t |r(t) - l(t)|. \tag{36}$$

In Extended Data Fig. 2m, neurons are sorted by the centre of mass of the absolute value of the choice selectivity of each neuron.

**Baseline.** In FINDR, cFINDR and MMDDM, the neuronal firing rate depends on a time-varying scalar baseline. In time step $t$ of trial $m$, conditioned on the value of the latents in a given time step, the spike count $y$ of each neuron is given by

$$y(m, t)|\mathbf{z}(m, t) \sim \text{Poisson}(h\{\mathbf{w}^\top \mathbf{z}(m, t) + b(m, t)\}), \qquad (37)$$

where $h$ is the softplus function and $\mathbf{w}$ is the encoding weight of the latent. The baseline $b$ incorporates putatively decision-independent variables as input to the neural spike trains including slow drifts in firing rates across trials and faster changes in each trial that are aligned to either the time from stimulus onset or the time from the animal leaving the fixation port. The baseline is learnt using a Poisson generalized linear model fit separately to the spike counts of each neuron. Details are provided in the Supplementary Methods, section 1.2.

**PCTH.** In trials for which a time of decision commitment (nTc) could be inferred, the spike trains were aligned to the predicted time of commitment and then averaged across those trials. The trial average was then filtered with a causal Gaussian kernel with a standard deviation of 0.05 s. The PCTHs were averaged in each of three groups of neurons: (1) neurons that were similarly engaged in evidence accumulation and decision commitment; (2) neurons more strongly engaged in evidence accumulation; and (3) neurons more strongly engaged in decision commitment. Each neuron was assigned to one of these groups according to its engagement index. Neurons with $-\frac{1}{3} \leq \text{EI} < \frac{1}{3}$ are considered to be similarly engaged in evidence accumulation and decision commitment, neurons with $\text{EI} \geq \frac{1}{3}$ are considered to be more strongly engaged in evidence accumulation, and those with $\text{EI} < -\frac{1}{3}$ are considered to be more strongly engaged in decision commitment.

For this analysis, we focused on only the 65 of 115 sessions for which the MMDDM improved the $R^2$ of the PSTHs and for which the inferred encoding weights were reliable across cross-validation folds ($R^2 > 0.9$). From this subset of sessions, there were 1,116 neurons similarly engaged in evidence accumulation and decision commitment, 414 neurons that were more engaged in decision decision commitment and 1,529 neurons that were more engaged in evidence accumulation.

To compute the shuffled PCTH, the predicted times of commitment were shuffled among only the trials in which commitment was detected. If the randomly assigned commitment time extended beyond the length of the trial, then the time of commitment was assigned to be the last time step of that trial.

**Trial-averaged trajectories in neural state space.** To measure trial-averaged dynamics in neural state space, we analysed PCs in a data matrix made by concatenating the PSTHs. The data matrix $X$ has dimensions $TC$-by-$N$, where $T$ is the number of time steps ($T = 100$), $C$ is the number of task conditions ($C = 2$ for choice-conditioned PSTHs and $C = 4$ for PSTHs conditioned on both choice and evidence strength) and $N$ is the number of neurons. The mean across rows is subtracted from $X$, and singular value decomposition is performed: $USV^\top = X$. The principal axes correspond to the columns of the right singular matrix $V$, and the projections of the original data matrix $X$ onto the principal axes correspond to the left singular matrix ($U$) multiplied by $S$, the rectangular diagonal matrix of singular values. The first two columns of the projections $US$ are plotted as trajectories in neural state space.

**Psychophysical kernel.** Kernels were time locked to either nTc of each trial (Fig. 5d and Extended Data Fig. 7a–d) or the first click in each trial (Extended Data Fig. 7e–h). We extended the logistic regression model presented in ref. 55 to include a lapse parameter (Supplementary Methods, section 1.4), and we confirmed that results were similar using generic logistic regression. A shuffling procedure was used to randomly permute the inferred time of commitment across trials without changing the behavioural choice and the times of the auditory clicks on each trial. In this randomly permuted sample, we selected trials for which the auditory stimuli were playing at least 0.2 s before and at least 0.2 s after the inferred time of commitment to compute the psychophysical kernel in the shuffled condition. For Fig. 5d, the prediction was generated using the MMDDM parameters that were fit to the data and the

same set of trials in the data. For Extended Data Fig. 7, temporal basis functions were used to parametrize the kernel, and the optimal number and type of basis function were selected used crossvalidated model comparison.

**Statistical tests.** Binomial CIs were computed using the Clopper–Pearson method. All other CIs were computed with a bootstrapping procedure using the bias-corrected and accelerated percentile method[56]. Unless otherwise specified, $P$ values comparing medians were computed using a two-sided Wilcoxon rank-sum test, which tests the null hypothesis that two independent samples are from continuous distributions with equal medians against the alternative hypothesis that they are not.

**Estimating the low-dimensional vector field without specifying a dynamical model.** For Extended Data Fig. 10d, we estimated the low-dimensional velocity vector field for each session using a method that does not specify a dynamical model (model-free approach). To obtain the model-free vector field, we first estimated single-trial firing rates of individual neurons by binning the spike trains into bins of $\Delta t = 10$ ms and convolving the spike trains with a Gaussian of $\sigma = 100$ ms centred at 0. Results were similar for other values of $\sigma$ around 100 ms. Next, for each neuron, we took the average across all trials in the session and subtracted this average from single-trial firing rate trajectories. These baseline-subtracted firing rate trajectories were then projected to the low-dimensional subspace spanned by the FINDR latent axes. We projected the estimated firing rates to the same subspace as FINDR to allow direct comparisons between the FINDR-inferred vector field and the model-free vector field.

We treated this low-dimensional projection of the baseline-subtracted firing rates as the latent trajectories in this model-free approach. To obtain velocity vector fields from the latent trajectories, we first estimated the instantaneous velocity $\dot{\mathbf{z}}$ at time point $t$ by computing $\dot{\mathbf{z}}_t = (\mathbf{z}_t - \mathbf{z}_{t-\Delta t})/\Delta t$ for all $t$ for all latent trajectories. We then divided the two-dimensional latent space into an eight-by-eight grid. For each cell $(i, j)$ from this eight-by-eight grid, we identified all states $\mathbf{z}_t$ from all trajectories that fell inside the cell $(i, j)$. We took the corresponding $\dot{\mathbf{z}}_t$ of the identified $\mathbf{z}_t$ values and took the average to compute the velocity for the cell $(i, j)$. We computed velocity vectors for all 64 cells. To compare vector fields, we took the cosine similarity between the velocity vector for cell $(i, j)$ from FINDR and the velocity vector for cell $(i, j)$ from the model-free approach and took the mean of these cosine similarities, $S_c(\text{FINDR, model free})$. In computing $S_c(\text{FINDR, model free})$, only cells that had a number of states greater than 1% of the total number of states were included. When the number of states used to estimate the velocity vector was less than 1% of the total number of states, we considered that cell $(i, j)$ to be outside the sample zone, analogous to the sample zone in Fig. 2.

To compare between a random vector field and the model-free vector field, we generated 1,000 random vector fields (with each of the 64 arrows in the eight-by-eight grid going in random directions) for each session and computed $S_c(\text{random, model free})$ for each random vector field.

For Extended Data Fig. 10e, we estimated the autonomous dynamics vector field around the origin as a model-free way of confirming our findings in Extended Data Fig. 10a. Similar to the method for Extended Data Fig. 10d, we convolved the spike trains with a Gaussian and projected the baseline-subtracted firing rate trajectories to the low-dimensional subspace spanned by the FINDR latent axes. However, to separate autonomous dynamics from input dynamics, we used a Gaussian with a smaller $\sigma$ (20 ms), with a window size $\pm 3\sigma$ around 0, and then excluded any $\dot{\mathbf{z}}_{t \pm 3\sigma}$ with time $t$ for which a click occurred from the estimation of the autonomous dynamics. When computing the average of $(\mathbf{z}_t - \mathbf{z}_{t-\Delta t})/\Delta t$ for one of the five pie slices, we required $\mathbf{z}_{t-\Delta t}$ to be inside the pie slice. For all sessions, the circle had a radius of 0.2

(in units of $z$). To further ensure that we estimated the autonomous dynamics, when computing the average, we only considered the trajectories for which the number of left clicks was equal to the number of right clicks during the epoch when they were in the pie slice.

## Inclusion and ethics statement

The animal procedures described in this study were approved by the Princeton University Institutional Animal Care and Use Committee.

## Reporting summary

Further information on research design is available in the Nature Portfolio Reporting Summary linked to this article.

## Data availability

The experimental data that support the findings of this study are available at Dryad[57] (https://doi.org/10.5061/dryad.sj3tx96dm).

## Code availability

Custom acquisition, postprocessing and analysis code is available at GitHub (https://github.com/Brody-Lab/decision_dynamics_commitment). Code implementing FINDR[20] is available at GitHub (https://github.com/Brody-Lab/findr/). Code implementing MMDDM is available at GitHub (https://github.com/Brody-Lab/fhmddm), and code for baseline estimation is available at GitHub (https://github.com/Brody-Lab/tzl_spglm).

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

**Acknowledgements** We thank J. Charlton, L. Ding, J. Gold, R. Goris, J. Maunsell and J. Pillow for their suggestions and comments; and J. Morrison, K. Osario, J. Teran and E. Valance for technical assistance. Analyses reported in this work were largely accomplished using the Princeton Research Computing resources at Princeton University, which is a consortium of groups led by the Princeton Institute for Computational Science and Engineering and the Office of Information Technology's Research Computing. This work was supported by grants from the US NIH (F32 MH115416, R01MH108358, R01MH138935 and 5U19NS132720) and the Simons Foundation (SF 542953). This study is the result of funding in whole or in part from the NIH. It is subject to the NIH Public Access Policy. By accepting this federal funding, we have given the NIH the right to make this paper publicly available in PubMed Central upon the official date of publication, as defined by the NIH.

**Author contributions** T.Z.L. and D.G. collected data. T.D.K. developed the FINDR model. T.Z.L. developed the MMDDM model. T.Z.L. and T.D.K. developed the baseline model used in fitting FINDR and MMDDM. T.Z.L. and T.D.K. performed analyses. A.G.B. and C.D.K. assisted with data collection. V.A.E. and B.D. assisted with analyses. T.Z.L., T.D.K. and C.D.B. wrote the manuscript and conceptualized the study.

**Competing interests** The authors declare no competing interests.

**Additional information**
**Correspondence and requests for materials** should be addressed to Thomas Zhihao Luo, Timothy Doyeon Kim or Carlos D. Brody.

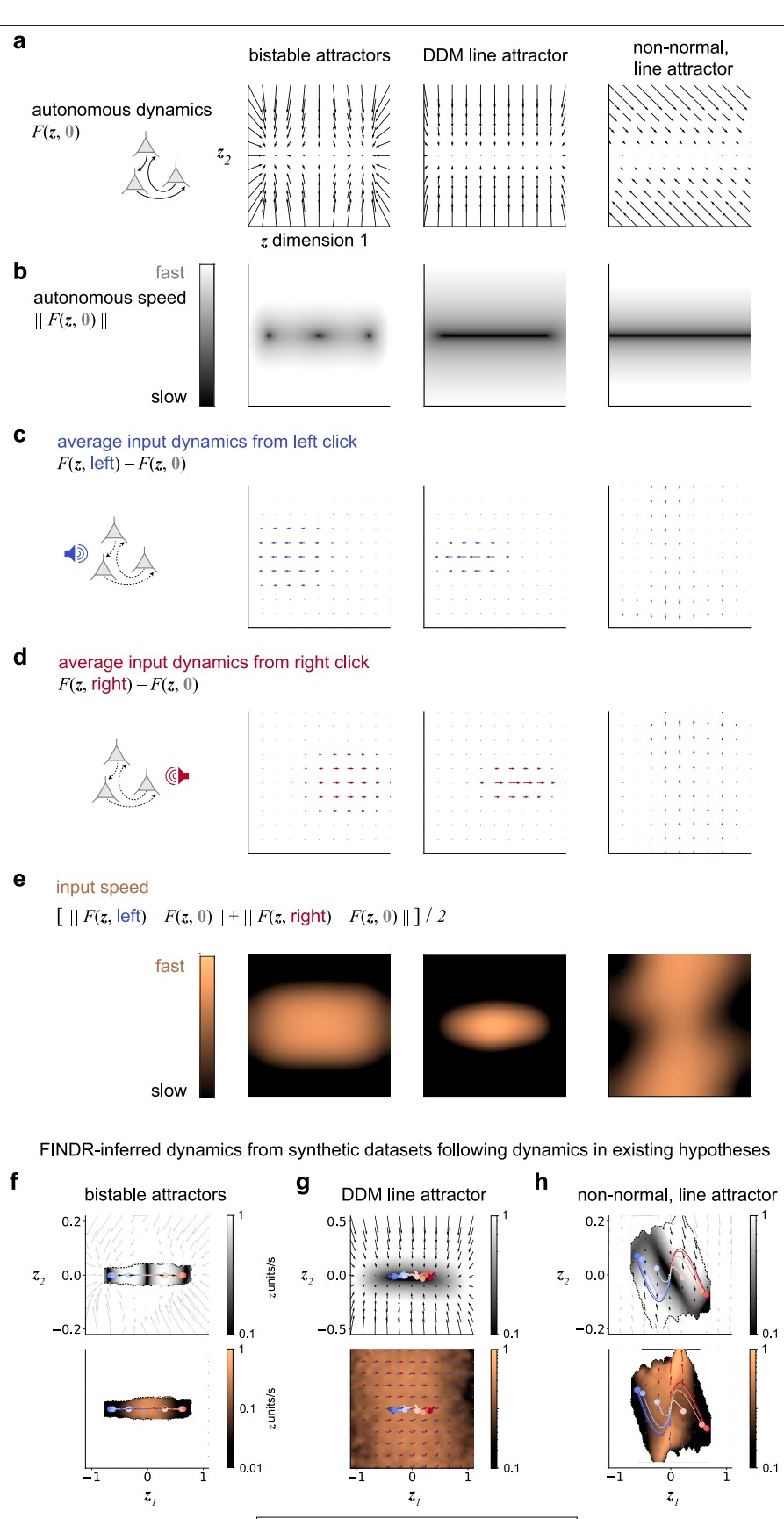

**a** bistable attractors    DDM line attractor    non-normal, line attractor

autonomous dynamics
$F(z, 0)$

$z_2$

$z$ dimension 1

**b** fast
autonomous speed
$\| F(z, 0) \|$
slow

**c** average input dynamics from left click
$F(z, \text{left}) - F(z, 0)$

**d** average input dynamics from right click
$F(z, \text{right}) - F(z, 0)$

**e** input speed
$[\ \| F(z, \text{left}) - F(z, 0) \| + \| F(z, \text{right}) - F(z, 0) \|\ ] / 2$

fast

slow

FINDR-inferred dynamics from synthetic datasets following dynamics in existing hypotheses

**f** bistable attractors

**g** DDM line attractor

**h** non-normal, line attractor

sample zone    left ——•——•——•—— right
evidence strength

**Extended Data Fig. 1** | See next page for caption.

**Extended Data Fig. 1 | FINDR can be used to distinguish between the dynamical systems hypotheses of perceptual decision-making.** In these hypotheses, the decision process is represented by the state of a dynamical system, which we refer to as the "decision variable ($z$)" and is depicted as two-dimensional here but may have fewer or more dimensions. An attractor is a set of states for which the dynamical system tends to move toward, from a variety of starting states. When $z$ is in an attractor state, small perturbations away from the attractor tend to return the system toward the attractor. An attractor can implement the commitment to a choice and the maintenance of the choice in working memory. **a**, In all these hypotheses, the attractors are implemented by the autonomous dynamics, which corresponds to the deterministic dynamics $F$ in the absence of inputs and depends only on $z$ itself. In the bistable attractors hypothesis, there are two discrete attractors, each of which corresponds to a choice alternative. In the DDM line attractor hypothesis, the autonomous dynamics form not only two discrete attractors but also a line attractor in between. The intervening line attractor allows an analog memory of the accumulated evidence when noise is relatively small. In the line attractor hypothesis with non-normal dynamics, the autonomous dynamics form a line attractor, and a separate readout mechanism is necessary for the commitment to a discrete choice. **b**, The autonomous speed is the magnitude of the autonomous dynamics. A dark region corresponds to a steady state, which can be an attractor, repeller, or saddle point. In the bistable attractors hypothesis, the left and right steady states are each centered on an attractor, and the middle is a saddle point. In both the DDM line attractor hypothesis and the hypothesis that has non-normal dynamics with a line attractor, the steady states correspond to attractors. **c-d**, Input dynamics corresponding to a left and right auditory pulse, respectively. Here we show the "effective" input dynamics, which is multiplied by the frequency $p(u|z)$ to account for the pulsatile nature and the statistics of the stimuli in our task (in contrast to Fig. 1e, in which the input dynamics were presented without the multiplication of the frequency, which is appropriate for stimuli that are continuous over time). Whereas in the bistable attractor and DDM line attractor, the inputs are aligned to the attractors, in the hypothesis that has non-normal dynamics with a line attractor, the inputs are not aligned. **e**, The input speed is the average of the magnitude of the average left input dynamics and the magnitude of the average right input dynamics. **f**, We simulated spikes that follow the bistable attractor dynamics in **a-e** to create a synthetic dataset with the number of trials, number of neurons, and firing rates that are typical of the values observed in our datasets. Then, we fit FINDR to this synthetic dataset from random initial parameters. The autonomous and input dynamics inferred by FINDR qualitatively match the bistable attractors hypothesis. **g-h**, FINDR-inferred dynamics qualitatively match the dynamics in Fig. 1f–h and a–e. In panel **g**, the sample zone covers the entirety of the plotted area.

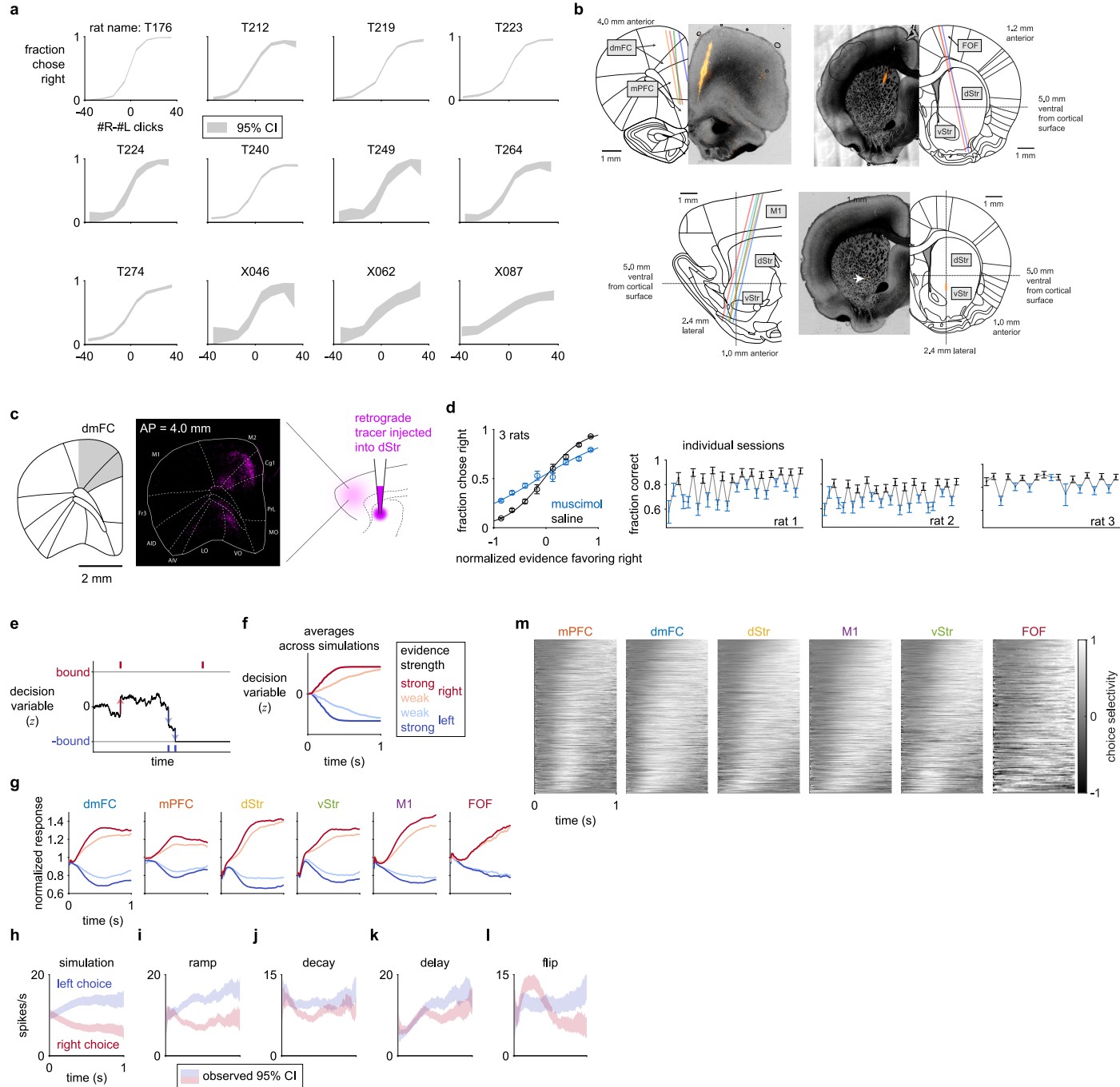

**Extended Data Fig. 2 | Behavioral performance, histological slices, anatomical tracing, causal necessity of dmFC, and the temporal profiles of individual neurons not being not consistent with an one-dimensional neural encoding of the latent variable in the drift-diffusion model (DDM).** **a**, Psychometric functions of each of the twelve rats recorded aggregated across recording sessions. **b**, Histological images of probe tracks. Each color indicates a probe chronically implanted in a rat. **c**, Dorsomedial frontal cortex (dmFC) provides a major input to the anterior dorsal striatum (dStr). **d**, dmFC is causally necessary for the auditory decision-making task studied here. N = 23,298 saline trials and 22,428 muscimol trials. Error bars indicate 95% binomial confidence intervals. **e**, In the DDM, noisy inputs are accumulated over time through a scalar latent variable ($z$) until the value of $z$ reaches a fixed bound, which triggers the commitment to a choice. **f**, In simulations of the DDM, $z$ ramps quickly when the evidence strength is strong and more slowly

when the strength is weak. **g**, Responses averaged across both trials and neurons resemble the trajectories of $z$ averaged across simulations. Only choice-selective neurons were included. Spikes after the animal began movement (i.e., removed its nose in the center port) were excluded. For this analysis only, error trials were excluded. N = 1324 (dmFC), 1076 (mPFC), 1289 (dStr), 714 (vStr), 822 (M1), 163 (FOF). **h**, The responses of a simulated neuron encoding the DDM with a single neural mode show the ramping dynamics. Shading indicates the bootstrapped 95% confidence interval of the trial-mean of the filtered response. **i**, A neuron with a ramp profile. **j**, A neuron recorded from the session with choice selectivity that decays over time. **k**, A neuron exhibiting a substantial delay in its choice selectivity. **l**, A neuron whose choice selectivity flips in sign. **m**, The diversity of the temporal profile of the choice selectivity of individual neurons is not consistent with a one-dimensional encoding of the DDM.

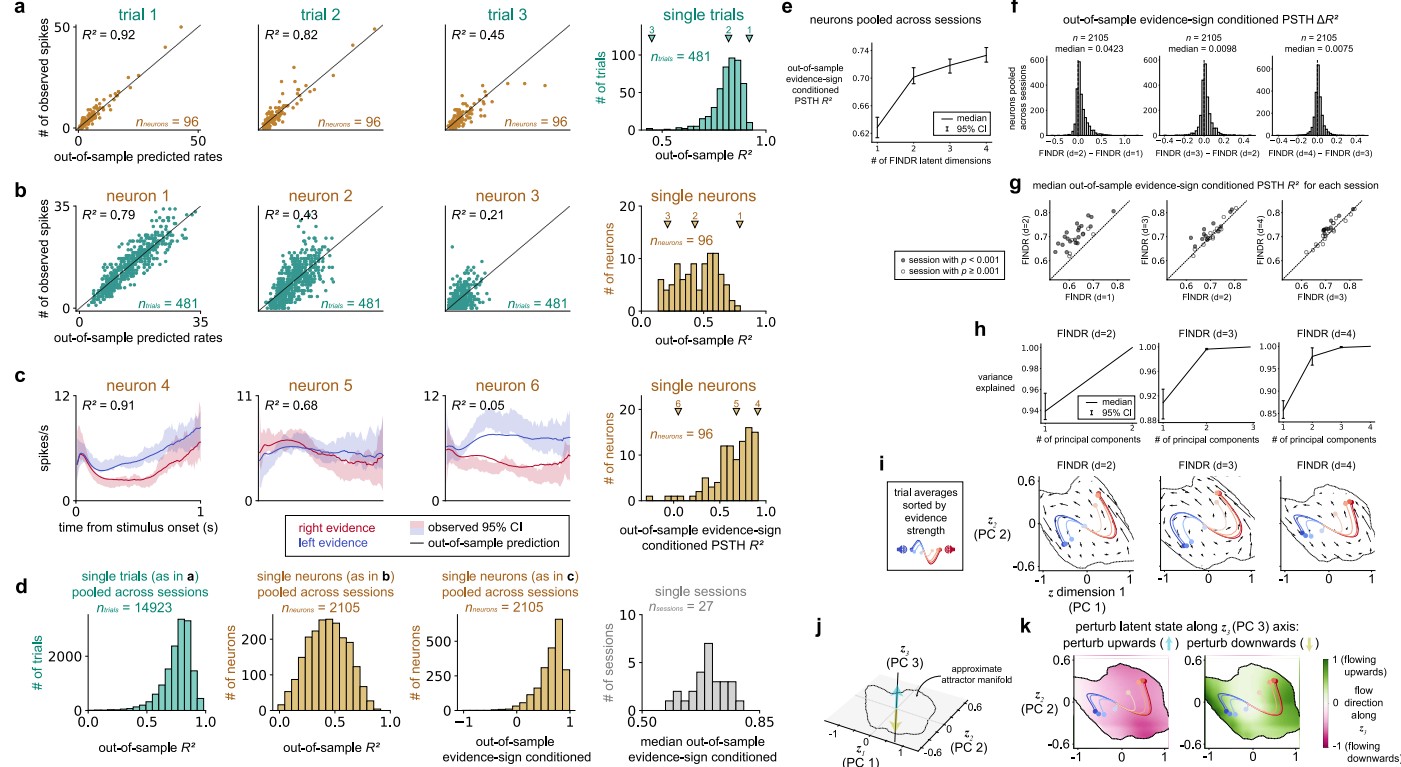

**Extended Data Fig. 3 | FINDR can well capture the neural responses and reveals 2-dimensional decision-making dynamics. a-b,** FINDR captures the underlying firing rates of the single-trial responses of individual neurons from the representative session in Fig. 2. **c,** FINDR captures the complex trial-averaged dynamics of individual neurons from the representative session in Fig. 2 as can be seen in the peristimulus time histograms (PSTH). The goodness-of-fit is measured using the coefficient of determination ($R^2$). Bold line indicates out-of-sample prediction by FINDR, and the shading indicates 95% confidence interval from the observed PSTH. **d,** FINDR captures the single-trial and trial-averaged responses of individual neurons pooled across 34 sessions. For the histogram showing single trials pooled across sessions, 34 trials that had $R^2 < 0$ are not shown. Results in **a-d** are 5-fold cross-validated. **e,** Across different FINDR models with latent dimensions (d) ranging from 1 to 4, we computed the median of the coefficient of determination ($R^2$) of the evidence-sign conditioned peri-stimulus time histogram (PSTH) of neurons pooled across sessions ($n = 2105$). **f,** The median difference in the $R^2$ between d = 2 and d = 1 is significantly different from zero (p < 0.001; Wilcoxon signed-rank test). Although the median differences are also significant for the comparison between d = 3 and d = 2 and the comparison between d = 4 and d = 3, the magnitude of the difference is relatively small (0.0098 and 0.0075, respectively) compared to the median difference between d = 2 and d = 1 (0.0423). **g,** We repeated the analysis in **f** without pooling neurons across sessions. Instead,

for each session, we computed the median PSTH $R^2$ across neurons recorded within that session. Each circle corresponds to a session, and a filled circle indicates a significant difference in the PSTH $R^2$ between FINDR models of different dimensionalities (p < 0.001; two-sided Wilcoxon signed-rank test; Supplementary Information 2.2). **h,** For FINDR models with either 3 or 4 latent dimensions, more than 97% of the variance is captured by the first two principal components (PC's). PCA was done separately for each session, and the error bars indicate the 95% confidence interval of the median across sessions ($n = 27$). **i,** For models with 2 or more dimensions, the vector fields and trajectories projected onto the first two dimensions are qualitatively similar. The vector fields and trajectories were shown for the representative session in Fig. 2. The dashed lines demarcate the well-sampled subregion of the state space (i.e., the sample zone). **j,** We evaluated FINDR models with latent dimensions higher than two to see whether the two-dimensional manifold relevant to decision-making dynamics is an approximate attractor manifold. The variance explained by the third PC in the FINDR model with three-dimensional latent dynamics was less than 0.5% (as shown in **h-i**), so we turned to the FINDR model with four-dimensional dynamics. In this model, the variance explained by the third PC was around 1.3%. We perturbed the latent states on the manifold along the PC 3 direction. **k,** When the latent states are perturbed (but not too far that the latent states go outside the range along the PC 3 axis covered by the sample zone; see Methods for details), the latent states flow toward the manifold.

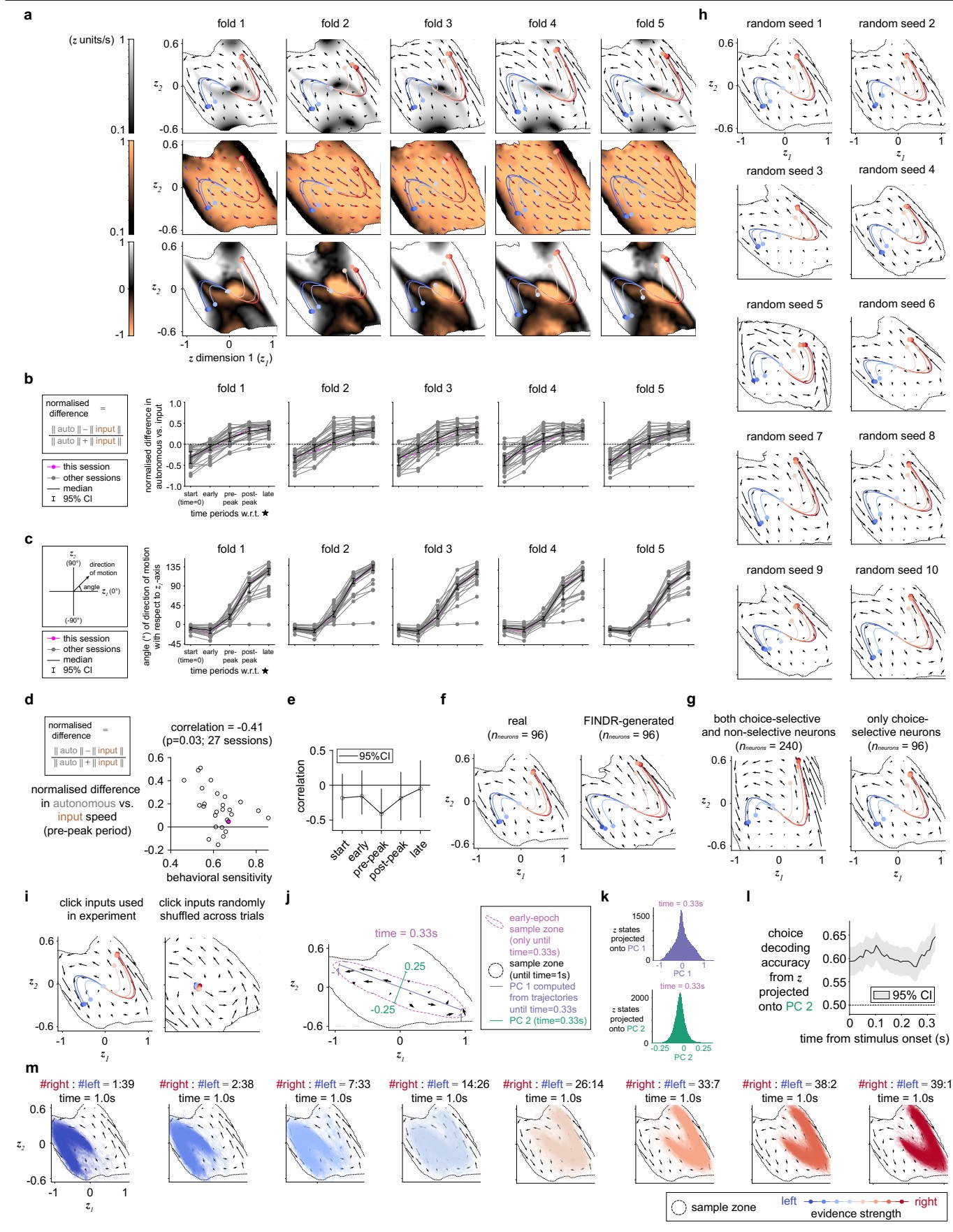

**Extended Data Fig. 4** | See next page for caption.

**Extended Data Fig. 4 | Consistency in FINDR-inferred dynamics. a**, FINDR-inferred input and autonomous dynamics are consistent across 5 different cross-validation folds as shown for the same session in Fig. 2. **b**, Normalized difference in the speed between autonomous and input dynamics for five different time periods ("start (time=0 s)" "early", "pre-peak", "post-peak", and "late") is consistent across folds ($n$ = 27; see Fig. 3c). **c**, The direction of motion of the trial-averaged trajectories and its angle with respect to the $z_1$-axis for different time periods is consistent across folds ($n$ = 21 out of 27 sessions; see Fig. 3d). **d**, Variability in the dynamics across sessions depends in part on the variability in the behavioral performance. For each each session, behavioral sensitivity was estimated as the parameter $\beta$ in a probit model $p(y|x) = \Phi(\beta*x + c)$, where y is the rat's left vs. right choice on each trial, x the log-ratio of the right vs. left click rate on that trial, $\Phi$ the normal cumulative distribution function, c the constant term in the probit model. The two-sided p-value of the Pearson's correlation was computed using a Student's t-distribution for a transformation of the correlation. Pink marker indicates the example session. **e**, The linear correlation between the difference in autonomous vs. input dynamics and behavioral sensitivity was negative for all epochs, but reliable only for the pre-peak epoch. The 95% confidence intervals were computed by bootstrapping across sessions ($n$ = 27). **f**, FINDR reliably recovers the FINDR-inferred dynamics. After fitting FINDR to a dataset, the model parameters were used to simulate a synthetic dataset using the exact same set of sensory stimuli in the real dataset and containing the same number of neurons and trials. From new initial parameter values, FINDR was fit to the simulated data to infer the "FINDR-generated" vector fields. **g**, FINDR is fit to both choice-selective and non-selective neurons. We find similar dynamics to when FINDR is fit to only choice-selective neurons. **h**, We find vector fields that are consistent across multiple different random seeds that change the initialization in the deep neural networks of FINDR and the order in which the mini-batches of the training data are supplied to FINDR during training. **i**, Curved trial-averaged latent trajectories predicted by FINDR depend on the click inputs. When FINDR was fit to data in which the click inputs were randomly shuffled across trials, the trial-averaged latent trajectories remain near the origin. **j**, The dynamics are two-dimensional even in the beginning of the decision period. An early-epoch sample zone indicated by the dotted line was computed using trajectories that were truncated at time=0.33 s. The early-epoch sample zone delimits the portion of the state occupied by at least 50 of 5000 simulated single-trial trajectories. **k**, When we compute the PCs for the trajectories truncated at time=0.33 s and project the trajectories onto PC 2, the standard deviation along this direction is 20.4% of the standard deviation along PC 1. **l**, We can decode behavioral choice from logistic regression significantly better than chance (dashed line) from the projections of the truncated trajectories onto PC 2. Bold line indicates the mean, and the shading indicates 95% confidence interval. **m**, Single-trial latent trajectories extending to time=1.0 s, simulated using stimuli of different evidence strength, which is quantified by the ratio of right and left inputs.

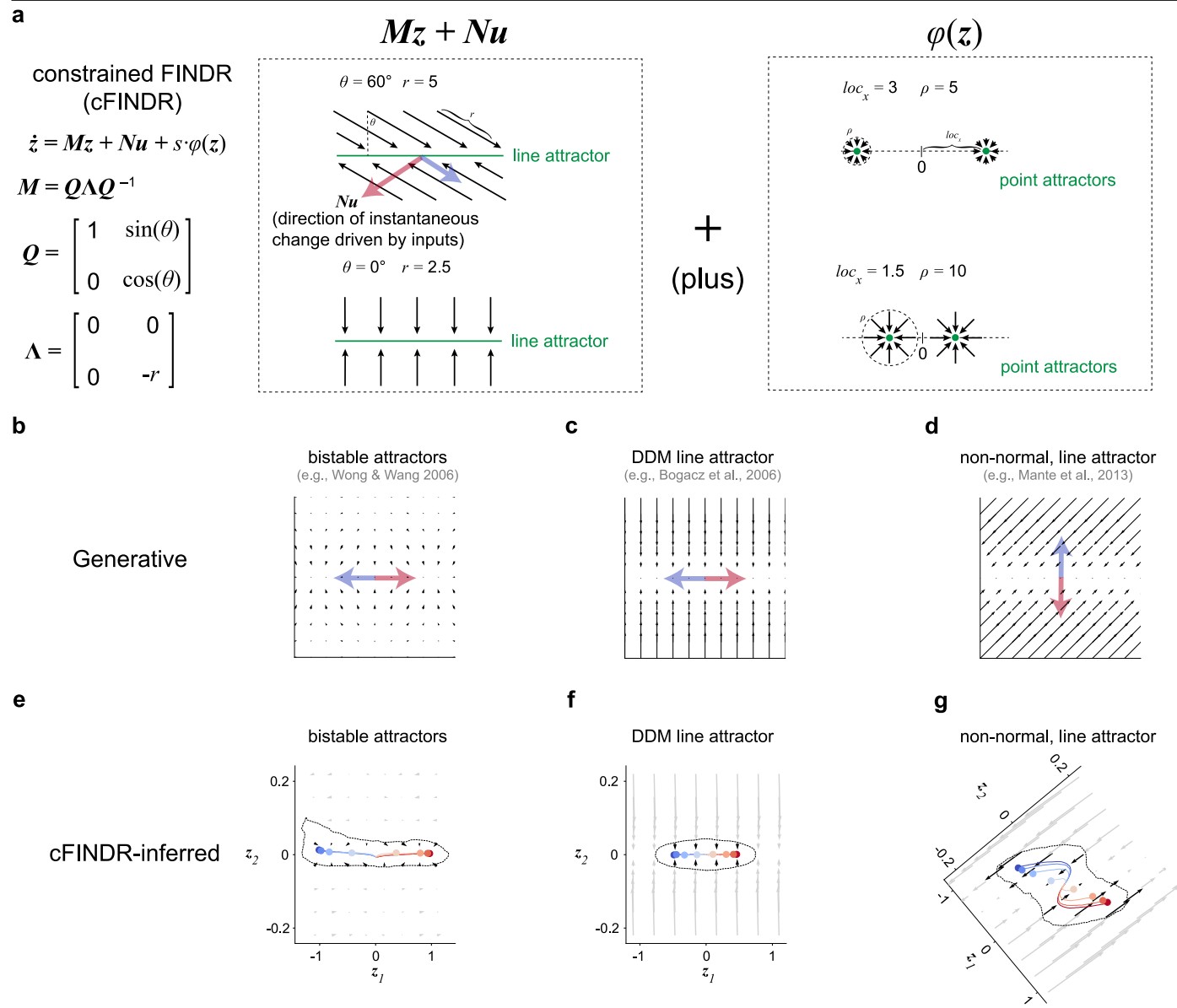

**Extended Data Fig. 5 | The data are better captured by FINDR than by a variant of FINDR constrained to parametrize the dynamics described by previously proposed hypotheses. a**, The constrained FINDR (cFINDR) model replaces the neural networks parametrizing $F$ in FINDR with a linear combination of affine dynamics, specified by $M$ and $N$, and bistable attractor dynamics specified by $\varphi$. The dynamics are furthermore constrained to be two-dimensional. **b**-**g**, cFINDR model can generate and infer dynamics described by previous hypotheses. **b**, Example bistable attractor dynamics generated from cFINDR. **c**, Example DDM line attractor dynamics generated from cFINDR. **d**, Example non-normal dynamics with a line attractor generated from cFINDR. **e**, cFINDR-inferred dynamics from a synthetic dataset generated using the bistable attractor dynamics in **b**. **f**, cFINDR-inferred dynamics from a synthetic dataset generated using the DDM line attractor dynamics generated in **c**. **g**, cFINDR-inferred dynamics from a synthetic generated using the non-normal dynamics with a line attractor in **d**.

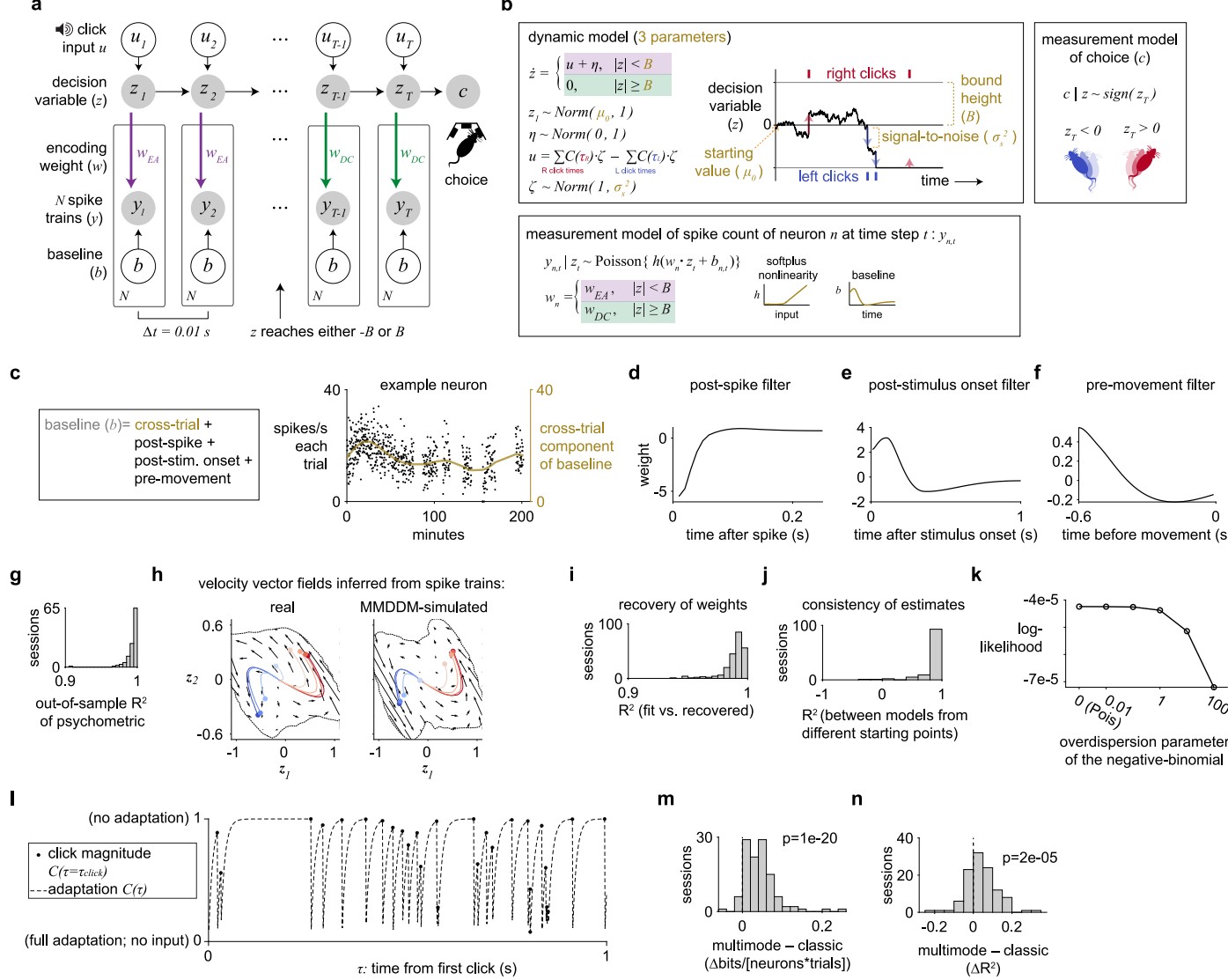

**Extended Data Fig. 6** | See next page for caption.

**Extended Data Fig. 6 | Multi-mode drift-diffusion model (MMDDM).**
**a**, Directed graph of the MMDDM for a trial with $T$ time steps and $N$ simultaneously recorded neurons. At each time step, the decision variable $z$ depends on external click input ($u$) and its value in the previous time step. The spike train depends on $z$ and also a time-varying baseline input. The behavioral choice ($c$) is the sign of the decision variable at the last time step. In this example trial, $z$ reaches the bound, and the encoding weight of $z$ of each neuron changes from $w_{EA}$ to $w_{DC}$. **b**, The MMDDM is an instance of a state-space model, which consists of a dynamic model governing the probability distributions of the latent states (here, scalar decision variable $z$) and measurement models specifying the conditional distributions of the emission (here, spike counts $y$ and the rat's choice $c$) given the value of the latent states. In the dynamic model, $z$'s time derivative ($\dot{z}$) is a piecewise linear function. When the absolute value of $z$ is less than the bound height $B$, the velocity depends on external click input ($u$) and i.i.d. Gaussian noise ($\eta$). When $z$ reaches either $-B$ or $B$, the time derivative is zero. The input of each click emitted at time $\tau$ on $z$ is scaled by the depressive adaptation from previous clicks, parametrized by $C(\tau)$, and it is corrupted by i.i.d. multiplicative Gaussian noise $\zeta$ with variance $\sigma_s^2$. The parameter $\sigma_s^2$ is one of the three parameters learned during fitting and represents the signal-to-noise of the system. The behavioral choice ($c$) is the sign of the decision variable at the last time step. The mapping from $z$ to spike train response ($y$) passes through the softplus nonlinearity $h$ and depends on baseline $b$ and encoding weight $w$. The encoding weight is either $w_{EA}$ and $w_{DC}$ depending on $z$. The three parameters that are fit in MMDDM consist of the bound height $B$, the mean $\mu_0$ of starting distribution, and the signal-to-noise of each momentary input. **c**, The baseline input consists of a cross-trial component, parametrized by smooth temporal basis functions, as shown for an example neuron. **d**, The spike history filter of the same neuron. **e**, The post-stimulus filter of the neuron. This filter does not depend on the content of the click train and only depends on the timing of the first click, which is always a simultaneous left and right click. **f**, The kernel of the same neuron to account for movement anticipation. The kernel does not depend on the actual choice of the animal. **g**, The psychometric function is well captured across sessions. **h**, The vector field inferred from real spike trains is confirmed to be similar to that inferred from MMDDM-simulated spike trains for the session "T176_2018_05_03". **i**, After fitting the model to each recording session, the learned parameters are used to simulate a data set, using the same number of trials and the same auditory click trains. The simulations are used to fit a new model, the recovery model, starting from randomized parameter values. The encoding weights of the accumulated evidence of the recovery model are compared against the weights used for the simulation (which were learned by fitting to the data) using the coefficient-of-determination metric. **j**, Consistency in the encoding weights between the training models during five-fold cross-validation. For each session, a coefficient-of-determination was computed for each pair of training models (10 pairs), and the median is included in the histogram. **k**, Whereas the Poisson distribution requires the mean to be the same as the variance, the negative binomial distribution is a count response model that allows the variance to be larger than the mean $\mu$, with an additional parameter $\alpha$, the overdispersion parameter, that specifies the variance to be equal to $\mu + \alpha\mu^2$. When the overdispersion parameter is zero, the distribution is equivalent to a Poisson. Fitting the data to varying values of the overdispersion parameter shows that log-likelihood is maximized with a Poisson distribution for the conditional spike count response. Similarly, when the overdispersion parameter was learned from the data, the best-fit values were all close to zero. **l**, The magnitude of the input after sensory adaptation of each click in a simulated Poisson auditory click train. Based on previous findings[24], the adaptation strength ($\varphi$) is fixed to 0.001, and the post-adaptation recovery rate ($k$) to 100. The generative click rate is 40 Hz, as in the behavioral task. **m**, Sensory adaptation is not critical to the improvement in fit by the MMDDM compared to the single mode DDM. Even without modeling sensory adaptation–by setting $\varphi = 1$ and $k = 0$, such that every click has the same input magnitude–the out-of-sample log-likelihood is reliably improved by the MMDDM compared to the single mode DDM. **n**, The out-of-sample goodness-of-fit of the PSTH's is also reliably improved even in the absence of sensory adaptation. **m-n**, P-values were computed using two-sided sign tests.

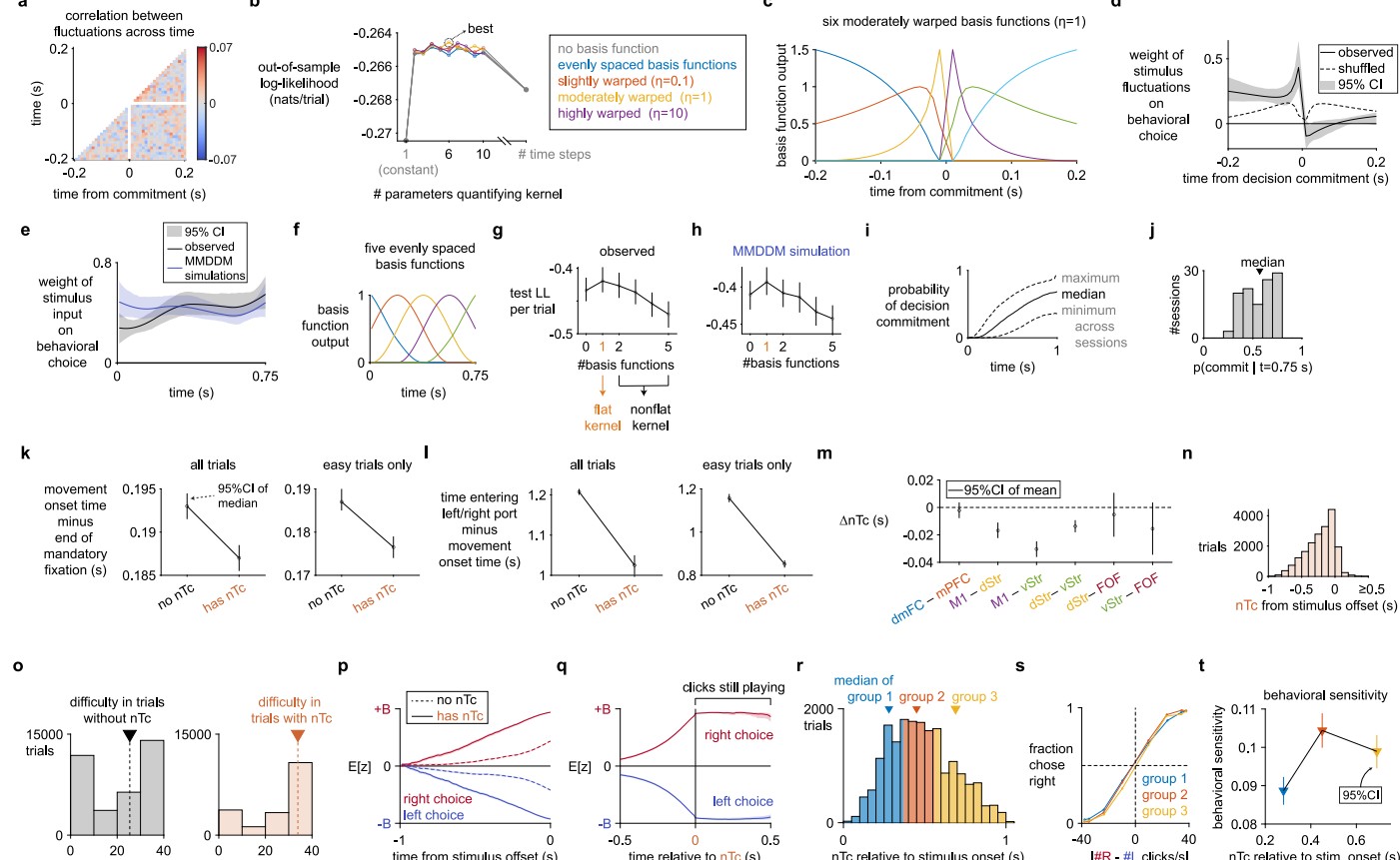

**Extended Data Fig. 7** | See next page for caption.

**Extended Data Fig. 7 | nTc and psychophysical kernels. a**, For the inferred weights of the stimulus fluctuations to be interpretable, the click input fluctuations must not be strongly correlated across time steps. On each time step on each trial, the fluctuation in auditory click input was computed by counting the observed difference in right and left clicks at that time step, and then subtracting from it the expected difference given the random processes used to generate the stimulus. The input fluctuations at time step of $t = 0$ s were excluded because they are strongly correlated with the input fluctuations before decision commitment and strongly anti-correlated with input fluctuations after commitment. **b**, To determine the time resolution of the kernel that best captures the weight of the input fluctuations, 10-fold cross-validation was performed to compare kernels quantified by different numbers of parameters and types of basis functions. The kernel with the lowest temporal resolution is a constant, represented by a single parameter, implying that fluctuations across time have the same weight. At the highest time resolution, the kernel can be parametrized by a separate weight for each time step. At intermediate time resolution, the kernel is parametrized by basis functions that span the temporal window. The basis functions can be evenly spaced across the temporal window, or stretched such that time near $t = 0$ s is represented with higher resolution and time far from $t = 0$ s with lower resolution. The most likely model had six moderately stretched ($\eta = 1$) basis functions. **c**, The optimal model's set of six moderately stretched ($\eta = 1$) basis functions. **d**, Even when using basis functions, the psychophysical kernel is consistent with the core prediction of MMDDM: The psychophysical weight of the stimulus fluctuations on the behavioral choice ceases after the time of decision commitment. Note that no basis function was used in the analysis in Fig. 5d. **e**, In contrast to the commitment-aligned kernel, the kernel aligned to the onset of the auditory click trains is smooth. Mean stimulus onset-aligned psychophysical kernel across sessions, estimated using a model with five temporal basis functions. For each session, 10-fold cross-validation was performed on fitting the kernel model to the data, and ten estimated kernels were averaged. Then, the kernels were averaged across sessions. **f**, The onset-aligned psychophysical kernel is parametrized by five evenly spaced radial basis functions. **g**, Cross-validated model comparison shows that a temporally flat psychophysical kernel is most likely given the observed data. **h**, Similarly, given the simulated choices generated by the MMDDM, the out-of-sample log-likelihood is maximized by assuming a flat kernel. **g-h**, N = 115 sessions. **i**, The approximately flat psychophysical kernel inferred from MMDDM-simulated choices is consistent with the MMDDM's prediction of the probability of decision commitment given the stimulus: throughout the trial, the probability of decision commitment is relatively low, and at no point in the trial is decision commitment an absolute certainty. **j**, At $t = 0.75$ s, the window used to compute the psychophysical kernel, the median probability of decision commitment across sessions is 0.57. **k**, A small but statistically significant effect of whether decision commitment was reached on the "movement onset time", i.e., the time when the rat withdraws its nose from the fixation port minus the earliest time when the rat is allowed to do so. The effect is not simply due to trial difficulty because it remains when we consider only easy trials (right: left click rate either greater than 38:1 or less than 1:38). **k-m**, N = 35962 trials (without nTc), 19095 (with nTc), 10261 (without nTc among easy trials), 7962 (with nTc among easy trials). **l**, Similar effect of whether commitment was reached on the rat's "movement execution time", i.e., the time when the rat reaches either the left or right port minus the time when it withdrew its nose from the fixation port. **m**, Relative timing of decision commitments between pairs of simultaneously recorded brain regions. For each pair of regions, the comparison was made on only the trials on which the threshold for commitment was crossed for both regions. N = 3936 trials (dmFC vs. mPFC), 7024 (M1 vs. dStr), 6251 (M1 vs. vStr), 8463 (dStr vs. vStr), 529 (dStr vs. FOF), 487 (vStr vs. FOF). **n**, Inferred times of commitment, relative to stimulus offset. **m-n**, P-values were computed using two-sided sign tests. **o**, As expected from the model, nTc's occur more often in easier trials, i.e., trials with larger generative (experimentally controlled) difference between the left and right click rate. **p**, As expected from the model, the mean value of the latent variable (the expectation under the posterior probability given the spikes and choice) reaches values of larger magnitude on trials on which nTc could be inferred compared to trials on which an nTc could not be inferred. Shading indicates 95% bootstrapped confidence intervals across sessions. **q**, Consistent with the model, even when considering only the period while the clicks were still playing, the mean of the latent variable abruptly plateaus after the nTc. **r**, The trials on which nTc could be estimated were separated into three groups using the terciles of the distribution of nTc relative to stimulus onset. **s**, Psychometric function of each group, showing the fraction of a right choice against the generative (i.e., experimentally specified) difference between the right and left click rates. **t**, Behavioral sensitivity is higher for trials with longer nTc. A logistic model with two terms (bias and slope) was fit to regress the choice on each trial against the generative difference in click rate. Data are presented as the best-fit slope parameters and their 95% confidence intervals, computed by bootstrapping across trials. N = 6120 trials (first tercile), 6545 (second tercile), 6336 (last tercile).

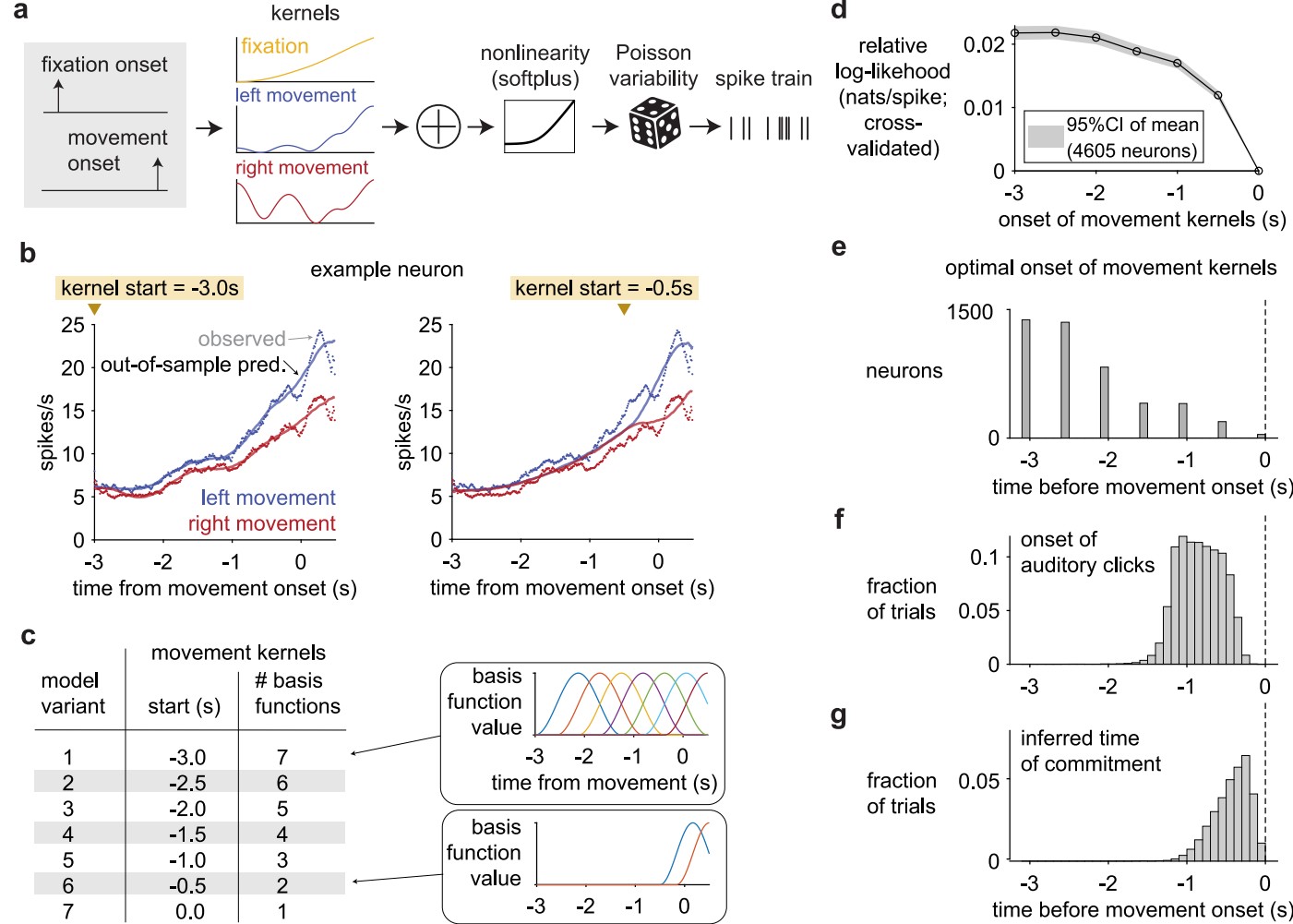

**Extended Data Fig. 8 | The distribution of commitment times inferred from MMDDM does not match the distribution of start time of peri-movement kernels. a**, Separately for each choice-selective neuron (N = 4605), peri-movement kernels are estimated using Poisson generalized linear models (GLM)[40,58]. The inputs (i.e., regressors) to the model depend on two events that occur on each trial: onset of fixation (i.e., when the rat inserts its nose into the center port), and the time when the rat leaves the center port and begins to move toward the side port. An impulse (i.e., delta function) at the time of each event is convolved with a linear filter, or kernel, to parametrize the time-varying input related to that event. At each time step, the sum of the inputs is fed through a rectifying nonlinearity (softplus) to specify the neuron's Poisson firing rate at that time. Three kernels, related to fixation, leftward movement, and rightward movement, are learned by maximizing the marginal likelihood[40]. **b**, Example neuron. Two GLM variants were fitted to the same neuron, and for each GLM variant, the observed peri-event time histogram (PETH) is overlaid the cross-validated, model-predicted PETH. The choice-dependence of the PETH of this neuron is well captured by the model variant whose peri-movement kernels start −3.0 s before and 0.5 s after movement onset (left), but less well captured by another variant whose peri-movement kernels time base are limited to −0.5 to 0.5 s (right). **c**, To identify the optimal start of the movement kernel for each neuron, cross-validated (5-fold) model comparison was performed on seven model variants that vary in the start time of the movement kernels and the number of radial basis functions used to parametrize the kernels. The end time of the movement kernel (0.5 s), and the parametrization of the fixation-related kernel (−1.5 s to 2.0 s and 4 basis functions) are identical for all variants. **d**, The out-of-sample log-likelihood is highest for the model variant whose peri-movement kernels start at −3.0 s. **e**, For each neuron, the GLM variant with the highest out-of-sample log-likelihood determines the optimal start of the peri-movement kernels. The mode of the distribution is at −3.0 s. **f**, The start of peri-movement kernels for most neurons precede the time of the first click. **g**, The start of peri-movement kernels for most neurons precede the earliest commitment time inferred from MMDDM.

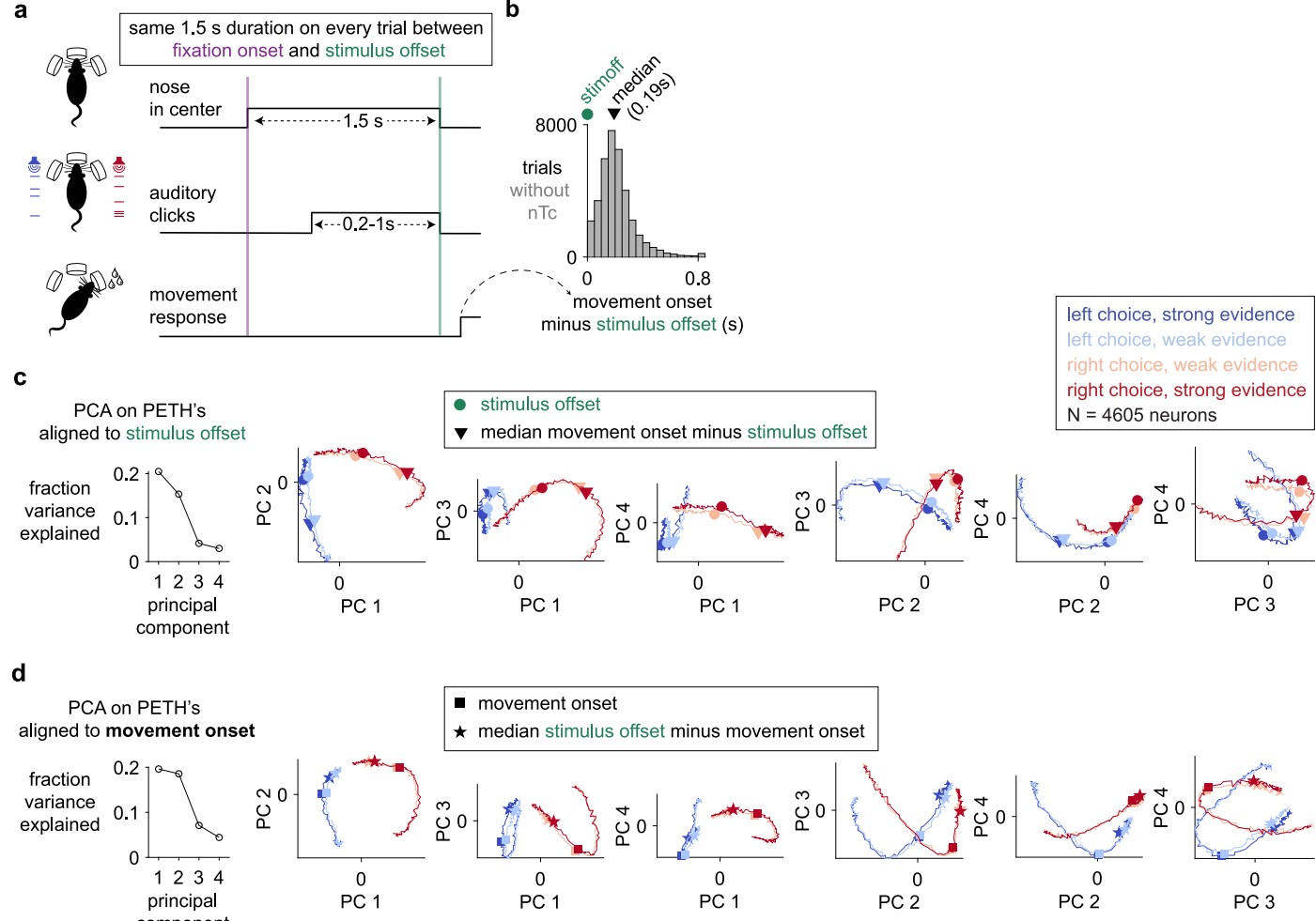

**Extended Data Fig. 9 | Changes in neural responses after stimulus offset are more closely aligned to movement onset than stimulus offset. a**, Relative timing of task events. The offset of the auditory click train stimulus always occurred at the end of the 1.5 s minimum fixation period on every trial. **b**, The median time of movement onset relative to stimulus offset across trials without a neurally inferred time of commitment (nTc) is 0.192 s. The rightmost bin contains trials for which the movement onset is 0.8 s or more after stimulus offset. **c**, Principal component analysis (PCA) was performed on peri-event time histograms (PETH's) aligned to stimulus offset (circles) and averaged across trials without a neurally inferred time of commitment (nTc). Spikes were counted in 10 ms bins, and the PETH was not additionally filtered. Spikes after the animal moved away from the fixation port (i.e., movement onset) were included. For each neuron and each trial condition, the PETH is a 100-element vector. Concatenating across 4605 choice-selective neurons and 4 trial conditions gave a 4605-by-400 matrix. The mean of each row (i.e., the average response of each neuron) was subtracted from the matrix, and PCA was performed on the resulting matrix. Triangles indicate the median time of movement onset. Projections are scaled by the standard deviation explained by each PC. **d**, PCA performed PETH's aligned to movement onset offset and averaged across trials without nTc.

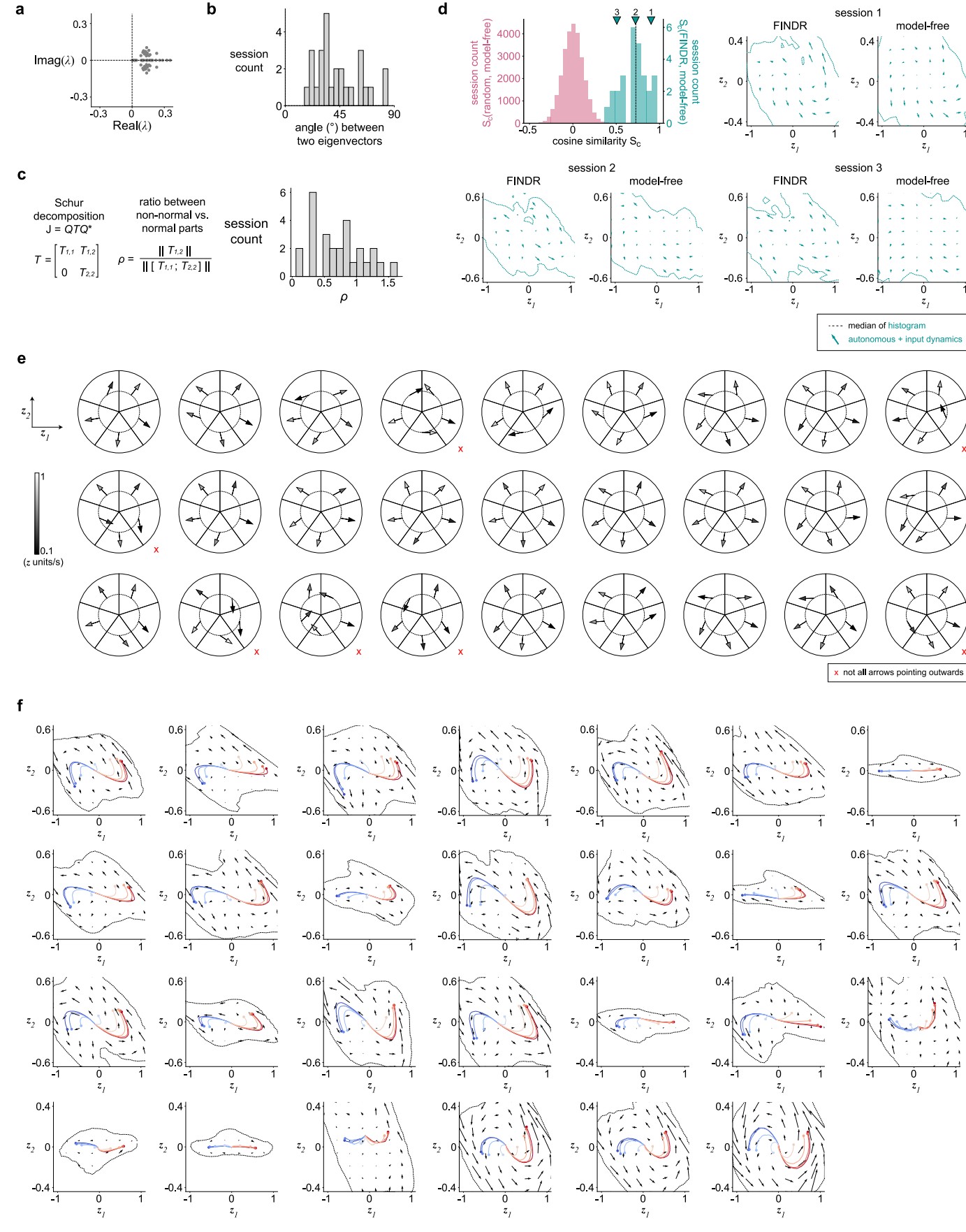

**Extended Data Fig. 10** | See next page for caption.

**Extended Data Fig. 10 | Further analyses and validation of the dynamics discovered by FINDR. a**, When we computed the eigenvalues of the numerical Jacobian J obtained from the detected slow point around the origin, the real components of both eigenvalues were greater than zero for all sessions ($n = 27$), indicating that the origin is not a stable point. Units of $\lambda$ are sec$^{-1}$. **b**, To quantify how non-normal the dynamics are around the origin, we computed the angle between the two eigenvectors of J. 90° indicates that the dynamics are normal, and angle less than 90° indicates that the dynamics are non-normal. **c**, We further evaluated the non-normality of the discovered dynamics around the origin by taking the Schur decomposition $J = QTQ^*$ and computing the ratio between the non-normal part and the normal part of the dynamics, $\rho = ||T_{1,2}||/||[T_{1,1}; T_{2,2}]||$. $\rho > 0$ indicates that the dynamics are non-normal, with higher values of $\rho$ indicating stronger non-normality. **d**, Here we estimated the low-dimensional vector field for each session using a method that does not specify a dynamical model ("model-free" approach). We compared the vector fields estimated using this approach to the FINDR-inferred vector fields. To obtain the model-free vector field, we first estimated single-trial firing rates of individual neurons by binning the spike trains in $\Delta t = 10$ ms bins and convolving the spike trains with a Gaussian of $\sigma = 100$ ms. Then, we projected the estimated single-trial population firing rate trajectories onto the subspace spanned by the FINDR latent axes. This allows direct comparisons between vector fields. For each evaluation point $(i, j)$ on an 8-by-8 grid of the latent state space $\mathbf{z}$, we estimated the velocity arrow by taking the average of $\dot{\mathbf{z}} \approx (\mathbf{z}_t - \mathbf{z}_{t-\Delta t})/\Delta t$ for all $t$ across all trajectories that fall inside the cell corresponding to the point $(i, j)$. To compare vector fields, we measured $S_c$, the mean of the cosine similarities

between the vector arrows of the model-free approach and the vector arrows from FINDR inside the sample zone. The median of the $S_c$'s across all sessions was 0.73. Three example sessions from across the distribution are shown, with session 2 around the median $S_c$ of the histogram. For both FINDR and the model-free approach, the colored trajectories were obtained by trial-averaging based on the evidence strength. To compare between a random vector field and the model-free vector field, for each session, we generated 1,000 random vector fields by randomizing the direction of each arrow in the 8-by-8 grid. **e**, We assessed the dynamical stability around the origin using a model-free approach similar to **d**. We estimated the autonomous velocity around the initial starting point (indicated as the center of the circle) of the model-free latent trajectories by taking the average of $\dot{\mathbf{z}} \approx (\mathbf{z}_t - \mathbf{z}_{t-\Delta t})/\Delta t$ for all $t$ across all trajectories that fall inside each of the 5 pie slices. Here we excluded time points where clicks affect the dynamics $(\mathbf{z}_t - \mathbf{z}_{t-\Delta t})/\Delta t$, and only considered the trajectories with #L clicks = #R clicks during the epoch when they are in the pie slice, when computing the estimate of the autonomous dynamics arrow. When computing the average of $(\mathbf{z}_t - \mathbf{z}_{t-\Delta t})/\Delta t$ for one of the pie slices, we required $\mathbf{z}_{t-1}$ to be inside the pie slice. The circles have a radius of 0.2 (in the units of $\mathbf{z}$). We found that all five arrows were pointing outwards ($p < 0.5^5 = 0.03125$) for 20 out of 27 sessions, consistent overall with the stability analysis in **a**. **f**, FINDR-inferred vector fields for all recording sessions ($n = 27$) with more than 30 neurons and 400 trials, and sessions where the animal performed with greater than 80% accuracy. These fits were used for the summary plots in Fig. 3. The vector field represents the autonomous dynamics and the trajectories are trial averages sorted by the evidence strength of each trial.

# Reporting Summary

## Statistics

For all statistical analyses, confirm that the following items are present in the figure legend, table legend, main text, or Methods section.

| n/a | Confirmed | |
|---|---|---|
| ☐ | ☒ | The exact sample size (*n*) for each experimental group/condition, given as a discrete number and unit of measurement |
| ☐ | ☒ | A statement on whether measurements were taken from distinct samples or whether the same sample was measured repeatedly |
| ☐ | ☒ | The statistical test(s) used AND whether they are one- or two-sided<br>*Only common tests should be described solely by name; describe more complex techniques in the Methods section.* |
| ☐ | ☒ | A description of all covariates tested |
| ☐ | ☒ | A description of any assumptions or corrections, such as tests of normality and adjustment for multiple comparisons |
| ☐ | ☒ | A full description of the statistical parameters including central tendency (e.g. means) or other basic estimates (e.g. regression coefficient) AND variation (e.g. standard deviation) or associated estimates of uncertainty (e.g. confidence intervals) |
| ☐ | ☒ | For null hypothesis testing, the test statistic (e.g. *F*, *t*, *r*) with confidence intervals, effect sizes, degrees of freedom and *P* value noted<br>*Give P values as exact values whenever suitable.* |
| ☐ | ☒ | For Bayesian analysis, information on the choice of priors and Markov chain Monte Carlo settings |
| ☒ | ☐ | For hierarchical and complex designs, identification of the appropriate level for tests and full reporting of outcomes |
| ☒ | ☐ | Estimates of effect sizes (e.g. Cohen's *d*, Pearson's *r*), indicating how they were calculated |

*Our web collection on statistics for biologists contains articles on many of the points above.*

## Software and code

Policy information about availability of computer code

| | |
|---|---|
| Data collection | https://github.com/Brody-Lab/decision_dynamics_commitment/tree/main/code/data_collection |
| Data analysis | Data were analyzed using code in the following repositories:<br>1) https://github.com/Brody-Lab/decision_dynamics_commitment<br>2) https://github.com/Brody-Lab/findr<br>3) https://github.com/Brody-Lab/FHMDDM<br>4) https://github.com/Brody-Lab/tzl_spGLM |

For manuscripts utilizing custom algorithms or software that are central to the research but not yet described in published literature, software must be made available to editors and reviewers. We strongly encourage code deposition in a community repository (e.g. GitHub). See the Nature Portfolio guidelines for submitting code & software for further information.

## Data

Policy information about availability of data

All manuscripts must include a data availability statement. This statement should provide the following information, where applicable:

- Accession codes, unique identifiers, or web links for publicly available datasets
- A description of any restrictions on data availability
- For clinical datasets or third party data, please ensure that the statement adheres to our policy

> The experimental data that support the findings of this study are available in Dryad with the identifier https://doi.org/10.5061/dryad.sj3tx96d

## Research involving human participants, their data, or biological material

Policy information about studies with human participants or human data. See also policy information about sex, gender (identity/presentation), and sexual orientation and race, ethnicity and racism.

| | |
|---|---|
| Reporting on sex and gender | N/A |
| Reporting on race, ethnicity, or other socially relevant groupings | N/A |
| Population characteristics | N/A |
| Recruitment | N/A |
| Ethics oversight | N/A |

Note that full information on the approval of the study protocol must also be provided in the manuscript.

# Field-specific reporting

Please select the one below that is the best fit for your research. If you are not sure, read the appropriate sections before making your selection.

☒ Life sciences ☐ Behavioural & social sciences ☐ Ecological, evolutionary & environmental sciences

For a reference copy of the document with all sections, see nature.com/documents/nr-reporting-summary-flat.pdf

# Life sciences study design

All studies must disclose on these points even when the disclosure is negative.

| | |
|---|---|
| Sample size | Sample sizes were chosen based on conventions in the field of systems neuroscience, where comparable electrophysiological studies in rats typically include 3-12 rats (12 in this study) and 48-200 sessions (here 115), as in Hanks et al., 2015 (Nature); Yartsev et al., 2018 (eLife); Pagan et al., 2025 (Nature). No formal power analysis was conducted, as the goal was to discover robust effects in a biologically realistic context rather than test a specific effect size. The number of animals and sessions reflects a balance between statistical power and the feasibility of time-intensive recordings and behavioral training. |
| Data exclusions | Electrophysiological data in time periods beyond the relevant epoch of the behavior being studied were excluded |
| Replication | We repeated the electrophysiological recording in 115 separate sessions, each involving a single animal, with a total of 12 animals (some contributing to multiple sessions). Each session served as an independent replicate. |
| Randomization | All animal subjects were part of a single experimental group. |
| Blinding | Blinding was not relevant because all animal subjects form a single experimental group. |

# Reporting for specific materials, systems and methods

We require information from authors about some types of materials, experimental systems and methods used in many studies. Here, indicate whether each material, system or method listed is relevant to your study. If you are not sure if a list item applies to your research, read the appropriate section before selecting a response.

## Materials & experimental systems

| n/a | Involved in the study |
|---|---|
| ☒ | Antibodies |
| ☒ | Eukaryotic cell lines |
| ☒ | Palaeontology and archaeology |
| ☐ | ☒ Animals and other organisms |
| ☒ | Clinical data |
| ☒ | Dual use research of concern |
| ☒ | Plants |

## Methods

| n/a | Involved in the study |
|---|---|
| ☒ | ChIP-seq |
| ☒ | Flow cytometry |
| ☒ | MRI-based neuroimaging |

# Animals and other research organisms

Policy information about [studies involving animals](); [ARRIVE guidelines]() recommended for reporting animal research, and [Sex and Gender in Research]()

| | |
|---|---|
| Laboratory animals | Rattus norvegicus, Long-Evans, 6 months to 24 months old |
| Wild animals | No wild animal was used in the study |
| Reporting on sex | Only male rats were used for this study because their larger body is necessary for the implants used to collect large-scale electrophysiological data. Similar previous studies using rats in systems neuroscience focused on using only male rats. |
| Field-collected samples | No field-collected samples were used in the study |
| Ethics oversight | The animal procedures described in this study were approved by the Princeton University Institutional Animal Care and Use Committee and were carried out in accordance with National Institutes of Health standards. |

Note that full information on the approval of the study protocol must also be provided in the manuscript.

# Plants

| | |
|---|---|
| Seed stocks | Report on the source of all seed stocks or other plant material used. If applicable, state the seed stock centre and catalogue number. If plant specimens were collected from the field, describe the collection location, date and sampling procedures. |
| Novel plant genotypes | Describe the methods by which all novel plant genotypes were produced. This includes those generated by transgenic approaches, gene editing, chemical/radiation-based mutagenesis and hybridization. For transgenic lines, describe the transformation method, the number of independent lines analyzed and the generation upon which experiments were performed. For gene-edited lines, describe the editor used, the endogenous sequence targeted for editing, the targeting guide RNA sequence (if applicable) and how the editor was applied. |
| Authentication | Describe any authentication procedures for each seed stock used or novel genotype generated. Describe any experiments used to assess the effect of a mutation and, where applicable, how potential secondary effects (e.g. second site T-DNA insertions, mosiacism, off-target gene editing) were examined. |

