## [Peer Review File · Nature]

Transitions in dynamical regime and neural mode during perceptual decisions

Corresponding Author: Dr Thomas Luo

Version 0:

Reviewer comments:

Referee #1

(Remarks to the Author)

Dynamical system theory, including attractor models, provides a mathematical explanation for how neural dynamics produce computations. Luo & Kim et al. performed large-scale electrophysiology in rats performing a perceptual decision-making task (auditory evidence accumulation task). They applied a novel, unsupervised, deep learning-based method to uncover dynamical regimes underlying decision-making. Stochastic sensory input in the task allowed for dissociating internally generated attractors and input-driven dynamics. Interestingly, their data-driven approach revealed a rapid transition from the input-driven regime to the internally-generated regime within a trial, which beautifully explains 1) the quenching of the contribution of sensory evidence on decision after commitment and 2) the diverse and complex single-neuron activity patterns often observed in perceptual decision-making tasks. These findings are interesting and important, and the methods are novel and versatile. There are relatively minor issues to be addressed before publication, as listed below.

- 1) L124-127: L124 describes that dmFC provides a major anatomical input to dorsal Str, whereas the following sentence says it provides input to ventral Str. Does dmFC project to both dorsal and ventral Str, or is this a typo? EDF2c only shows retrograde labeling from dorsal Str.
- 2) In addition to validation using synthetic data (L150-154), it would be nice to have a shuffle control (e.g., shuffle sensory input across trials and run FINDR).
- 3) Fig2&EDF6: Please report the number of neurons used for the analysis. Also, how many neurons/spikes are required to infer the dynamics successfully? Does it heavily rely on subsets of neurons with strong selectivity?
- 4) Fig3m: how is 'shuffle' computed?
- 5) EDF12e is beautiful and consistent with Fig5b. It should be in the main figure.
- 6) Is the predicted timing of decision commitment the same across brain areas? Or are there time lags across areas? Can you analyze it based on simultaneous recordings?
- 7) Does the timing of decision commitment affect behavior? For example, does it explain reaction time? Does the predicted timing of decision commitment reflect the transition from motor planning to movement, i.e., do rats start moving at that time?
- 8) Why did you exclude neurons that are not choice-selective (according to Methods)? Non-choice selective neurons can encode urgency, choice commitment, etc. Does FINDR infer similar or completely different dynamics when non-choice-selective neurons are added?
- 9) Performance varies across animals, according to EDF2. Do the inferred dynamics explain these behavioral variabilities (e.g., shallow attractors for bad performers, etc.)?

Referee #2

(Remarks to the Author)

The authors of this study attempt to infer latent dynamics from neural activity during a decision making task. They claim to have found a new dynamical mechanism that better explains their data than a line attractor (Mante) or bistable attractor (Wong and Wang). They also propose a modified DDM that they argue better explains their data than the DDM. I agree with the authors that it is important to test dynamical systems mechanisms from neural data, to understand the neural substrate

behind decision making. However, I don't agree with the authors that their claims are substantiated by their analyses. After reading this manuscript, I actually have renewed faith in the Mante, Wong/Wang and Bogacz papers as simplified but robust (and timeless) models of decision making. If the authors can point out where I misunderstood I am happy to reconsider, because I think the motivation behind this study is very important for the field.

Major Concern 1: The proposed dynamical mechanism does not appear to match the authors' own vector fields

The authors have claimed to have found evidence of bistable attractor dynamics. A close look at Figure 2c,d reveals that this description does not match their own vector fields. First of all, the apparent fixed points are only slow points as the speed never gets to zero. However, even more concerning, is that the vector field *around the slow points* are not consistent with attractor dynamics. The field around [-.5,6] points *away* from the slow region in the upper right plane, and similarly for the bottom right region of the phase plane for the bottom left attractor. There are no fixed point attractors in this phase plane. (If anything, it looks like there might be a stable limit cycle but that is aside the point) If the authors don't agree, I would urge them to replot this vector field in a normalised form (since the magnitudes are shown in the speed plot anyway).

If we inspect ED Figure 6a, we see that the vector fields for each recording have completely different dynamical landscapes. Some of these (T176_2018_05_31) do appear to have attractors corresponding to the two choices, but also have several spurious fixed points that aren't explained in terms of computational mechanism. Others (X046_2019_09_23) only have one fixed point! From what I see, the authors' claims do not match their own data.

Major Concern 2: Degeneracy in the model and possibility for other mechanisms to explain the data equally well

The neural network function approximates used to estimate $F(z,u)$ is likely to be highly degenerate and have multiple solutions that give similar performance. Therefore, I don't believe that the authors have sufficiently ruled out other dynamical mechanisms. Moreover the inconsistency of the identified vector fields in ED Figure 6 makes me especially wonder if the neural network model is just too parameterized. Could the authors precise how many parameters it has? In 2a, are only the green connections plus a scalar baseline per hidden unit fitted? Can they provide evidence that other dynamical mechanisms like the line attractor cannot explain neural activity as well? In Fig 2h, the trajectories on the right look extremely similar to the line attractor shown in Mante where the trajectories move to different points along the line determined by the strength of evidence.

On this point, even within the model class proposed by the authors, it doesn't appear clear to me that their mechanism is a better fit than other mechanisms. In ED Figure 4 the authors show R^2 (is it cross-validated?) but only for one example session. They should show summary statistics across all data. In ED Figure 5 it is unclear first of all how the cross validation is working and what the bits/spike is used rather than a standard measure like variance explained, mean squared error, etc. More importantly, it looks like a 1-dimensional latent variable has the same explanatory power as the 2-dimensional system that they have chosen. But, in the 1-d case, this would then just result in a standard bistable attractor as in the Wang and Wong paper they cite.

Minor concerns:

Mechanistically, and computationally (in terms of what it means for the decoder) I don't understand the idea behind the MMDDM that the weights change after the decision commitment. Is this the only difference from a DDM? Could the authors help to give some interpretation?

I am not sure how to interpret the results considering the variability of the recorded regions across sessions. Could this explain why the vector fields give such different results?

Will the data and code be made available?

Lack of mathematical precision:

- * Throughout, "intrinsic dynamics" should be replaced by "autonomous dynamics", which is the more precise term. It should also be clarified in the text that the input-driven dynamics are determined by the input and the autonomous/intrinsic dynamics together, not just the input (e.g. in line 68-69 which is not true in general, only in your proposed model where the autonomous dynamics are zero before decision commitment).
- * In a bistable dynamical system there are two stable fixed points and a saddle, not an unstable fixed point as stated on line 158 and elsewhere.
- * Line 212 - Gaussian with diagonal covariance. Isotropic or non? This was also unspecified in the methods.
- * Methods line 114-120: Unclear what this is accomplishing, please clarify this section and include equations / better notation to explain this section.
- * Methods Section 9: Equations 8-10 are imprecise. It appears you are doing some kind of mean-field approximation or expectation of the latent dynamics over the input stochasticity. But, this can be derived exactly and written in precise formulation, rather than just weighting the equation by the conditional probabilities. Moving from Equation 7 to Equation 8 is improper, and needs better notation which will help in contrast to the picture in Figure 1 that shows the discrete and stochastic inputs.
- * Lines 153-160 which describe a novel model have no equations. The only equation for MMDDM that I see is in Figure 3c where it appears that there is actually no dynamics - it's just evidence accumulation with a stochastic transition to a decision?

(Remarks to the Author)

Luo et al. examined attractor dynamics of populations of neurons recorded from various brain regions of rats performing perceptual decision making. Using their new method called FINDR, they revealed population dynamics that differed from existing models. In the early period, there were little intrinsic dynamics and the population state was largely modulated by sensory input. In the later period, the population state was more strongly modulated by intrinsic dynamics that were not aligned with the direction of the input dynamics. Thus, there was a transition in dynamics from input-driven to intrinsic. The authors propose that the earlier state corresponds to evidence accumulation and the later state corresponds to decision commitment. They developed an extension of DDM (MMDDM) that fitted the neural data well and was consistent with the results of psychophysical reverse correlation. The authors claim that their model accounts for a variety of previously reported phenomena such as curved neural trajectories and ramping vs. stepping.

The paper performed extensive data collection and sophisticated analysis including the application of their novel analysis method and the development of a new computational model. These were executed with high quality and explained effectively in the manuscript. Revealing the system dynamics (attractors) underlying cognitive functions has been of great interest to the systems neuroscience community, and the authors' successful application of their new method is commendable. Furthermore, the authors built a new model (MMDDM) with great attention to the details of its implementation. However, after carefully reading the manuscript, I would like to raise several important questions and concerns. My main concerns are the extent of the conceptual advance provided by their conclusions, the validity of focusing on a representative dataset to draw their conclusions, and the possibility of alternative models.

Major

1.
The main conclusion of the paper is the transition from an evidence accumulation mode to a decision commitment (or action execution) mode, but there are multiple previous studies that reported distinct response dynamics for evidence accumulation and action planning in various task settings (Bennur & Gold 2011, Park et al. 2014, Okazawa et al. 2021, Charlton & Goris 2022, Yates et al. 2017). These studies have often shown that there are early response dynamics reflecting decision formation and then it is replaced by later dynamics encoding choice or action plan. The authors have cited and discussed these studies (page 12), but after reading it, I still find it difficult to judge whether the conclusions of this paper conceptually go beyond these previous findings. I understood that the present paper brought a quantitative framework, directly fitting the attractor and input dynamics to the data and providing a full model (MMDDM) to account for this phenomenology. Thus, I do not doubt the novelty of the paper. Nevertheless, I am not fully convinced that this quantitative modeling of known properties significantly updates/changes our perspective on the neural mechanisms of decision making enough to be published in Nature.

An important claim of the paper is that MMDDM provides a parsimonious explanation for a variety of experimental findings (page 12), which is probably meant to propose MMDDM as a unified model of perceptual decision making, but I am a bit reserved about this statement. First, while the authors cited Mante 2013 and Aoi 2020, MMDDM does not explain why curved trajectories could become context dependent (the lack of intrinsic dynamics in MMDDM cannot explain how the network could become context dependent). Second, the model is said to account for diversity in single neuron dynamics (e.g., Meister et al 2013) and mixed stimulus and choice selectivity, but mixed selectivity is quite common in any RNNs or other latent state models. Thus, I thought this statement does not give MMDDM a unique status.

2.
The key figures showing the attractor and input dynamics (Fig. 2c-h) are from a representative session. These figures look great, but the results from all the sessions should be shown to support this. The authors show the results from a few more sessions in Extended Fig. 6, and not surprisingly, there is some variability in the patterns across sessions, which makes it hard to read whether they indicate the same things. I am not sure if FINDR could be fitted to all the data combined, or if it could be averaged across sessions. If that is not possible, the authors can generate metrics such as the relative strength of input and intrinsic dynamics, the angle between the input and commitment dynamics, and plot their distributions across sessions. Fig. 3i-j shows the results from all sessions, but this only compares the model fit accuracy between MMDDM and DDM and does not directly support the claims made in Fig. 2c-h.

3.
According to FINDR, there is little intrinsic dynamics during decision formation along both z_1 and z_2 axes, but I was not sure whether the dynamics along z_2 axis could be convincingly inferred from the data. The population state starts from the origin in the space and quickly diverges toward z_1 axis due to input (Fig. 2h), so I thought there were not many cases where the population state goes along z_2 axis, then the FINDR-inferred dynamics along this axis are not very reliable. The authors wrote "only the well-sampled subregion of the state space is shown" (page 7) so I thought my concern did not apply, but I then realized that the same subregion was drawn for all the time points shown in Fig. 2h. If the authors draw a well-sampled subregion for each time point, is it possible that, early in the decision (0.33s), the neural state is confined to a narrower region that expands along z_1 ? If so, z_2 axis does not play a critical role in decision formation (because the neural state rarely goes along that direction)? This might require some revision of the authors' discussion regarding the role of this axis such as history effects (page 13).

4. The authors brought up multiple attractor models in Fig. 1, but as far as I could see, they did not directly test (fit) the RNN line attractor (Mante 2013). For the line attractor, Fig. 3i shows the difference in out-of-sample log-likelihood between the authors' MMDDM and DDM (instantiated as a line attractor) and the results look promising (also bistable attractors in Extended Fig. 12). Line and bistable attractors only predict a linear neural trajectory, so it makes sense that they perform poorly in explaining the observed curved trajectories. But it is not clear how poorly the RNN line attractor fits the data compared to the authors' model, since both models predict similar curved trajectories. Is it possible to perform a quantitative comparison?

5. MMDDM inferred that the commitment (bound crossing) occurred in 34.6% of the trials (Method, page 6), so the accumulated evidence did not reach a bound before stimulus offset in the other ~65% of the trials. This number looks reasonable, but I wonder how FINDR and MMDDM treat stimulus offset in these trials. Stimulus offset is a go cue in this task, so it would trigger rats to commit to a choice and initiate their action, so I expect it to become a strong external drive to influence the neural dynamics (presumably forcing the network state to make a transition to the commitment mode). What do neural trajectories look like near stimulus offset? The models do not need to account for this?

Minor

1. The authors contrasted MMDDM with "classic DDM" (Fig. 3), but this classic DDM was instantiated as a line attractor. Behavioral DDM does not specifically prescribe how it should be implemented in neural circuits, so is it more accurate to call it "line attractor" instead of "classic DDM"?

2. Fig. 3m: how many sessions/rats were used to calculate the reverse correlation? Could there be error bars in the plot or statistical tests?

3. Extended Fig. 4: Is this the result of a single session? Could the authors plot the fit quality of all the sessions?

4. The authors selected ~10% of choice selective neurons to perform all the analysis (Method, page 3). But neurons that do not have strong choice selectivity could still contribute to the population dynamics. Does this selection affect the results of their analysis?

Version 1:

Reviewer comments:

Referee #2

(Remarks to the Author)

Thanks to the authors for writing a detailed response clarifying what is consistent and novel about their inferred dynamics. I do feel more convinced about the revised results but I am still concerned about the accuracy of the inferred dynamical mechanism. I agree with the authors that whether the slow points were attractors or not was not really important to the main message of the paper. I am partly assuaged (but only partly) by the additional tests and corrections. I will cut to the chase.

Major concern 1: What, actually, is the dynamical mechanism found by FINDR?

To me it looks like non-normal nonlinear dynamics that lead to two slow points representing the different choices. If you look at what is happening in a neighborhood around the origin, it looks like the slow manifold is partially aligned with a fast mode that points in the direction of the two lobes of the slow S-shaped mode (the direction shown at time = 0.33s in 2h). The input directions are aligned with that fast mode which would give you textbook non-normal amplification. I want to stress here that the novelty is not as important, in my opinion, as the accuracy – if the answer is simply non normal amplification, that would be okay.

I appreciate the efforts taken by the authors to design refit a new model (cFINDR) to argue that the dynamics are not non-normal. It is an interesting approach. However, there are a lot of differences between cFINDR and FINDR. For example, the dynamics inferred by cFINDR appear very linear (ED Figure 7h). This is not so surprising since cFINDR constrains the latent dynamics to be almost linear (except the phi, but that is used only for the bistability). In contrast FINDR is parameterized by a neural network and can be very nonlinear. I don't see how cFINDR could find nonlinear non-normal dynamics and I wonder if FINDR fits the data better simply because it is nonlinear.

How can this be addressed? Well, one possibility is to constrain FINDR to linear normal dynamics, but I am concerned it

wouldn't fit the data well either. A much simpler approach is to check if the Jacobian at the origin is normal. You can do this by taking the Schur decomposition of the Jacobian at the origin (when $u=0$), and calculating the Frobenius norm of the upper triangular part (not including the eigenvalues on the diagonal). Maybe take this as a fraction of the Frobenius norm of the diagonal, to get a ratio of the non-normal part to the normal part? But I am not sure how to make a strong statement from this. You could compare that value to the same value of the Jacobian calculated at other parts of the space to show that the origin has strongly non-normal dynamics for input amplification?

Major concern 2: Can we trust the flow field found by FINDR?

It is notoriously difficult to infer dynamics from neural data. Latent dynamics cannot be verified in ground truth in data (not including the toy models). Work from Valerio Mante, Lea Duncker, Guillaume Hennequin, and others have shown that the wrong dynamics can be inferred (which they verify with perturbation studies and trial-to-trial variability). Furthermore, FINDR has yet to be published meaning that it has not passed peer review.

How can this be addressed? One possibility is if you have perturbation data which would kick the trajectories into different regions of the flow field to make sure those results are consistent with what you have found with FINDR. A second possibility is to look at the trial-to-trial variability to analyze if those trajectories are consistent with the flow fields at a single trial resolution. A third possibility is to apply other methods to identify low dimensional dynamics (recurrent switching linear dynamical systems from Scott Linderman, low rank RNNs from Srdjan Ostojic) to check that the inferred mechanism is robust.

Minor concerns:

I am surprised that the origin is an unstable fixed point. Did you fit to data before the clicks or only after onset of the clicks? (I can't find the methods anymore to check this - did they disappear? Am I blind?) I wonder if this is why the origin is unstable.

Did the authors remove the flow fields inferred from other animals? ED Fig 6 is replaced with validation across folds and seeds. That's fine, but I think it is very important for transparency and interpretation to show the inferred flow fields across ALL animals.

Finally I have the same minor comment regarding mathematical precision. I just quickly flipped through the supplementary information and caught a few odd statements

- Eq 17, this is not a Fokker-Planck equation. This is just an OU process, which is a type of sde. The Fokker-Planck equation is a pde that describes the evolution of the distribution of an OU process.

- Section 3.1.4, $C(dt) = \phi$, where C was defined as an ode above. I am not sure what this means, it is not standard terminology.

This was from a quick skim but I urge the authors to go through a bit more carefully. I think it is important to get these details clear so that the paper is replicable.

(Remarks on code availability)

Referee #3

(Remarks to the Author)

The authors did a great job of addressing my concerns. I understood that the change in dynamics mode is not a transition from decision to action. The results including all sessions (Fig. 2k, l) are exactly what I wanted to see. I agree that the quantitative comparison with the authors' FINDR and Mante's attractor was not easy because there is no established way to model Mante attractor, so I think the authors did the best they could do. Overall, I have no further comments.

Just one point about Fig. 3n. The green "predicted" psychophysical kernel is added in the revised manuscript, but is it a quantitative prediction of MMDDM? I am asking because I am not sure if MMDDM really predicts a flat kernel before nT_c . A decision bound can have an additional effect (since choice is directly determined by the bound crossing, there would be stronger correlation between input pulse and choice at the moment of bound crossing), so it would not be flat. If it is not a quantitative prediction, the authors may decide to drop it or take care of it in some way. But I do not need to check the revision again.

(Remarks on code availability)

Version 2:

Reviewer comments:

Referee #2

(Remarks to the Author)

The reviewers did an outstanding job at responding to my concerns. I am now really excited about the findings of this manuscript and am looking forward to seeing it in print.

(I only have one small comment - when I suggested comparison with other methods, I meant to check that the inferred vector fields are similar. This isn't shown in the PSTH R squared analysis. I think it would be interesting and useful for readers who want to perform a similar analysis on their own data. However I leave this up to the authors. Their model free analysis was sufficient to respond to my concern about robustness of the findings.)

(Remarks on code availability)

We are very grateful to the reviewers for their thoughtful comments, which have helped us greatly improve the manuscript.

Below, reviewer comments will be in blue, and our responses in black. For easy reference, we have also included an Appendix with copies of many figure panels and selected excerpts from the Methods within this responses document; the hyperlinks (e.g., Extended Data Figure 6i) point to these panels within this document, so reviewers can simply click and find the corresponding figure panel. No access to the Internet is required.

TABLE OF CONTENTS:

Luo*, Kim*, response to reviewers.....	2
Summary of major changes.....	2
Referee #1 (Remarks to the Author):.....	3
Minor concerns:.....	3
Referee #2 (Remarks to the Author):.....	6
Major Concern 1:.....	6
Major Concern 2:.....	8
Minor concerns:.....	10
Referee #3:.....	13
Major concern 1:.....	14
Major concern 2:.....	17
Major concern 3:.....	17
Major concern 4:.....	18
Major concern 5:.....	20
Minor concerns:.....	20
Appendix.....	22
Figure 2i-l.....	22
Figure 3l,m,n.....	23
Extended Data Figure 4.....	24
Extended Data Figure 5a-e.....	25
Extended Data Figure 6a-c.....	26
Extended Data Figure 6d.....	27
Extended Data Figure 6f.....	27
Extended Data Figure 6g.....	28
Extended Data Figure 6h.....	28
Extended Data Figure 6i.....	29
Extended Data Figure 6j-l.....	29
Extended Data Figure 6m.....	30
Extended Data Figure 6n.....	30
Extended Data Figure 7.....	31
Extended Data Figure 10b-d.....	32

	2
Extended Data Figure 10k-l.....	32
Extended Data Figure 10m.....	33
Extended Data Figure 10n.....	33
Extended Data Figure 14.....	34
Extended Data Figure 15.....	36
Methods Section 2.....	37
Methods Section 3: FINDR.....	38
Methods Section 3.1-2.....	39
Methods Section 3.3.....	39
Methods Section 3.6: cFINDR.....	40
Methods Section 4: MMDDM.....	41
Methods Section 5.4: Baseline.....	43

Luo*, Kim*, response to reviewers

Summary of major changes.

We are very grateful for all the reviewers' comments, which have been extremely helpful towards improving the manuscript. Major changes in this submission include

- We fit FINDR to many more sessions. The method requires a sufficient number of trials and neurons, so we fit it to all sessions that had 400 or more trials and 30 or more choice-selective neurons, and overall behavioral performance of 80% correct or greater. This gave 27 sessions across multiple rat subjects. Focusing on these sessions, with sufficient trials, neurons, and good behavior led to greater consistency across subjects and sessions (see next bullet point).
- We developed new measures to quantify key aspects in the FINDR-inferred flow fields. These new measures now allowed us to demonstrate that the key results from FINDR are consistent across subjects and sessions (Fig. 2i,j,k,l) and consistent across cross-validation folds (Extended Data Fig. 6a,b,c).
- Our original submission had a description of the estimation of latent space dimensionality that proved rather confusing. We have replaced it with a simpler approach, and now show that, consistently across all 27 sessions, 2 latent dimensions are better than 1 dimension at describing out-of-sample data not used in the fitting (Extended Data Fig. 5c); and 3 dimensions are often better, but explain a negligible further amount of variance. The revised manuscript therefore focuses on analyses using 2 dimensions.
- We developed a head-to-head comparison between FINDR-inferred dynamics versus dynamics that correspond to the three pre-existing hypotheses in **Fig. 1** (namely, bistable attractors, DDM line attractor, and line attractor with non-normal dynamics). The results provide strong support for the FINDR-inferred dynamics being a substantially better fit to the data than any of the three pre-existing hypotheses (Extended Data Fig. 7).

- Some of our most novel findings relate to “nTc,” the putative neurally-inferred time of decision commitment signal. We have further emphasized nTc in the abstract, text, and the discussion, and we added a new paragraph to the Discussion describing how decision commitment is distinct from action planning, both conceptually and at the neural level (Extended Data Fig. 14).
- We include new data panels demonstrating that our neurally-inferred time of decision commitment (“nTc”) has a timing that is highly variable relative to motor response onset (Fig. 3l) or to stimulus onset (Fig. 3m), or to stimulus offset (Extended Data Fig. 10n). It is thus an internal event, largely defined by coordination across neurons, not by its timing with respect to external features.

Point-by-point responses to each of the referees’ comments follow.

Referee #1 (Remarks to the Author):

Dynamical system theory, including attractor models, provides a mathematical explanation for how neural dynamics produce computations. Luo & Kim et al. performed large-scale electrophysiology in rats performing a perceptual decision-making task (auditory evidence accumulation task). They applied a novel, unsupervised, deep learning-based method to uncover dynamical regimes underlying decision-making. Stochastic sensory input in the task allowed for dissociating internally generated attractors and input-driven dynamics. Interestingly, their data-driven approach revealed a rapid transition from the input-driven regime to the internally-generated regime within a trial, which beautifully explains 1) the quenching of the contribution of sensory evidence on decision after commitment and 2) the diverse and complex single-neuron activity patterns often observed in perceptual decision-making tasks. These findings are interesting and important, and the methods are novel and versatile. There are relatively minor issues to be addressed before publication, as listed below.

We are grateful to the reviewer for their encouraging words.

Minor concerns:

1) L124-127: L124 describes that dmFC provides a major anatomical input to dorsal Str, whereas the following sentence says it provides input to ventral Str. Does dmFC project to both dorsal and ventral Str, or is this a typo? EDF2c only shows retrograde labeling from dorsal Str.

Thank you for catching this. We have modified the sentence to

The dorsomedial frontal cortex (dmFC) is a major anatomical input to dStr⁴², as confirmed by our retrograde tracing (Extended Data Fig. 2c) and is also causally necessary for the task (Extended Data Fig. 1d). The dmFC is interconnected with the medial prefrontal cortex (mPFC), and less densely, the FOF, the primary motor cortex (M1)⁴³, and the anterior ventral striatum (vStr)⁴².

2) In addition to validation using synthetic data (L150-154), it would be nice to have a shuffle control (e.g., shuffle sensory input across trials and run FINDR).

Thank you for suggesting this analysis. We have performed a shuffle control in which sensory inputs were shuffled across trials and found that the latent trial-averaged trajectories are highly different between the FINDR fits to shuffled and actual data. We have incorporated this finding in Extended Data Fig. 6i.

3) Fig2&EDF6: Please report the number of neurons used for the analysis. Also, how many neurons/spikes are required to infer the dynamics successfully? Does it heavily rely on subsets of neurons with strong selectivity?

Thank you for this question. We now report the number of neurons used for the analyses in **Fig. 2c-h** and Extended Data Fig. 6. We are able to infer the dynamics successfully for sessions with >30 neurons, >400 trials, and >80% correct in the behavioral performance. The results do not depend on using only the subset of neurons with strong selectivity, because similar vector fields were inferred using either only the subset of choice-selective neurons or both choice-selective and non-selective neurons, as shown in Extended Data Fig. 6g.

4) Fig3m: how is 'shuffle' computed?

Thank you for suggesting this clarification. We have clarified our shuffling procedure in Methods Section 5.7:

In Figure 3n and Extended Data Fig. 10, a shuffling procedure was used to randomly permute the inferred time of commitment across trials, without changing the behavioral choice and the times of the auditory clicks on each trial. Within this randomly permuted sample, we selected trials for which the auditory stimuli were playing at least 0.2s before and at least 0.2s after the inferred time of commitment to compute the psychophysical kernel in the “shuffled” condition.

5) EDF12e is beautiful and consistent with Fig5b. It should be in the main figure.

Thank you. We have now incorporated EDF12e as **Fig. 5f**.

6) Is the predicted timing of decision commitment the same across brain areas? Or are there time lags across areas? Can you analyze it based on simultaneous recordings?

Thank you for suggesting this analysis. We inferred the neurally inferred times of commitment (nTc) separately using neurons from different brain regions and compared the nTc 's between brain regions that were simultaneously recorded. We observed a small but statistically reliable lead by primary motor cortex (M1) over either dorsal striatum (dStr) and ventral striatum (vStr), and also a lead by dStr over vStr. These leads are on the order of tens of milliseconds. We did not observe a difference when comparing the differences in commitment times between dorsomedial frontal cortex (dmFC), medial prefrontal cortex (mPFC), or the frontal orienting fields (FOF). We have included this result in Extended Data Fig. 10m.

7) Does the timing of decision commitment affect behavior? For example, does it explain reaction time?

Thank you for suggesting these analyses. To clarify, the task performed by the rats is not a reaction time paradigm: the rat must maintain its nose in the fixation port for 1.5 seconds on each trial. The auditory clicks always terminate at the end of the 1.5 s fixation period and can start 0.5s-1.3s after the onset of fixation. We have added in the legend of **Fig. 1a** to clarify:

The earliest time when a rat can respond is fixed relative to the moment of inserting its nose in the center port (i.e., not a reaction time paradigm).

Does the predicted timing of decision commitment reflect the transition from motor planning to movement, i.e., do rats start moving at that time?

This is a very important point that we did not sufficiently emphasize in the original submission. No, the neurally-inferred time of decision commitment, which we now abbreviate as “nTc,” does not mark movement onset. It is instead an internal—covert—event. nTc is highly variable relative to stimulus onset (new Fig. 3l), relative to stimulus offset (new Extended Data Fig. 10n) and relative to movement onset (new Fig. 3m).

Following the reviewer’s suggestion, we further analyzed whether there was a relationship between nTc and behavior. We found that on each trial, whether or not a time of commitment could be inferred (i.e., whether or not nTc occurred) has a small but statistically significant effect on that trial’s “movement onset time” (time when the rat withdraws its nose from the fixation port minus the earliest time when the rat is allowed to do so; about 10 ms faster when nTc occurs) and also on the “movement execution time” (time when the rat reaches either the left or right port minus the time when it withdrew its nose from the fixation port; >100 ms faster when nTc occurs). These effects are not simply due to trial difficulty because they can be seen even when we consider only easy trials (right : left click rate either greater than 38:1 or less than 1:38). We have now included panels showing these data in Extended Data Fig. 10k-l.

8) Why did you exclude neurons that are not choice-selective (according to Methods)? Non-choice selective neurons can encode urgency, choice commitment, etc. Does FINDR infer similar or completely different dynamics when non-choice-selective neurons are added?

Thank you for this helpful question. Because neurons multiplex many signals, some of which may not be related to the animal’s decision, we focused on the choice-selective neurons to maximize the chance that the inferred vector field is relevant to the animal’s decision. Indeed, non-choice selective neurons can encode urgency, choice commitment, etc, and would be important to examine in detail in future work. We compared the vector field inferred using either only choice-selective neurons or to all neurons that have a firing rate > 2 spikes/s (a criterion that all “choice-selective” neurons met) for the representative session shown in **Fig. 2**. We found that both the dynamics and the trial-averaged trajectories to be highly similar whether we used only choice-selective neurons or both choice-selective and non-choice selective neurons. This is shown in Extended Data Fig. 6g.

9) Performance varies across animals, according to EDF2. Do the inferred dynamics explain these behavioral variabilities (e.g., shallow attractors for bad performers, etc.)?

We found the relative strength of the autonomous vs. input dynamics (**Fig. 2i-k**) before the change in the neural mode in each session to correlate with the animal’s behavioral sensitivity in that session—that is, stronger input dynamics in the neural activity corresponds to slightly greater sensitivity to inputs in the behavior. We have incorporated this finding in Extended Data Fig. 6d.

Referee #2 (Remarks to the Author):

The authors of this study attempt to infer latent dynamics from neural activity during a decision making task. They claim to have found a new dynamical mechanism that better explains their data than a line attractor (Mante) or bistable attractor (Wong and Wang). They also propose a modified DDM that they argue better explains their data than the DDM. I agree with the authors that it is important to test dynamical systems

mechanisms from neural data, to understand the neural substrate behind decision making. However, I don't agree with the authors that their claims are substantiated by their analyses. After reading this manuscript, I actually have renewed faith in the Mante, Wong/Wang and Bogacz papers as simplified but robust (and timeless) models of decision making. If the authors can point out where I misunderstood I am happy to reconsider, because I think the motivation behind this study is very important for the field.

We are very grateful to the reviewer for their careful and thorough reading of the manuscript. We take one of the reviewer's points to be that if one is to challenge well-established hypotheses (such as Mante, Wong/Wang or Bogacz), one had better have very strong evidence to back this challenge up, and that it was not clear that evidence of the requisite strength was being presented.

We appreciate the point and we agree with it. We are grateful for the reviewer's openness to seeing whether we can indeed provide such evidence. We believe we are now able to present the required strong evidence. Following the reviewer's comments, and as described in more detail below, we have:

- corrected our writing so our description of the vector fields in the text matches what FINDR inferred.
- simplified and improved our method to infer latent dimensionality, and we believe we now conclusively show that 2 latent dimensions are better than 1 (Extended Data Fig. 5c).
- fit FINDR to many more sessions than before and developed quantitative metrics of the key observations in these fits; using these metrics we can now show that the key results are consistent across subjects and sessions (Fig. 2i,j,k,l) and consistent across cross-validation folds (Extended Data Fig. 6a,b,c).
- developed and carried out head-to-head comparisons between the FINDR results and the previous 3 hypotheses, which we believe now conclusively show that the FINDR-inferred vector fields are a better description of the data than the previous 3 hypotheses (Extended Data Fig. 7).

Further points are in the detailed responses below.

We hope the reviewer will find the new results as convincing as we did.

Major Concern 1:

The authors have claimed to have found evidence of bistable attractor dynamics. A close look at Figure 2c,d reveals that this description does not match their own vector fields. First of all, the apparent fixed points are only slow points as the speed never gets to zero. However, even more concerning, is that the vector field *around the slow points* are not consistent with attractor dynamics. The field around $[-.5,6]$ points *away* from the slow region in the upper right plane, and similarly for the bottom right region of the phase plane for the bottom left attractor. There are no fixed point attractors in this phase plane. (If anything, it looks like there might be a stable limit cycle but that is aside the point) If the authors don't agree, I would urge them to replot this vector field in a normalised form (since the magnitudes are shown in the speed plot anyway).

If we inspect ED Figure 6a, we see that the vector fields for each recording have completely different dynamical landscapes. Some of these (T176_2018_05_31) do appear to have attractors corresponding to the two choices, but also have several spurious fixed points that aren't explained in terms of computational mechanism. Others (X046_2019_09_23) only have one fixed point! From what I see, the authors' claims do not match their own data.

Thank you for raising these concerns. From what we understand, the concerns expressed here are two: 1) Describing the vector fields as containing attractors was incorrect. 2) Different sessions lead to different dynamics, so what is consistent across sessions is unclear.

Regarding (1), we must apologize for our writing. Indeed, the reviewer is correct, describing the dynamics as bistable was simply incorrect. Many sessions do not show bistability or have even one attractor. Describing the vector fields correctly is essential, and we are grateful to the reviewer for pointing out the problem. However, we must stress that none of the conclusions of our study depend on bistability or the presence of attractors in the dynamics: indeed, correcting the problem in our writing did not change anything in our title, abstract, conclusions, or discussion.

Why is it that correcting our original description doesn't change any conclusions? It is because our task had a maximum time of 1.0 second from stimulus onset to stimulus offset. This means that in practice, our data cannot make a distinction between dynamics that, on a timescale of one second, approach an attractor versus dynamics that go through a slow point. Moreover, the key features in the flow fields that led to the simplified MMDDM model and the discovery of nTc as a marker for decision commitment do not depend on the presence of attractors in the flow fields. Nevertheless, the reviewer is absolutely correct that how we describe the vector fields should match the actual vector fields. We are embarrassed that this wasn't the case, and have removed references to the FINDR-inferred flow fields having attractors or bistability.

While none of our conclusions depended on whether the endpoints of the dynamics were attractors or even slow points, the conclusions *do* depend on the nature of the dynamics before the endpoints, namely the transition from input-dominated to autonomous-dominated dynamics, and the change in direction of the neural trajectories (**Fig. 2**). Our description in the text now focuses on these aspects.

2) The second concern the reviewer raised here is in regards to consistency across sessions. We agree that this is an important point-- conclusions should only be drawn from features that are consistent across sessions.

There are two key aspects in the FINDR-inferred flow fields that are used to motivate MMDDM and draw the main conclusions about decision commitment. The first is that, as time unfolds, there is a transition from input-dominated to autonomous-dominated dynamics. The second is that there is a substantial change in the direction in which the neural trajectories evolve, i.e., a neural mode change. We now document that those two features are consistent across sessions and subjects (please see new Figs. 2i,j,k,l).

A third interesting aspect that is not part of our main conclusions regarding decision commitment, but is nevertheless interesting and is consistent across all sessions, is that the origin was never recovered as a stable or saddle point (both eigenvalues have positive real parts, Extended Data Fig. 6n). In other words, there is not even an approximate line attractor as in Wong and Wang 2006, because there is no 1-dimensional stable manifold.

Major Concern 2:

The neural network function approximates used to estimate $F(z,u)$ is likely to be highly degenerate and have multiple solutions that give similar performance. Therefore, I don't believe that the authors have sufficiently ruled out other dynamical mechanisms.

We thank the reviewer for this important comment. To address the possibility of degeneracy in the neural network used to estimate $F(z, \mathbf{u})$, we confirmed that

- The dynamics are similar across different subsets of trials from a session (Extended Data Fig. 6a-c)
- The dynamics are similar across different random seeds used to initialize the $F(z, \mathbf{u})$ network (Extended Data Fig. 6h).
- FINDR-recovered synthetic dynamics are very similar to the generating dynamics (Extended Data Fig. 6f).

Most importantly, we will again stress the point that the main features our conclusions depend on are consistent across subjects and sessions. It is impossible for the FINDR-inferred dynamics to be exactly identical across sessions. The important parts are the features that are consistent across sessions, now documented in Fig. 2i,j,k,l.

In 2a, are only the green connections plus a scalar baseline per hidden unit fitted?

Thank you for this important question. We want to first clarify that a separate baseline is fit for each *neuron* and not for each hidden unit. We have now added a summary of the calculation of the baseline in Methods Section 5.4: Baseline, while retaining the more detailed description in the Supplementary Information as we have done previously.

We now describe in detail the parameters that were optimized in FINDR in Methods Section 3: FINDR. The green connections in **Fig. 2a** schematize the neural networks parametrizing the function F , which was used to compute the vector fields in all the figures. In addition to fitting the parameters in F , we also fitted the covariance matrix Σ of the additive Gaussian noise and the encoding weight matrix W . Moreover, because the posterior probability distributions of z given the spike trains and sensory inputs are needed to compute the loss function during optimization, we fitted an additional neural network G that approximates the posterior probability distributions of z . (We are working within a sequential variational autoencoder framework.) The neural networks parametrizing F and G are independent. Importantly, G was never used to generate any vector field shown in the manuscript and was used only to learn F , Σ , and W .

Moreover the inconsistency of the identified vector fields in ED Figure 6 makes me especially wonder if the neural network model is just too parameterized. Could the authors precise how many parameters it has?

Thank you for this helpful comment. We have now specified the number of parameters in Methods Sections 3.1-3.2. Large parameter count and the possibility of overfitting are common to many neural network models. We address overfitting by imposing an L2 regularization weight of 10^{-7} on all parameters, and also optimizing the *number of parameters* in each neural network as a hyperparameter. The high out-of-sample R^2 's measuring the goodness-of-fit of our model suggest that our model did not overfit (Extended Data Fig. 4).

Can they provide evidence that other dynamical mechanisms like the line attractor cannot explain neural activity as well? In Fig 2h, the trajectories on the right look extremely similar to the line attractor shown in Mante where the trajectories move to different points along the line determined by the strength of evidence. On this point, even within the model class proposed by the authors, it doesn't appear clear to me that their mechanism is a better fit than other mechanisms.

Thank you very much for this very important question. We completely agree, a head-to-head comparison between the results from FINDR versus previous hypotheses was needed. This was not

trivial, for to our knowledge, while those previous hypotheses are very well known, they have never been fit directly to spiking data. There are thus no well-established methods to do that.

(Note: for clarity, we have renamed the “RNN line attractor” hypothesis (Mante et al., 2013) to be “non-normal dynamics, line attractor.”)

To fit the three previous hypotheses in a way that best facilitates a direct apples-to-apples, head-to-head comparison against FINDR, we developed a version of FINDR in which the vector field function $F()$, instead of being a gated neural network, was constrained to be a function that allows only the three pre-existing hypotheses (see Methods Section 3.6: Constrained FINDR (cFINDR)). Different settings of that function’s parameters can implement the three pre-existing hypotheses. We called this constrained version “cFINDR”. We could then use the machinery of FINDR to fit to the data either cFINDR (flow fields must conform to one of the three pre-existing hypotheses), or the full FINDR (flow fields can be anything parametrized by the gated neural network).

We first confirmed, by generating synthetic spiking data from each of the previously existing hypotheses, that fitting cFINDR to that synthetic data recovers the hypothesis used to generate the spikes (Extended Data Fig. 7b-g).

We then fit both cFINDR and the full FINDR to the experimental data and asked, using out-of-sample data not used in the fitting, which of the two described the data better. cFINDR has far fewer parameters than full FINDR, so all else being equal, one would expect cFINDR to generalize to out-of-sample data better than full FINDR. Nevertheless, the results consistently and strongly favored the full FINDR across all 27 sessions (Extended Data Fig. 7i). We believe that this conclusively demonstrates that the full FINDR-inferred flow fields are a better description of the data than any of the three previously existing hypotheses.

We have added the following section to the Results in the main text. Please note the section in italics (it is in italics for emphasis only here, not in the main text itself), which was prompted by the reviewer’s comment that the Mante et al. curved trajectories look similar to the ones FINDR inferred. The section in italics points out a difference that we believe underlies the better fits from FINDR.

To perform a head-to-head comparison with the three hypotheses of **Fig. 1d-h**, we constructed a variant of FINDR in which the gated neural network that parametrizes $F()$ was replaced by a parametrization of the dynamics that was constrained to describe the three pre-existing hypotheses (Extended Data Fig. 7). If the data were well-described by one of these three hypotheses, we would expect this variant (which we refer to as cFINDR, for constrained FINDR) to fit the data well, and in particular, to fit out-of-sample data sets better than FINDR, since it has far fewer parameters than FINDR. However, unconstrained FINDR consistently fit the data better than cFINDR, confirming that previous hypotheses do not adequately capture the data. *While the previous hypothesis suggesting non-normal dynamics with a line attractor can generate curved trial-averaged trajectories apparently similar to those we see in the data (Extended Data Fig. 7g), there is a key difference, which is that in this previous hypothesis, the turn from the initial flow direction induced by the inputs happens early, for the autonomous dynamics causing it are strong the moment the latent state departs from the line attractor. However, our data suggests that there is a more prolonged initial phase of flow along the input directions before the turn, with the stronger autonomous dynamics happening much later in the decision-making process. We believe this underlies the much better fits to the data for FINDR than with cFINDR.*

In ED Figure 4 the authors show R^2 (is it cross-validated?) but only for one example session. They should show summary statistics across all data.

Thank you very much for this important comment. We have modified the labels on all plots in Extended Data Fig. 4 to indicate that the R^2 is cross-validated. Extended Data Fig. 4 now shows not only an example session but summary statistics across all 27 sessions to which FINDR was fit (the 27 sessions were selected on the criteria of >400 trials, >30 choice-selective neurons, and >80% fraction of trials correctly completed by the animal).

In ED Figure 5 it is unclear first of all how the cross validation is working and what the bits/spike is used rather than a standard measure like variance explained, mean squared error, etc. More importantly, it looks like a 1-dimensional latent variable has the same explanatory power as the 2-dimensional system that they have chosen. But, in the 1-d case, this would then just result in a standard bistable attractor as in the Wang and Wong paper they cite.

Thank you for this helpful comment. Extended Data Fig. 5a-e now report out-of-sample R^2 of the evidence-sign conditioned PSTH instead of bits/spike across sessions. Using this metric, we found that a 2-dimensional system has more explanatory power than a 1-dimensional system, either across choice-selective neurons pooled across sessions (Extended Data Fig. 5a-b) or across sessions (Extended Data Fig. 5c).

We wish to further clarify that as in the initial submission, even though FINDR with dimensionality > 2 can better capture the data, the vast majority of the variance is captured within the first 2 latent dimensions (Extended Data Fig. 5d), and the vector fields and trajectories projected onto the first two dimensions are qualitatively similar (Extended Data Fig. 5e). Therefore, as in the initial submission, the Results focus on the dynamics using FINDR with 2-dimensional latent dynamics.

Minor concerns:

Mechanistically, and computationally (in terms of what it means for the decoder) I don't understand the idea behind the MMDDM that the weights change after the decision commitment. Is this the only difference from a DDM? Could the authors help to give some interpretation?

Thank you for this important question. Yes, the difference between MMDDM and a DDM consists in only a change in the neurons' encoding weight of the latent accumulator variable after decision commitment; the latent dynamics are identical between MMDDM and a DDM. To emphasize the point that the MMDDM is very similar to the DDM, when we introduce it we now describe it as

In what we will call the “multi-mode” or “minimally-modified” drift-diffusion model (MMDDM)

Mechanistically, the weight change after decision commitment could result from inputs from ascending midbrain neurons. This possibility is suggested by a recent finding in a working memory task that midbrain neurons, in response to an external auditory cue, trigger rapid reorganization of motor cortex activity to switch from planning-related activity to a motor command that initiates movement in mice (Inagaki et al., 2022).

Computationally, a decoder that aims to read out a categorical behavioral choice from the neural activity can improve its accuracy by selectively reading out from neurons whose post-commitment encoding weights are large in magnitude, while a decoder that aims to read out a graded representation of accumulated evidence would more strongly weigh neurons whose pre-commitment encoding weight is large in magnitude.

We have added to the Discussion the computational (in terms of a decoder) and mechanistic interpretations of the weight change after commitment.

The neural mode change indicates that a downstream decoder of the categorical choice can improve its accuracy by selectively reading out from neurons whose post-commitment weights are large in magnitude. A possible mechanism for the neural mode change is an input from ascending midbrain neurons, which is suggested by a recent finding in a working memory task that midbrain neurons, in response to an external auditory cue, trigger rapid reorganization of motor cortex activity to switch from planning-related activity to a motor command that initiates movement in mice (Inagaki et al., 2022).

I am not sure how to interpret the results considering the variability of the recorded regions across sessions. Could this explain why the vector fields give such different results?

Thank you very much for this question. We would like to first re-emphasize that, in terms of a transition from input-dominated to autonomous-dominated dynamics, and a change in neural mode, the results are consistent across sessions, subjects, and recorded regions (Figs. 2i,j,k,l).

Going beyond these features, we have not yet found a reliable relationship between variability in the vector fields and differences in the identity of the recorded regions. However, we found a reliable negative correlation between the animal's behavioral sensitivity in each session and the relative strength of the autonomous vs. input dynamics before the change in the neural mode in that session. This correlation indicates that some of the variability in the vector fields depends in part on the variability in the animal's behavioral performance. We have incorporated this finding in Extended Data Fig. 6d.

Will the data and code be made available?

Yes, absolutely.

Lack of mathematical precision:

* Throughout, "intrinsic dynamics" should be replaced by "autonomous dynamics", which is the more precise term.

Thank you for this helpful comment. Our previous use of the term "intrinsic dynamics" was motivated by other papers in the literature, such as the review Vyas et al., 2020, or Vahidi et al., 2023 or Vinograd et al., 2024, but we are happy to change "intrinsic dynamics" to "autonomous dynamics" throughout the manuscript, and have now done so.

It should also be clarified in the text that the input-driven dynamics are determined by the input and the autonomous/intrinsic dynamics together, not just the input (e.g. in line 68-69 which is not true in general, only in your proposed model where the autonomous dynamics are zero before decision commitment).

Thank you for this valuable comment. We have modified line 68-69 to state:

The input dynamics are the changes in z driven by sensory inputs \mathbf{u} , which can be distinguished from the autonomous dynamics as $F(z, \mathbf{u}) - F(z, \mathbf{0})$. The input dynamics can depend on z (**Fig. 1e**; Extended Data Fig. 1c-e).

* In a bistable dynamical system there are two stable fixed points and a saddle, not an unstable fixed point as stated on line 158 and elsewhere.

Thank you for this helpful correction. We have amended the manuscript so that when we refer to the bistable dynamical mechanism, we specify there is a saddle point, and that only along a 1-D manifold is the saddle point an unstable fixed point.

* Line 212 - Gaussian with diagonal covariance. Isotropic or non? This was also unspecified in the methods.

Thank you for this helpful comment. We have specified that the diagonal covariance is not constrained to be isotropic in the legend of **Fig. 2a**:

The deterministic component F is approximated using a gated feedforward network, and stochasticity $\boldsymbol{\eta}$ is modeled as a Gaussian with diagonal covariance (not constrained to be isotropic).

and also in the Methods:

136

Σ is a d -dimensional diagonal matrix, where the diagonal elements

137 need not be equal to each other. At each time step, firing rates of N simultaneously recorded neurons r_i are given by

* Methods line 114-120: Unclear what this is accomplishing, please clarify this section and include equations / better notation to explain this section.

Thank you for pointing out the lack of clarity in this section. It is very helpful. We have revised the notation and included additional equations to better explain this section, which is now Methods Section 3.3: Latent space transformation.

* Methods Section 9: Equations 8-10 are imprecise. It appears you are doing some kind of mean-field approximation or expectation of the latent dynamics over the input stochasticity. But, this can be derived exactly and written in precise formulation, rather than just weighting the equation by the conditional probabilities. Moving from Equation 7 to Equation 8 is improper, and needs better notation which will help in contrast to the picture in Figure 1 that shows the discrete and stochastic inputs.

Thank you very much for this helpful comment. We have changed Methods Section 2 to correct the notations used in previous Equations 7-8 (now Equations 4-5). Moreover, we corrected previous Equation 10 (now Equation 6) to specify that we are indeed computing the expectation of the latent dynamics over the input stochasticity (specifically, $p(\mathbf{u} | z)$, the conditional distribution of the input \mathbf{u} given the latent z). We furthermore explain that we computed the $p(\mathbf{u} | z)$ using simulations rather than deriving it because $p(z)$ does not have an analytical form.

* Lines 153-160 which describe a novel model have no equations. The only equation for MMDDM that I see is in Figure 3c where it appears that there is actually no dynamics - it's just evidence accumulation with a stochastic transition to a decision?

Thank you for suggesting this clarification. The dynamics of MMDDM is indeed “just evidence accumulation with a stochastic transition to a decision”—identical to the dynamics of a behavioral DDM. The only modification is that the neural encoding changes at the same time when evidence accumulation transitions to decision commitment.

We have expanded Methods Section 4: MMDDM to include 14 equations; previously, the equations were in the Supplementary Information.

Referee #3:

Luo et al. examined attractor dynamics of populations of neurons recorded from various brain regions of rats performing perceptual decision making. Using their new method called FINDR, they revealed population dynamics that differed from existing models. In the early period, there were little intrinsic dynamics and the population state was largely modulated by sensory input. In the later period, the population state was more strongly modulated by intrinsic dynamics that were not aligned with the direction of the input dynamics. Thus, there was a transition in dynamics from input-driven to intrinsic. The authors propose that the earlier state corresponds to evidence accumulation and the later state corresponds to decision commitment. They developed an extension of DDM (MMDDM) that fitted the neural data well and was consistent with the results of psychophysical reverse correlation. The authors claim that their model accounts for a variety of previously reported phenomena such as curved neural trajectories and ramping vs. stepping.

The paper performed extensive data collection and sophisticated analysis including the application of their novel analysis method and the development of a new computational model. These were executed with high quality and explained effectively in the manuscript. Revealing the system dynamics (attractors) underlying cognitive functions has been of great interest to the systems neuroscience community, and the authors' successful application of their new method is commendable. Furthermore, the authors built a new model (MMDDM) with great attention to the details of its implementation. However, after carefully reading the manuscript, I would like to raise several important questions and concerns. My main concerns are the extent of the conceptual advance provided by their conclusions, the validity of focusing on a representative dataset to draw their conclusions, and the possibility of alternative models.

Major concern 1:

The main conclusion of the paper is the transition from an evidence accumulation mode to a decision commitment (or action execution) mode, but there are multiple previous studies that reported distinct response dynamics for evidence accumulation and action planning in various task settings (Bennur & Gold 2011, Park et al. 2014, Okazawa et al. 2021, Charlton & Goris 2022, Yates et al. 2017). These studies have often shown that there are early response dynamics reflecting decision formation and then

it is replaced by later dynamics encoding choice or action plan. The authors have cited and discussed these studies (page 12), but after reading it, I still find it difficult to judge whether the conclusions of this paper conceptually go beyond these previous findings. I understood that the present paper brought a quantitative framework, directly fitting the attractor and input dynamics to the data and providing a full model (MMDDM) to account for this phenomenology. Thus, I do not doubt the novelty of the paper. Nevertheless, I am not fully convinced that this quantitative modeling of known properties significantly updates/changes our perspective on the neural mechanisms of decision making enough to be published in *Nature*.

We are very grateful for this comment—it very much helped us clarify and highlight what is new, different, and important in our work, in comparison to previous studies such as the ones the reviewer cited. The key point is that our major findings with respect to the neural mode change are about decision commitment, not action preparation, and these two are very distinct concepts that correspond to different neural events.

There are three key distinctions between our findings and the previous findings:

- 1) After decision commitment, further sensory evidence is ignored, since the subject has already committed to a choice. But the beginning and development of action preparation/planning do not imply that decision formation has come to a halt or that further sensory evidence will not affect the subject's choice. For example, preliminary action preparation in perceptual decision-making tasks is commonly observed to begin even before the sensory stimulus is presented, in part driven by choice biases induced by previous trials (Yates 2017, also Extended Data Fig. 14). Such early action preparation in no sense implies that sensory evidence subsequent to it will be ignored. In contrast, decision commitment implies that, once the subject has committed to a choice, subsequent evidence *will* be ignored (for example, within the behavioral DDM framework, this corresponds to reaching a “sticky” bound: the evidence accumulator “sticks” there, meaning it ignores subsequent evidence inputs).

Thus a critical panel in our manuscript is Fig. 3n, which presents behavioral analysis showing that *before* each trial's neurally-inferred time of commitment (“nTc”), sensory evidence affects the subject's decisions; but *after* nTc, further evidence is indeed ignored. These are the critical data that demonstrate that nTc indeed corresponds to decision commitment, not simply action planning. We are not aware of any previous study with a similar or even close trial-by-trial observation linking neural activity and behavioral evidence of decision commitment. We have changed the title of **Fig. 3** to emphasize the importance of this panel.

Of the papers the reviewer cited, Bennur 2011, Park 2014, and Yates 2017 focus on action planning and do not refer in any way whatsoever to commitment. Okazawa 2021 refers to commitment only to indicate that recordings shown in their final Fig. 7 (from areas PAG and SEF) were done using a reaction time task, for which it is assumed that response initiation is immediately subsequent to decision commitment. Charlton 2022 are the closest to our work, since they do indeed refer to commitment per se, and hypothesize that a change in neural mode could correspond to commitment. But they used a task in which all the sensory evidence (a static visual stimulus) was presented simultaneously at the beginning of the trial. This precluded measuring whether evidence presented *after* a putative commitment time is ignored by the subject, and thus they could not obtain data with which to either confirm or refute their hypothesis.

- 2) The previous studies cited (which are all nonhuman primate studies) were mostly carried out before the advent of high-yield recording methods for primates, and they all averaged activity over trials. Here, in contrast, Neuropixel probes allowed simultaneous recording from many tens to hundreds of units. These simultaneous recordings form the fundamental basis on which nTc can be estimated on each individual trial, which is essential: the new panels in Fig. 3m,l (which we should have included in the original submission) show that nTc is very broadly distributed across trials. That is, nTc is an internal event that is not time-locked to stimulus onset (Fig. 3m), nor to stimulus offset (Extended Data Fig. 10n), nor to motor response onset (Fig. 3l). Without knowledge of nTc for each individual trial, then, it is essentially impossible to observe phenomena time-locked to nTc, and it would have therefore been impossible to obtain the critical neural-behavioral panel Fig. 3n. (If we align data to any of stimulus onset, offset, or motor response onset, the step in Fig. 3n is blurred out so much as to become invisible.)

In sum, precise estimation of nTc on each trial is critical, and could not be done in the previous studies.

- 3) While Fig. 3n is a critical behavioral observation, our manuscript contains an equally important neural observation relating to the distinction between commitment and action planning: at the neural level, ignoring further sensory inputs corresponds to input dynamics becoming weak compared to autonomous dynamics. Observing this requires estimating the autonomous and the input dynamics. Estimating those flow fields is a core part of what the FINDR method does (it does not merely estimate trajectories in latent space), which led to the finding that after the change in neural mode, input dynamics become negligible compared to the autonomous dynamics (Fig. 2).

Notably, this signature of decision commitment is entirely distinct from action planning, which could occur with or without the input dynamics becoming negligible. To our knowledge, flow fields have never before been estimated for perceptual decision-making, and there is therefore no previous paper making a similar observation.

In sum, our study is conceptually distinct from the previous studies cited in (a) reporting correlates at the neural level of decision commitment (not merely action planning), in the sense that after commitment, but before movement onset, sensory inputs no longer affect neural activity in the recorded regions; (b) using large-scale simultaneous recordings to estimate, on a single trial by single trial basis, the putative time of commitment (“nTc”), which was found to be highly variable, and not time-locked to stimulus onset, offset, or motor response onset; (c) furnishing proof at the behavioral level of nTc corresponding to commitment in the sense that, despite its broad variability across trials, nTc is time-locked to the moment after which further sensory evidence no longer affects the subject’s decisions.

All of these are about commitment, not action preparation or execution. To our knowledge, none have been reported previously. We added a paragraph in the Discussion to include these points. It reads

While the timing of the nTc signal reported here makes it very distinct from motor execution, the signal is also distinct from action preparation or planning. The beginning of action planning carries no implication as to whether sensory evidence presented subsequent to that will or will not be ignored. Indeed, in perceptual decision-making tasks, preliminary action preparation, driven by choice biases induced by previous trials, is often observed to begin even before the sensory stimulus, as reported previously^{54,56,62} and found in our own data (Extended Data Fig. 14). In contrast, commitment to a decision implies that evidence presented subsequent to the commitment will no longer affect the subject’s

choice. Here we found that nTc corresponds to such a decision commitment moment. This was the case both at the neural level, where it correlates with a substantial decrease in the effect of sensory inputs on neural responses in the regions we recorded (**Fig. 2**); and at the whole-organism behavioral level, in the sense that sensory evidence before nTc affects the subject's choices, but sensory evidence after nTc does not (**Fig. 3n**).

Following the reviewer's citation of Yates 2017, we thought it would be valuable to follow their procedure and ask *when* the Yates 2017 approach provided evidence of action preparation, so as to compare it to our estimated commitment time nTc . In their GLM-based approach, the action preparation estimate comes in the form of movement kernels, one for left choices and one for right choices. We fit a family of 7 different GLM's that differed only in the time of the *onset* of the two movement-related kernels. These varied, across the seven GLMs, from 3 seconds before movement onset to 0 seconds before movement onset, and always ended 0.5 seconds after movement onset (Extended Data Fig. 14). Using out-of-sample log-likelihood, we asked which start time for the movement kernel accounted best for the out-of-sample data.

In contrast to the inferred times of decision commitment, which is mostly limited to 1 to 0 seconds before movement onset (Extended Data Fig. 14g), we found that for most neurons, the encoding of the action plan begins well before the inferred times of decision (Extended Data Fig. 14e) and even before the onset of the auditory clicks (Extended Data Fig. 14f). These results are consistent with Yates 2017 and Park 2014. We include them here as evidence that action planning and decision commitment are distinct not only conceptually, but also at the neural level.

An important claim of the paper is that MMDDM provides a parsimonious explanation for a variety of experimental findings (page 12), which is probably meant to propose MMDDM as a unified model of perceptual decision making, but I am a bit reserved about this statement. First, while the authors cited Mante 2013 and Aoi 2020, MMDDM does not explain why curved trajectories could become context dependent (the lack of intrinsic dynamics in MMDDM cannot explain how the network could become context dependent). Second, the model is said to account for diversity in single neuron dynamics (e.g., Meister et al 2013) and mixed stimulus and choice selectivity, but mixed selectivity is quite common in any RNNs or other latent state models. Thus, I thought this statement does not give MMDDM a unique status.

Thank you for the helpful comment. We agree that a central claim is that MMDDM provides a parsimonious explanation for a variety of experimental findings, but we *do not* propose MMDDM to be a unique or unified model of perceptual decision making, for the same reasons that you have pointed out. We think of MMDDM as a stepping stone toward a unified model—a simple yet useful approximation. We see the major contribution of MMDDM coming from being used to provide behavioral confirmation that there exists an internal event during perceptual decision-making, which can be inferred from spiking data, after which sensory inputs are, behaviorally speaking, ignored (Fig. 3n).

To clarify this, we have added to the Discussion:

However, we do not see MMDDM as a unique or a unified model of perceptual decision-making. Rather, we see it as a simple yet useful approximation, a “minimally-modified” DDM, and a stepping stone toward a unified model of decision-making.

Major concern 2:

The key figures showing the attractor and input dynamics (Fig. 2c-h) are from a representative session. These figures look great, but the results from all the sessions should be shown to support this. The authors show the results from a few more sessions in Extended Fig. 6, and not surprisingly, there is some variability in the patterns across sessions, which makes it hard to read whether they indicate the same things. I am not sure if FINDR could be fitted to all the data combined, or if it could be averaged across sessions. If that is not possible, the authors can generate metrics such as the relative strength of input and intrinsic dynamics, the angle between the input and commitment dynamics, and plot their distributions across sessions. Fig. 3i-j shows the results from all sessions, but this only compares the model fit accuracy between MMDDM and DDM and does not directly support the claims made in Fig. 2c-h.

Yes, completely agreed and thank you for the suggestions. We have now fitted FINDR to all sessions that had more than 400 trials, 30 or more simultaneously recorded choice-selective neurons, and behavior with overall performance greater than 80%. (FINDR requires a sufficient number of trials and neurons to be able to obtain reliable estimates.) This comprises 27 sessions. We followed the reviewer's suggestion and developed metrics for relative strength of input vs autonomous dynamics, and of the angle of the trajectories across all the fitted sessions, which we now report in Fig. 2i-l. These new panels now show that the key findings—a switch from input-dominant to autonomous-dominant dynamics, together with a large turn in the direction of the neural trajectories—are consistent across subjects and sessions.

Data from all 27 sessions are now included in all measures of quality of fits (Extended Data Fig. 4d), estimates of latent space dimensionality (Extended Data Fig. 5a-d), measures of consistency in the FINDR-inferred dynamics (Extended Data Fig. 6a-c), and head-to-head comparison between previous hypotheses vs FINDR (Extended Data Fig. 7i; this last we discuss more extensively in our response to R3 Major Comment 4 below.)

Major concern 3:

According to FINDR, there is little intrinsic dynamics during decision formation along both z_1 and z_2 axes, but I was not sure whether the dynamics along z_2 axis could be convincingly inferred from the data. The population state starts from the origin in the space and quickly diverges toward z_1 axis due to input (Fig. 2h), so I thought there were not many cases where the population state goes along z_2 axis, then the FINDR-inferred dynamics along this axis are not very reliable. The authors wrote “only the well-sampled subregion of the state space is shown” (page 7) so I thought my concern did not apply, but I then realized that the same subregion was drawn for all the time points shown in Fig. 2h. If the authors draw a well-sampled subregion for each time point, is it possible that, early in the decision (0.33s), the neural state is confined to a narrower region that expands along z_1 ? If so, z_2 axis does not play a critical role in decision formation (because the neural state rarely goes along that direction)? This might require some revision of the authors' discussion regarding the role of this axis such as history effects (page 13).

Thank you for the question. In our original submission, we only showed trajectories that were averages over trials (each one an average over a given evidence strength). Showing only these trial-averaged trajectories was a mistake on our part, for it failed to give readers a sense of the magnitude of the noise

in the stochastic individual-trial trajectories, and may have created the impression that the region of the latent space sampled in the first 0.33 seconds was extremely narrow. Extended Data Fig. 6m now shows how the latent space is sampled by individual trials. Particularly for trials with the weakest overall evidence (lightest blue and lightest pink), trajectories are dominated by noise, which moves them off the trial-averaged trajectories, and they sample the region around the origin well.

We followed the reviewer's suggestion, and computed the zone that is well-sampled within the first 0.33 sec. (Although we note that later times can also contribute to sampling around the origin.) We refer to this region as the "early-epoch sample zone," and now show it in Extended Data Fig. 6j. We interpret the epoch of time=[0, 0.33s] as the *initial phase* of the decision formation process, rather than as decision formation *per se*. We computed principal components (PCs) for the trajectories truncated at time=0.33s. The first of the early-epoch PCs lies approximately in the direction of the trial-averaged trajectories, while the second one, orthogonal to the first, captures noise w.r.t. the trial averages. As suspected by the reviewer, the standard deviation along this PC2 direction was significantly narrower than along PC1, it was 20% of the standard deviation along PC 1 (Extended Data Fig. 6k). Nevertheless, variability along this early-epoch PC2 is significant enough to decode behavioral choice from logistic regression significantly better than chance (Extended Data Fig. 6l). This demonstrates that position along the early epoch PC2 can indeed affect ultimate choices, as we had speculated in our discussion.

We are grateful for this comment and suggestion from the reviewer. It both helped us to clarify, and led to what we felt were interesting new results in Extended Data Fig. 6l.

Major concern 4:

The authors brought up multiple attractor models in Fig. 1, but as far as I could see, they did not directly test (fit) the RNN line attractor (Mante 2013). For the line attractor, Fig. 3i shows the difference in out-of-sample log-likelihood between the authors' MMDDM and DDM (instantiated as a line attractor) and the results look promising (also bistable attractors in Extended Fig. 12). Line and bistable attractors only predict a linear neural trajectory, so it makes sense that they perform poorly in explaining the observed curved trajectories. But it is not clear how poorly the RNN line attractor fits the data compared to the authors' model, since both models predict similar curved trajectories. Is it possible to perform a quantitative comparison?

Thank you, yes, we completely agree, a head-to-head comparison between the results from FINDR versus previous hypotheses was needed. This was not trivial, for to our knowledge, while those previous hypotheses are very well known, they have never been fit directly to spiking data. There are thus no well-established methods to do that.

(Note: for clarity, we have renamed the "RNN line attractor" hypothesis (Mante et al., 2013) to be "non-normal dynamics, line attractor.")

To fit the three previous hypotheses in a way that best facilitate a direct apples-to-apples, head-to-head comparison against FINDR, we developed a version of FINDR in which the vector field function $F()$, instead of being a gated neural network, was constrained to be a function that allows only the three pre-existing hypotheses (see Methods Section 3.6: Constrained FINDR (cFINDR)). Different settings of that function's parameters can implement the three pre-existing hypotheses. We called this constrained

version “cFINDR”. We could then use the machinery of FINDR to fit to the data either cFINDR (flow fields must conform to one of the three pre-existing hypotheses), or the full FINDR (flow fields can be anything parametrized by the gated neural network).

We first confirmed, by generating synthetic spiking data from each of the previously existing hypotheses, that fitting cFINDR to that synthetic data recovers the hypothesis used to generate the spikes (Extended Data Fig. 7b-g).

We then fit both cFINDR and the full FINDR to the experimental data and asked, using out-of-sample data not used in the fitting, which of the two described the data better. cFINDR has far fewer parameters than full FINDR, so all else being equal, one would expect cFINDR to generalize to out-of-sample data better than full FINDR. Nevertheless, the results consistently and strongly favored the full FINDR across all 27 sessions (Extended Data Fig. 7i). We believe that this conclusively demonstrates that the full FINDR-inferred flow fields are a better description of the data than any of the three previously existing hypotheses.

We have added the following section to the Results in the main text:

To perform a head-to-head comparison with the three hypotheses of **Fig. 1d-h**, we constructed a variant of FINDR in which the gated neural network that parametrizes $F()$ was replaced by a parametrization of the dynamics that was constrained to describe the three pre-existing hypotheses (Extended Data Fig. 7). If the data were well-described by one of these three hypotheses, we would expect this variant (which we refer to as cFINDR, for constrained FINDR) to fit the data well, and in particular, to fit out-of-sample data sets better than FINDR, since it has far fewer parameters than FINDR. However, unconstrained FINDR consistently fit the data better than cFINDR, confirming that previous hypotheses do not adequately capture the data. While the previous hypothesis suggesting non-normal dynamics with a line attractor can generate curved trial-averaged trajectories apparently similar to those we see in the data (Extended Data Fig. 7g), there is a key difference, which is that in this previous hypothesis, the turn from the initial flow direction induced by the inputs happens early, for the autonomous dynamics causing it are strong the moment the latent state departs from the line attractor. However, our data suggests that there is a more prolonged initial phase of flow along the input directions before the turn, with the stronger autonomous dynamics happening much later in the decision-making process. We believe this underlies the much better fits to the data for FINDR than with cFINDR.

Major concern 5:

MMDDM inferred that the commitment (bound crossing) occurred in 34.6% of the trials (Method, page 6), so the accumulated evidence did not reach a bound before stimulus offset in the other ~65% of the trials. This number looks reasonable, but I wonder how FINDR and MMDDM treat stimulus offset in these trials. Stimulus offset is a go cue in this task, so it would trigger rats to commit to a choice and initiate their action, so I expect it to become a strong external drive to influence the neural dynamics (presumably forcing the network state to make a transition to the commitment mode). What do neural trajectories look like near stimulus offset? The models do not need to account for this?

To clarify the task timing (also detailed in Extended Data Fig. 15a): The onset of fixation is followed after a variable period by the stimulus onset; the stimulus offset (which coincides with the “Go” cue) is always 1.5 s after fixation onset (thus stimulus duration is variable); and the onset of the movement to

report the decision is initiated by subject, and is therefore occurs a short (~200 ms) but variable period after stimulus offset (Extended Data Fig. 15b). The rats were trained on this one and only task timing for months.

The question of what happens at stimulus offset on trials without nTc is an interesting one, although we do wish to emphasize that our manuscript focuses on neural dynamics during the stimulus itself. Nevertheless, following the reviewer's suggestion, we examined the neural trajectories near stimulus offset (Extended Data Fig. 15). We did not observe substantial abrupt changes in neural activity time-locked to stimulus offset, on either trials on which a neurally inferred time of commitment (nTc) could not be inferred. In contrast, the substantial rapid changes in neural activity occurred after movement onset (which we did not include in analyses using either MMDDM or FINDR).

To carry out the analysis (Extended Data Fig. 15c-d), we performed principal component analysis (PCA) on peri-event time histograms (PETH) aligned to stimulus offset across. The PETH's ran from -0.5 s before stimulus offset to 0.5 after stimulus offset, including spikes after the onset of the animal moving away from the fixation port (i.e., movement onset). PETH's were not additionally filtered. The neural changes after stimulus offset are more closely aligned to movement onset than to stimulus offset (Extended Data Fig. 15c-d).

Given the lack of any substantial abrupt change in the neural activity aligned to stimulus offset, even on trials without nTc, we decided to not include stimulus offset as an external input in either FINDR or MMDDM. We think this makes sense because the timing of stimulus offset is the same on every trial aligned to the moment of fixation, and the rat has been trained on this timing for months. Had we varied the timing of stimulus offset relative to the onset of fixation, then treating stimulus offset as an input to the latent dynamics would certainly have been necessary.

Minor concerns:

1.

The authors contrasted MMDDM with "classic DDM" (Fig. 3), but this classic DDM was instantiated as a line attractor. Behavioral DDM does not specifically prescribe how it should be implemented in neural circuits, so is it more accurate to call it "line attractor" instead of "classic DDM"?

Thank you for this helpful suggestion. We have revised our nomenclature of "classic DDM" to be "DDM line attractor."

2.

Fig. 3m: how many sessions/rats were used to calculate the reverse correlation? Could there be error bars in the plot or statistical tests?

Thank you for this helpful question. The reverse correlation did not exclude any session or rat and used all 115 sessions and 12 rats of the manuscript's dataset. We have now noted this in the legend of Fig. 3n (previously Fig. 3m). To reduce visual clutter, error bars and statistical tests (in the form of model comparisons) are separately provided in Extended Data Fig. 10b-d.

3.

Extended Fig. 4: Is this the result of a single session? Could the authors plot the fit quality of all the sessions?

Thank you for this comment. We have added Extended Data Fig. 4d to plot the fit quality of all 27 sessions to which FINDR was fit (the 27 sessions were selected on the criteria of >400 trials, >30 choice-selective neurons, and >80% fraction of trials correct).

4.

The authors selected ~10% of choice selective neurons to perform all the analysis (Method, page 3). But neurons that do not have strong choice selectivity could still contribute to the population dynamics. Does this selection affect the results of their analysis?

Thank you for this helpful question. We compared the vector field inferred using either only choice-selective neurons or to all neurons that have a firing rate > 2 spikes/s (a criterion that all “choice-selective” neurons met) for the representative session shown in **Fig. 2**. We found that both the dynamics and the trial-averaged trajectories to be highly similar whether we used only choice-selective neurons or both choice-selective and non-choice selective neurons. This is shown in Extended Data Fig. 6g.

Appendix

Figure 2i-l

Figure 2... **i**, To quantify how the difference in the speed between autonomous and input dynamics changes over the course of a trial, we identify the time point at which the latent trajectories curve (indicated as stars), and compute the difference in \mathbf{g} before and after this time point. **j**, To define when the trajectories curve, we compute the curvature of the trial-averaged trajectories and define the “peak” to be the time point when the curvature is maximum. We then define 200ms time periods with respect to the peak (“pre-peak” and “post-peak”), and with respect to the start and end of the trial (“early” and “late”). **k**, Difference in the speed between autonomous and input dynamics for five different time periods (“start (time=0s)” “early”, “pre-peak”, “post-peak”, and “late”), computed across vector fields inferred from recording sessions that had more than 30 neurons and 400 trials, and sessions where the animal performed with greater than 80% accuracy ($n=27$). The difference is normalized to lie between -1 and 1. **l**, For sessions where the FINDR model with two-dimensional decision variable z fit significantly better than the FINDR model with one-dimensional z ($n=21$ out of 27; Extended Data Fig. 5), the direction of motion of the trial-averaged trajectories is computed, and its angle with respect to the z_1 -axis for different time periods is shown.

Figure 3l,m,n

Figure 3l,m,n, nTc occurs at highly variable times within the trial but precisely indicates the moment when sensory evidence ceases affecting a subjects’ decisions. **l**, nTc is highly variable across trials relative to stimulus onset. **m**, nTc is also highly variable relative to the moment the rat leaves the fixation port. The leftmost bin contains trials on which nTc occurred more than 1s before movement. **n**, When trials are aligned to nTc, behavioral analysis (using logistic regression between stimulus clicks and the subject’s choice) shows that sensory evidence before nTc affects the subject’s decision, but sensory evidence after nTc does not.

Extended Data Figure 4

Extended Data Figure 4. FINDR can well capture the neural responses. **a-b**, FINDR captures the underlying firing rates of the single-trial responses of individual neurons from the representative session in Fig. 2. **c**, FINDR captures the complex trial-averaged dynamics of individual neurons from the representative session in Fig. 2 as can be seen in the peristimulus time histograms (PSTH). The goodness-of-fit is measured using the coefficient of determination (R^2). **d**, FINDR captures the single-trial and trial-averaged responses of individual neurons pooled across 27 sessions. For the histogram showing single trials pooled across sessions, 34 trials that had $R^2 < 0$ are not shown. Results in **a-d** are 5-fold cross-validated.

Extended Data Figure 5a-e

Extended Data Figure 5. FINDR reveals 2-dimensional decision-making dynamics. **a**, Across different FINDR models with latent dimensions (d) ranging from 1 to 4, we computed the median of the coefficient of determination (R^2) of the evidence-sign conditioned peri-stimulus time histogram (PSTH) of neurons pooled across sessions. **b**, The median difference in the R^2 between $d=2$ and $d=1$ is significantly different from zero ($p < 0.001$; Wilcoxon signed-rank test). Although the median differences are also significant for the comparison between $d=3$ and $d=2$ and the comparison between $d=4$ and $d=3$, the magnitude of the difference is relatively small (0.0098 and 0.0075, respectively) compared to the median difference between $d=2$ and $d=1$ (0.0423). **c**, We repeated the analysis in **b** without pooling neurons across sessions. Instead, for each session, we computed the median PSTH R^2 across neurons recorded within that session. Each circle corresponds to a session, and a filled circle indicates a

significant difference in the PSTH R^2 between FINDR models of different dimensionalities ($p < 0.001$; Wilcoxon signed-rank test). **d**, For FINDR models with either 3 or 4 latent dimensions, more than 97% of the variance is captured by the first two principal components (PC's). PCA was done separately for each session, and the error bars indicate the 95% confidence interval of the median across sessions. **e**, For models with 2 or more dimensions, the vector fields and trajectories projected onto the first two dimensions are qualitatively similar. The vector fields and trajectories were shown for the representative session in **Fig. 2**. The dashed lines demarcate the well-sampled subregion of the state space (i.e., the sample zone).

Extended Data Figure 6a-c

Extended Data Figure 6. Consistency in FINDR-inferred dynamics. **a**, FINDR-inferred input and autonomous dynamics are consistent across 5 different cross-validation folds, as shown for the same session in **Fig. 2**. **b**, Normalized difference in the speed between autonomous and input dynamics for five different time periods (“start (time=0s)”, “early”, “pre-peak”, “post-peak”, and “late”) is consistent across folds. **c**, The direction of motion of the trial-averaged trajectories and its angle with respect to the z_1 -axis for different time periods is consistent across folds.

Extended Data Figure 6d

d

Extended Data Figure 6... d, Variability in the dynamics across sessions depends in part on the variability in the behavioral performance. For each each session, behavioral sensitivity was estimated as the parameter β in a probit model $p(y | x) = \Phi(\beta * x + c)$, where y is the rat’s left vs. right choice on each trial, x the log-ratio of the right vs. left click rate on that trial, Φ the normal cumulative distribution function, c the constant term in the probit model. The p-value of the Pearson’s correlation was computed using a Student’s t-distribution for a transformation of the correlation. Pink marker indicates the example session.

Extended Data Figure 6f

f

Extended Data Figure 6... f, FINDR reliably recovers the FINDR-inferred dynamics. After fitting FINDR to a dataset, the optimized model parameters were used to simulate a synthetic dataset using the exact same set of sensory stimuli in the real dataset and containing the same number of neurons and trials. From new initial parameter values, FINDR was fit to the simulated data to infer the “FINDR-generated” vector fields.

Extended Data Figure 6g

g

Extended Data Figure 6... g, FINDER is fit to both choice-selective and non-selective neurons. We find similar dynamics to when FINDER is fit to only choice-selective neurons.

Extended Data Figure 6h

h

Extended Data Figure 6... h, We find vector fields that are consistent across multiple different random seeds that change the initialization in the deep neural networks of FINDR and the order in which the mini-batches of the training data are supplied to FINDR during training.

Extended Data Figure 6i

Extended Data Figure 6... i, Curved trial-averaged latent trajectories predicted by FINDR depend on the click inputs. When FINDR was fit to data in which the click inputs were randomly shuffled across trials, the trial-averaged latent trajectories remain near the origin.

Extended Data Figure 6j-l

Extended Data Figure 6... j, The dynamics are two-dimensional even in the beginning of the decision period. An early-epoch sample zone indicated by the dotted line was computed using trajectories that were truncated at time=0.33s. The early-epoch sample zone delimits the portion of the state occupied by at least 50 of 5000 simulated single-trial trajectories. **k**, When we compute the PC's for the trajectories truncated at time=0.33s and project the trajectories onto PC 2, the standard deviation along this direction is 20.4% of the standard deviation along PC 1. **l**, We can decode behavioral choice from logistic regression significantly better than chance (dashed line) from the projections of the truncated trajectories onto PC 2.

Extended Data Figure 6m

Extended Data Figure 6... m, Single-trial latent trajectories extending to time=1.0s, simulated using stimuli of different evidence strength, which is quantified by the ratio of right and left inputs.

Extended Data Figure 6n

Extended Data Figure 6... n, When we computed the eigenvalues of the numerical Jacobian obtained from the detected slow point around the origin, the real component of both eigenvalues were above zero for all sessions ($n=27$).

Extended Data Figure 7

Extended Data Figure 7. The data are better captured by FINDR than by a variant of FINDR constrained to parametrize the dynamics described by previously proposed hypotheses. **a**, The constrained FINDR (cFINDR) model replaces the neural networks parametrizing F in FINDR with a linear combination of affine dynamics, specified by M and N , and bistable attractor dynamics specified by φ . The dynamics are furthermore constrained to be two-dimensional. **b-g**, cFINDR model can

generate and infer dynamics described by previous hypotheses. **b**, Example bistable attractor dynamics generated from cFINDR. **c**, Example DDM line attractor dynamics generated from cFINDR. **d**, Example non-normal dynamics with a line attractor generated from cFINDR. **e**, cFINDR-inferred dynamics from a synthetic dataset generated using the bistable attractor dynamics in **b**. **f**, cFINDR-inferred dynamics from a synthetic dataset generated using the DDM line attractor dynamics generated in **c**. **g**, cFINDR-inferred dynamics from a synthetic generated using the non-normal dynamics with a line attractor in **d**. **h**, cFINDR-inferred dynamics and FINDR-inferred dynamics on a real representative session. **i**, The coefficient-of-determination (R^2) of the evidence-sign conditioned peri-stimulus time histogram (PSTH) computed using fits of FINDR to the data is significantly greater than the R^2 's computed using fits of cFINDR (Wilcoxon signed-rank test).

Extended Data Figure 10b-d

Extended Data Figure 10... b, To determine the time resolution of the kernel that best captures the weight of the input fluctuations, 10-fold cross-validation was performed to compare kernels quantified by different numbers of parameters and types of basis functions. The kernel with the lowest temporal resolution is a constant, represented by a single parameter, implying that fluctuations across time have the same weight. At the highest time resolution, the kernel can be parametrized by a separate weight for each time step. At intermediate time resolution, the kernel is parametrized by basis functions that span the temporal window. The basis functions can be evenly spaced across the temporal window, or stretched such that time near $t=0$ s is represented with higher resolution and time far from $t=0$ s with lower resolution. The most likely model had six moderately stretched ($\eta=1$) basis functions. **c**, The optimal model's set of six moderately stretched ($\eta=1$) basis functions. **d**, Even when using basis functions, the psychophysical kernel is consistent with the core prediction of MMDDM: The psychophysical weight of the stimulus fluctuations on the behavioral choice ceases after the time of decision commitment. Note that no basis function was used in the analysis in Fig. 3n.

Extended Data Figure 10k-l

Extended Data Figure 10... k, Whether nTc could be inferred on a given trial has a small but statistically significant effect on that trial's "movement onset time," i.e., the time when the rat withdraws its nose from the fixation port minus the earliest time when the rat is allowed to do so. The effect is not simply due to trial difficulty because it remains when we consider only easy trials (right : left click rate either greater than 38:1 or less than 1:38). I, Similar effect of whether commitment was reached on the rat's "movement execution time," i.e., the time when the rat reaches either the left or right port minus the time when it withdrew its nose from the fixation port.

Extended Data Figure 10m

m

Extended Data Figure 10... m, Comparison between the neurally inferred times of commitment (nTc) between pairs of simultaneously recorded brain regions, separately using neurons from different brain regions. For each pair of regions, the difference was computed using only the subset of trials on which the nTc could be separately inferred using spike trains from either region.

Extended Data Figure 10n

n

Extended Data Figure 10... n, Distribution of nTc times relative to stimulus offset. Note that with trial durations uniformly distributed between [0.2, 1.0] seconds, there are more short duration trials than long duration trials, accounting for part of the rightwards shift of the distribution.

Extended Data Figure 14

Extended Data Figure 14. The distribution of commitment times inferred from MMDDM does not match the distribution of start time of peri-movement kernels. **a**, Separately for each choice-selective neuron ($N=4605$), peri-movement kernels are estimated using Poisson generalized linear models (GLM) (Park et al. 2014; Yates et al. 2017). The inputs to the model depend on two events that occur on each trial: onset of fixation (i.e., when the rat inserts its nose into the center port), and the time when the rat leaves the center port and begins to move toward the side port. An impulse (i.e., delta function) at the time of each event is convolved with a linear filter, or kernel, to parametrize the time-varying input related to that event. At each time step, the sum of the inputs is fed through a rectifying nonlinearity (softplus) to specify the neuron's Poisson firing rate at that time. Three kernels, related to fixation, leftward movement, and rightward movement, are learned by maximizing the marginal likelihood (Park et al. 2014). **b**, Example neuron. Two GLM variants were fitted to the same neuron, and for each GLM variant, the observed peri-event time histogram (PETH) is overlaid the cross-validated, model-predicted PETH. The choice-dependence of the PETH of this neuron is well captured by the model variant whose peri-movement kernels start -3.0 s before and 0.5 s after movement onset (left), but less well captured by another variant whose peri-movement kernels time base are limited to -0.5 to 0.5 s (right). **c**, To identify the optimal start of the movement kernel for each neuron, cross-validated (5-fold) model comparison was performed on seven model variants that vary in the start time of the movement kernels and the number of radial basis functions used to parametrize the kernels. The end time of the movement kernel (0.5 s), and the parametrization of the fixation-related kernel (-1.5 s to 2.0 s and 4 basis functions) are identical for all variants. **d**, The out-of-sample log-likelihood is highest for the model variant whose peri-movement kernels start at -3.0

s. **e**, For each neuron, the GLM variant with the highest out-of-sample log-likelihood determines the optimal start of the peri-movement kernels. The mode of the distribution is at -3.0s. **f**, The start of peri-movement kernels for most neurons precede the time of the first click. **g**, The start of peri-movement kernels for most neurons precede the earliest commitment time inferred from MMDDM.

Extended Data Figure 15

Extended Data Figure 15. Changes in neural responses after stimulus offset are more closely aligned to movement onset than stimulus offset. **a**, Relative timing of task events. The offset of the auditory click train stimulus always occurred at the end of the 1.5 s minimum fixation period on every trial. **b**, The median time of movement onset relative to stimulus offset across trials without a neurally inferred time of commitment (nTc) is 0.192s. The rightmost bin contains trials for which the movement onset is 0.8s or more after stimulus offset. **c**, Principal component analysis (PCA) was performed on peri-event time histograms (PETH's) aligned to stimulus offset (circles) and averaged across trials without a neurally inferred time of commitment (nTc). Spikes were counted in 10 ms bins, and the PETH was not additionally filtered. Spikes after the animal moved away from the fixation port (i.e., movement onset) were included. For each neuron and each trial condition, the PETH is a 100 element vector. Concatenating across 4605 choice-selective neurons and 4 trial conditions gave a 4605-by-400 matrix. The mean of each row (i.e., the average response of each neuron) was subtracted from the matrix, and PCA was performed on the resulting matrix. Triangles indicate the median time of movement onset. Projections are scaled by the standard deviation explained by each PC. **d**, PCA performed PETH's aligned to movement onset offset and averaged across trials without nTc.

Methods Section 2

96 2 Autonomous and input dynamics

97 The class of dynamical systems we study here is specified by

$$\dot{z} = F(z, \mathbf{u}) \quad (1)$$

98 for some generic function F , with z the latent decision variable and \mathbf{u} the external input to the system from the auditory
99 clicks in the behavioral task. At each moment, there may be no click, a click from the left, or a right click. When time
100 is discretized to sufficiently short steps, \mathbf{u} is one of three values

$$\mathbf{u} = \begin{cases} [0; 0] = \mathbf{0} & \text{representing when there is no click,} \\ [1; 0] & \text{representing when there is a left click,} \\ [0; 1] & \text{representing when there is a right click.} \end{cases} \quad (2)$$

101 We define the autonomous dynamics of the system as

$$\dot{z}_{\text{autonomous}} = F(z, \mathbf{0}), \quad (3)$$

102 and the average input dynamics as

$$\dot{z}_{\text{input}} = p(\mathbf{u}|z)[F(z, \mathbf{u}) - F(z, \mathbf{0})], \quad (4)$$

103 and specifically, the average left and right input dynamics as

$$\begin{aligned} \dot{z}_{\text{left}} &= p(\mathbf{u} = [1; 0]|z)[F(z, [1; 0]) - F(z, \mathbf{0})], \\ \dot{z}_{\text{right}} &= p(\mathbf{u} = [0; 1]|z)[F(z, [0; 1]) - F(z, \mathbf{0})]. \end{aligned} \quad (5)$$

104 The sum of the autonomous dynamics and the average input dynamics is equal the expected value of \dot{z} computed over
105 the distribution $p(\mathbf{u}|z)$:

$$\begin{aligned} \mathbb{E}[\dot{z}] &= \sum_{\mathbf{u}} p(\mathbf{u}|z)F(z, \mathbf{u}) \\ &= p(\mathbf{u} = \mathbf{0}|z)F(z, \mathbf{0}) + p(\mathbf{u} = [1; 0]|z)F(z, [1; 0]) + p(\mathbf{u} = [0; 1]|z)F(z, [0; 1]) \\ &= \left[1 - p(\mathbf{u} = [1; 0]|z) - p(\mathbf{u} = [0; 1]|z)\right]F(z, \mathbf{0}) + p(\mathbf{u} = [1; 0]|z)F(z, [1; 0]) + p(\mathbf{u} = [0; 1]|z)F(z, [0; 1]) \quad (6) \\ &= F(z, \mathbf{0}) + p(\mathbf{u} = [1; 0]|z)[F(z, [1; 0]) - F(z, \mathbf{0})] + p(\mathbf{u} = [0; 1]|z)[F(z, [0; 1]) - F(z, \mathbf{0})] \\ &= \dot{z}_{\text{autonomous}} + \dot{z}_{\text{left}} + \dot{z}_{\text{right}} \end{aligned}$$

106 In the main text, Figure 2c plots $\dot{z}_{\text{autonomous}}$, and Figure 2e plots \dot{z}_{left} and \dot{z}_{right} . $F(z, \text{left})$ in the main text is defined
107 as $p(\mathbf{u} = [1; 0]|z)F(z, [1; 0]) + [1 - p(\mathbf{u} = [1; 0]|z)]F(z, \mathbf{0})$ and $F(z, \text{right})$ as $p(\mathbf{u} = [0; 1]|z)F(z, [0; 1]) + [1 - p(\mathbf{u} =$
108 $[0; 1]|z)]F(z, \mathbf{0})$.

109 Since $p(\mathbf{u}|z) = p(z|\mathbf{u})p(\mathbf{u})/p(z)$, and $p(z)$ in general does not have an analytical form, we estimate $p(\mathbf{u}|z)$ numeri-
110 cally. To do this, we train FINDR (Section 3, [3]) to learn F , and generate click trains for 5000 trials in a way that is
111 similar to how clicks are generated for the task done by our rats. Then, we simulate 5000 latent trajectories from the
112 learned F and the generated click trains. We then bin the state space of z and ask, for a single bin, how many times the
113 latent trajectories cross that bin in total and how many of the latent trajectories, when crossing that bin had $\mathbf{u} = [1; 0]$
114 (or $\mathbf{u} = [0; 1]$). That is, we estimate $p(\mathbf{u} = [1; 0]|z)$ with $\frac{\# \text{ of latent states with } \mathbf{u}=[1;0] \text{ in the bin that covers } z}{\# \text{ of latent states in the bin that covers } z}$. For Figure 2 in the
115 main text, because z is 2-dimensional, we use bins of 8-by-8 that cover the state space traversed by the 5000 latent
116 trajectories, and weigh the flow arrows of the input dynamics with the estimated $p(\mathbf{u}|z)$. Similarly, for the background
117 shading that quantifies the speed of input dynamics in Figure 2, we use bins of 100-by-100 to estimate $p(\mathbf{u}|z)$, and ap-
118 ply a Gaussian filter with $\sigma = 2$ (in the units of the grid) to smooth the histogram. A similar procedure was performed
119 in Extended Data Figures 3 and 6 to estimate $p(\mathbf{u}|z)$ numerically.

Methods Section 3: FINDR

3 FINDR

Detailed descriptions are provided in [3]. Briefly, to infer velocity vector fields (or flow fields) from the neural population spike trains, we used a sequential variational autoencoder (VAE) called Flow-field Inference from Neural Data using deep Recurrent networks (FINDR).

FINDR minimizes a linear combination of two losses: one for neural activity reconstruction (\mathcal{L}_1) and the other for vector-field inference (\mathcal{L}_2). To reconstruct neural activity, FINDR uses a deep neural network G that takes the spike trains of N simultaneously recorded neurons \mathbf{y} and the sensory click inputs \mathbf{u} on a given trial, and reconstructs \mathbf{y} from the d -dimensional latent decision variable \mathbf{z} :

$$\mathbf{z}_{t+1} = \mathbf{z}_t + \Delta t G(\mathbf{z}_t, \mathbf{u}_{1:T}, \mathbf{y}_{1:T}) + \boldsymbol{\eta}_t, \quad t = 1, 2, 3, \dots \quad (7)$$

Here, T is the number of time steps on a given trial, \mathbf{u}_t is a two-dimensional vector representing the number of left and right clicks played on a time step ($\Delta t = 0.01$ s), \mathbf{y}_t an N -dimensional vector of the spike counts on a time step, and $\boldsymbol{\eta}_t$ noise drawn from $\mathcal{N}(\mathbf{0}, \Delta t \boldsymbol{\Sigma})$ on each time step. $\boldsymbol{\Sigma}$ is a d -dimensional diagonal matrix, where the diagonal elements need not be equal to each other. At each time step, firing rates of N simultaneously recorded neurons \mathbf{r}_t are given by

$$\mathbf{r}_t = \text{softplus}(\mathbf{W}\mathbf{z}_t + \mathbf{b}_t) \quad (8)$$

where softplus is a function approximating the firing rate-synaptic current relationship (f-I curve) of neurons, \mathbf{W} a $N \times d$ matrix representing the encoding weights, \mathbf{b}_t a N -dimensional vector representing the putatively decision-irrelevant baseline input. The baseline \mathbf{b}_t is learned prior to fitting FINDR using the procedure described in Section 5.4 and in detail in the Supplementary Information. The reconstruction loss is given by

$$\mathcal{L}_1 = - \sum_{t=1}^T \log \text{Poisson}(\mathbf{y}_t | \mathbf{r}_t). \quad (9)$$

For vector field inference, we parametrize the vector field F with a gated feedforward neural network [2, 3]:

$$\dot{\mathbf{z}} \approx \frac{\mathbf{z}_{t+\Delta t} - \mathbf{z}_t}{\Delta t} = F(\mathbf{z}_t, \mathbf{u}_t). \quad (10)$$

F gives the discretized time derivative of \mathbf{z} . We find the vector field F that captures the latent trajectories \mathbf{z} inferred from G in Eq. (7) by minimizing

$$\mathcal{L}_2 = \sum_{t=1}^T [F(\mathbf{z}_t, \mathbf{u}_t) - G(\mathbf{z}_t, \mathbf{u}_{1:T}, \mathbf{y}_{1:T})]^\top \boldsymbol{\Sigma}^{-1} [F(\mathbf{z}_t, \mathbf{u}_t) - G(\mathbf{z}_t, \mathbf{u}_{1:T}, \mathbf{y}_{1:T})]. \quad (11)$$

The total loss that is minimized by FINDR is

$$\mathcal{L} = \mathcal{L}_1 + c\mathcal{L}_2 \quad (12)$$

where $c = 0.1$ is a fixed hyperparameter ($c = 0.0125$ in Extended Data Figure 3b). FINDR minimizes \mathcal{L} by using stochastic gradient descent (SGD) to learn \mathbf{W} , $\boldsymbol{\Sigma}$, the parameters of the neural network representing F , and the parameters of the neural network G . It can be shown that \mathcal{L} is an approximate upper bound on the marginal log-likelihood of the data, and that training FINDR this way is equivalent to performing inference and learning via a sequential auto-encoding variational Bayes (AEVB) algorithm that straightforwardly extends the standard AEVB [4, 3].

After training, we plot the vector field (i.e., a grid of $\dot{\mathbf{z}}$) using the learned F , and generate FINDR-predicted neural responses using Eq. (8) and

$$\mathbf{z}_{t+\Delta t} = \mathbf{z}_t + \Delta t F(\mathbf{z}_t, \mathbf{u}_t) + \boldsymbol{\eta}_t. \quad (13)$$

Eq. (13) is an Euler-discretized gated neural stochastic differential equations (gnSDE [2, 3]).

Methods Section 3.1-2

154 3.1 Parameters

155 The total number of free parameters P of the FINDR model is given by

$$\begin{aligned}
 P &= P_W + P_\Sigma + P_F + P_G, \\
 P_W &= N \times d, \\
 P_\Sigma &= d, \\
 P_F &\in \{90 + (64 + d)d, 150 + (104 + d)d, 300 + (204 + d)d\}, \\
 P_G &\in \{15900 + 300N + 100x + P_F, 61800 + 600N + 200x + P_F, 243600 + 1200N + 400x + P_F\}.
 \end{aligned} \tag{14}$$

156 P_W is the number of parameters in the encoding weight matrix \mathbf{W} , whose dimensions are the number of neurons N
 157 and latent dimensionality d . P_Σ is the parameter count in the diagonal covariance Σ of the additive Gaussian noise
 158 of the latent \mathbf{z} . The number of parameters in the neural networks parametrizing F (P_F) and G (P_G) are separate
 159 hyperparameters. Here, $x = \frac{P_F - d + d^2}{2d + 3}$.

160 3.2 Hyperparameters

161 The hyperparameters that were optimized (P_F, P_G, α) include the number of parameters of the network F (P_F), the
 162 number of parameters of the network G (P_G), and the learning rate $\alpha \in \{10^{-2}, 10^{-1.625}, 10^{-1.25}, 10^{-0.875}, 10^{-0.5}\}$. We
 163 identify the optimal values for these hyperparameters in $3 \times 3 \times 5 = 45$ grid search. The grid search was done separately
 164 for each set of training data for each of five cross-validation folds. Within each each training set, 3/4 of the trials were
 165 used to the optimize the parameters under a given set of hyperparameters, and the remaining 1/4 were held out to
 166 evaluate the model's performance for that set of hyperparameters. Test data were never used in the grid search.

Methods Section 3.3

167 3.3 Latent space transformation

168 Because the encoding weight matrix \mathbf{W} is not constrained to semi-orthogonal and can take only any real values,
 169 different combinations of \mathbf{W} and \mathbf{z}_t can give rise to the same firing rate vector \mathbf{r}_t , even when baseline \mathbf{b}_t is fixed. To
 170 uniquely identify the latent trajectories (except for redundancy from rotations and reflections), after optimization, we
 171 linearly transformed the latent space \mathbf{z} to $\tilde{\mathbf{z}}$

$$\tilde{\mathbf{z}}_t = \mathbf{S} \mathbf{V}^\top \mathbf{z}_t \tag{15}$$

172 where \mathbf{S} a $d \times d$ diagonal matrix containing the singular values of \mathbf{W} and \mathbf{V} a $d \times d$ matrix containing the right singular
 173 vectors

$$\mathbf{W} = \mathbf{U} \mathbf{S} \mathbf{V}^\top \tag{16}$$

174 \mathbf{U} is an $N \times d$ matrix containing the left singular vectors of \mathbf{W} (where N is the number of neurons). In the space of $\tilde{\mathbf{z}}$,
 175 the encoding weight matrix is a linear transformation that preserves angles and distances because \mathbf{U} is semi-orthogonal
 176 and can only give rise to an isometry such as rotation and reflection.

$$\begin{aligned}
 \mathbf{W} \mathbf{z} &= \mathbf{U} \mathbf{S} \mathbf{V}^\top \mathbf{z} \\
 &= \mathbf{U} \tilde{\mathbf{z}}
 \end{aligned} \tag{17}$$

177 To obtain meaningful axes for the transformed latent space $\tilde{\mathbf{z}}$, we generate 5000 different trajectories of $\tilde{\mathbf{z}}$ in
 178 generative mode (i.e., using F and Σ in Eq. (13), but not G), and perform principal component analysis (PCA) on
 179 the trajectories. The principal components (PCs) were used to define the axes of the decision variable $\tilde{\mathbf{z}}$. In the main
 180 text, the PC 1 axis of $\tilde{\mathbf{z}}$ was denoted as \mathbf{z}_1 , and the PC 2 axis of $\tilde{\mathbf{z}}$ was denoted as \mathbf{z}_2 . In all our analyses, the latent
 181 trajectories and vector fields inferred by FINDR are shown in the transformed latent space of $\tilde{\mathbf{z}}$, and scaled so that the
 182 latent trajectories along PC 1 lie between -1 and 1 .

Methods Section 3.6: cFINDR

205 3.6 Constrained FINDR (cFINDR)

206 The constrained FINDR model (cFINDR) replaces the neural networks parametrizing F in FINDR with a linear com-
 207 bination of affine dynamics, specified by \mathbf{M} and \mathbf{N} , and bistable attractor dynamics specified by φ . The dynamics are
 208 furthermore constrained to be two-dimensional.

$$\begin{aligned} \dot{\mathbf{z}} &\approx \frac{\mathbf{z}_{t+\Delta t} - \mathbf{z}_t}{\Delta t} = F(\mathbf{z}_t, \mathbf{u}_t) = \mathbf{M}\mathbf{z}_t + \mathbf{N}\mathbf{u}_t + s \cdot \varphi(\mathbf{z}_t), \\ \mathbf{M} &= \mathbf{Q}\mathbf{\Lambda}\mathbf{Q}^{-1}, \\ \mathbf{Q} &= \begin{bmatrix} 1 & \sin(\theta) \\ 0 & \cos(\theta) \end{bmatrix}, \\ \mathbf{\Lambda} &= \begin{bmatrix} 0 & 0 \\ 0 & -r \end{bmatrix}, \\ \varphi(\mathbf{z}_t) &= -\exp(-(\mathbf{z}_t - \mathbf{x})^2/\rho) \odot (\mathbf{z}_t - \mathbf{x}) - \exp(-(\mathbf{z}_t + \mathbf{x})^2/\rho) \odot (\mathbf{z}_t + \mathbf{x}). \end{aligned} \tag{18}$$

209 The matrix \mathbf{M} implements a line attractor located at $z_2 = 0$. The inputs \mathbf{u}_t are the same as in FINDR and represent the
 210 auditory clicks. The two discrete attractors are constrained such that $x_2 = 0$ and implemented through the function φ .
 211 The shape of the basin of attraction corresponding to each point attractor is specified by the parameter ρ . The relative
 212 contribution of the discrete attractors and the line attractor to the overall dynamics is specified by the scalar s .

213 The DDM line attractor hypothesis can be implemented in cFINDR by setting $\theta = 0$. Non-normal dynamics with
 214 a line attractor [6] can be implemented by setting $\theta \neq 0$. The bistable attractor hypothesis can be implemented by
 215 increasing ρ .

Methods Section 4: MMDDM

235 4 MMDDM

236 The multi-mode drift-diffusion model (MMDDM) is an instance of a state-space model, which consists of a dynamic
 237 model governing the time evolution of the probability distributions of the latent (i.e., hidden) states and measurement
 238 models specifying the conditional distributions of the observations (i.e., emissions) given the value of the latent state.

239 4.1 Dynamic model

240 The evolution of the scalar latent variable z is a piecewise linear function:

$$z(t+1) = \begin{cases} z(t) + u(t+1) + \eta, & -B < z(t) < B \\ B \cdot \text{sign}\{z(t)\}, & \text{otherwise} \end{cases} \quad (19)$$

241 When the absolute value of z is less than the bound height B (free parameter), its time evolution depends on momentary
 242 external input u and i.i.d. Gaussian noise η .

$$\eta \sim \mathcal{N}(0, \Delta t) \quad (20)$$

243 where Δt is the time step and set to be 0.01s (10 milliseconds). When z is either less than $-B$ or greater than B , it
 244 becomes to be fixed at the bound. The initial probability distribution of z is given by

$$z(t=1) \sim \mathcal{N}(\mu_0, 1) \quad (21)$$

245 whose mean μ_0 is a free parameter. On time step t , the input $u(t)$ is the total difference in the per-click input between
 246 the right and left clicks that occurred in the time interval $[t - \Delta t, t)$

$$u(t) = \sum_{\tau \in R} v(\tau; t) - \sum_{\tau \in L} v(\tau; t) \quad (22)$$

247 where $L(R)$ is the set of the left (right) click times and $v(\tau; t)$ is the per-click input of a click emitted at time τ on
 248 the latent variable at time step t . Note that $\tau \in \mathbb{R}$ indicates continuous time, whereas $t \in \mathbb{N}$ indexes a time step. The
 249 per-click input is given by

$$v(\tau; t) = D(\tau; t) \cdot C(\tau) \cdot \zeta \quad (23)$$

Continued on the next page:

250 where $D(\tau; t)$ indicates the integral over the interval $[t - \Delta t, t)$ of the Direct delta function δ delayed by τ :

$$D(\tau; t) = \int_{t-\Delta t}^{t-\epsilon} \delta(x - \tau) dx = \begin{cases} 1, & \tau \in [t - \Delta t, t) \\ 0, & \text{otherwise} \end{cases} \quad (24)$$

251 where ϵ is the machine epsilon. To account for sensory adaptation, the per-click input is depressed by preceding clicks
252 by a time-varying scaling factor given by the function $C(\tau)$, implemented according to previous work [1] (Supplemen-
253 tal Information). The per-click input is corrupted by i.i.d. multiplicative Gaussian noise ζ :

$$\zeta \sim \mathcal{N}(1, \sigma_s^2) \quad (25)$$

254 The free parameter σ_s^2 is the variance of the per-click noise. Variability in the dynamic model is fit to the data through
255 the per-click noise ζ rather than per-time step noise η based on previous findings [1]; our results are similar if we set
256 the variance of η rather than the variance of ζ to be a free parameter.

257 The dynamic model has three free parameters: bound height B , variance σ_s^2 of the per-click noise, and mean μ_0 of
258 the initial state. These parameters are learned simultaneously with the parameters of the measurement models.

259 4.2 Measurement model of behavioral choices

260 On each trial, the binary behavioral choice c (1=right, 0=left) is the sign of z on the trial's last time step T (the earlier
261 of 1 s after the onset of the clicks or immediately before the animal leaves the fixation port):

$$c \mid z(T) = \text{sign}\{z(T)\} \quad (26)$$

262 4.3 Measurement model of spike counts

263 On each time step t , given the value of z , the spike count y of neuron n is a Poisson random variable

$$y^{(n)}(t) \mid z(t) \sim \text{Poisson}(\lambda^{(n)}(t) \Delta t) \quad (27)$$

264 The firing rate λ is given by

$$\lambda^{(n)}(t) \mid z(t) \equiv h(w^{(n)} \cdot z(t) + b(t)) \quad (28)$$

265 where h is the softplus function used to approximate the neuronal frequency-current curve of a neuron:

$$h(x) = \log\{1 + \exp(x)\} \quad (29)$$

266 The encoding weight of z depends on z itself:

$$w^{(n)} = \begin{cases} w_{EA}^{(n)}, & -B < z < B \\ w_{DC}^{(n)}, & z \in \{-B, B\} \end{cases} \quad (30)$$

267 Each neuron has two scalar weights, w_{EA} and w_{DC} , that specify the encoding of the latent variable during the evidence
268 accumulation (EA) regime and the decision commitment (DC) regime, respectively. When the latent variable has not
269 yet reached the bound ($-B$ or B), all simultaneously recorded neurons encode the latent variable through their own
270 w_{EA} , and when the bound is reached, their own w_{DC} .

271 The bias b incorporates nuisance variables as input to the neural spike trains and is the sum of a component that
272 varies only across trials and a component that varies both across trials and over time steps t of each trial m :

$$b^{(n)}(m, t) = b_{\text{cross}}^{(n)}(m) + b_{\text{within}}^{(n)}(m, t) \quad (31)$$

273 The cross-trial component, $b_{\text{cross}}^{(n)}$ is a function of time in minute m from the first trial of the session (whereas t
274 indicates time within each trial relative to that trial's stimulus onset). The within-trial component consists of time-
275 varying influence from spike history, post-stimulus onset, and pre-movement onset.

$$b_{\text{within}}(m, t) = \tau_{\text{stim}}^{(m)} (k_{\text{stim}} * \delta)(t) + \tau_{\text{move}}^{(m)} (k_{\text{move}} * \delta)(t) + \sum_i \tau_{\text{spike}}^{(m,i)} (k_{\text{spike}} * \delta)(t) \quad (32)$$

276 where the symbol $*$ indicates convolution, τ_x indicates translation $\tau_x k(t) = k(t - \tau_x)$ by the time of event x , and δ is the
 277 Dirac delta function. The functions $b_{\text{cross}}^{(n)}$, k_{stim} , k_{move} , k_{spike} are learned and each parametrized as a linear combination
 278 of radial basis functions, following [9, 10] (Supplemental Information). Each neuron’s measurement model of the
 279 spike train has 19 parameters that are learned simultaneously with the parameters of the dynamic model (i.e., model
 280 of the latent variable).

Methods Section 5.4: Baseline

340 5.4 Baseline

In FINDR, cFINDR, and MMDDM, the firing rate of a neuron depends on a time-varying scalar baseline. On timestep t of trial m , conditioned on the value of the latents on a given time step, the spike count y of each neuron is given by

$$\left(y(m, t) \mid z(m, t) \right) \sim \text{Poisson} \left(h \{ \mathbf{w}^\top z(m, t) + b(m, t) \} \right)$$

341 where h is the softplus function and \mathbf{w} is the encoding weight of the latent. The baseline b incorporates putatively
 342 decision-independent variables as input to the neural spike trains including slow drifts in firing rates across trials and
 343 faster changes within each trial that are aligned to either the time from stimulus onset or the time from the animal
 344 leaving the fixation port. The baseline is learned using a Poisson generalized linear model fit separately to the spike
 345 counts of each neuron. Details are provided in the Supplementary Information.

We thank the reviewers for their comments. We will use blue for reviewer comments and black for our responses. For easy reference, we have also included an Appendix with copies of many figure panels within this document; the hyperlinks (e.g., Extended Data Figure 16a) point to these panels within this document, so reviewers can simply click and find the corresponding figure panel. No access to the Internet is required.

Referee #1 (Remarks to the Author):

Thanks to the authors for writing a detailed response clarifying what is consistent and novel about their inferred dynamics. I do feel more convinced about the revised results but I am still concerned about the accuracy of the inferred dynamical mechanism. I agree with the authors that whether the slow points were attractors or not was not really important to the main message of the paper. I am partly assuaged (but only partly) by the additional tests and corrections. I will cut to the chase.

Major concern 1: What, actually, is the dynamical mechanism found by FINDR?

To me it looks like non-normal nonlinear dynamics that lead to two slow points representing the different choices. If you look at what is happening in a neighborhood around the origin, it looks like the slow manifold is partially aligned with a fast mode that points in the direction of the two lobes of the slow S-shaped mode (the direction shown at time = 0.33s in 2h). The input directions are aligned with that fast mode which would give you textbook non-normal amplification. I want to stress here that the novelty is not as important, in my opinion, as the accuracy – if the answer is simply non normal amplification, that would be okay.

In brief: Yes, we agree that the dynamics are non-normal. The data, as we now show in Extended Data Fig. 16b, confirms this. However, we are less certain about how much explanatory power this observation provides, because we consistently find that the origin is unstable, meaning that amplification is expected whether or not the dynamics are normal or non-normal.

In more detail: As we now show in Extended Data Fig. 16b, for none of the 27 sessions was the angle between eigenvectors (in the 2-dimensional dynamics) equal to 90 degrees, implying non-normal dynamics. This confirms that the reviewer was correct in thinking that the dynamics are non-normal. However, we are less certain about how much explanatory power this observation provides, principally because in the dynamics that we found, and again consistent across all 27 sessions, the origin is unstable. (We will return to this important point in much more detail below.) Amplification therefore follows from the instability, regardless of whether the eigenvectors are normal or not.

Perhaps the reviewer is thinking of cases we are unfamiliar with, but in the papers we are familiar with, particularly in neuroscience (Goldman, *Neuron* 2009; Murphy and Miller, *Neuron* 2009; Daie et al., *bioRxiv* 2023), non-normal dynamics are used to explain transient amplification *when the origin is a stable point*. In that case, normal dynamics would not produce transient amplification, and the non-normality is the essence of the explanation. This is not the case when the origin is unstable.

Further reducing our confidence in the explanatory power of describing the dynamics as non-normal and nonlinear is the fact that non-normal, non-linear dynamics are, in a certain sense, generic. For example, if we take an artificial RNN with a $\tanh()$ nonlinearity and zero bias weights, the origin will be

a fixed point. The dynamics around this fixed point are normal only when there is some specific constraint (e.g., a symmetric weight matrix). For a random or noisy weight matrix, the dynamics will be non-normal with probability 1. For the sake of specificity, this example has been described for a particular type of RNN. But the point is more general: absent constraints that lead to normality, noise in weight matrices leads one to expect non-normal dynamics; and the brain is nonlinear. So in this sense we generically expect nonlinear, non-normal dynamics. (Again, if the origin were stable, and we needed to explain transient amplification, then the non-normality would indeed be a key part of the explanation, whether or not we expected non-normality.)

Perhaps the reviewer is thinking of a different case that we are not familiar with. If that is the case, we would welcome being guided by them. Currently, we are happy to point out in the manuscript that the dynamics are non-normal (Extended Data Fig. 16b). The reviewer is correct, that is indeed what the data say, and consistently so. But given the unstable origin, we are reluctant to emphasize the non-normality as the key factor for understanding the dynamics. Again, we would welcome guidance from the reviewer if they feel we are missing something important.

I appreciate the efforts taken by the authors to design refit a new model (cFINDR) to argue that the dynamics are not non-normal. It is an interesting approach. However, there are a lot of differences between cFINDR and FINDR. For example, the dynamics inferred by cFINDR appear very linear (ED Figure 7h). This is not so surprising since cFINDR constrains the latent dynamics to be almost linear (except the ϕ , but that is used only for the bistability). In contrast FINDR is parameterized by a neural network and can be very nonlinear. I don't see how cFINDR could find nonlinear non-normal dynamics and I wonder if FINDR fits the data better simply because it is nonlinear.

In brief: we regret the confusion: the purpose of cFINDR is not to argue that the dynamics are not normal. It is purely to make a head-to-head comparison to the three pre-existing hypotheses in Fig. 1. We believe it is well suited to that specific comparison. We have clarified this in the text.

In more detail: We must clarify that the purpose of cFINDR is *not* to “argue that the dynamics are not non-normal”. We hope the new Extended Data Fig. 16b, documenting the consistent non-normality in the dynamics that we inferred, and references to this Extended Data Fig. 16b in the text, will make it clear to all readers that we are in no sense claiming that the dynamics are normal. The purpose of cFINDR is much more particular: it is specifically focused on comparing to the three “classic” previous hypotheses in Fig. 1. A method to do that is needed because none of those three classic hypotheses have been fit directly to spiking data. We believe cFINDR serves this specific purpose adequately: both the classic DDM and the Mante, Sussillo et al. hypotheses are built around line attractors and focus on the linearized dynamics around it. These are captured well by the linear terms in cFINDR, which can describe both normal dynamics around a line attractor (**Fig. 1g**, “DDM line attractor”; Bogacz et al., 2006) and non-normal dynamics around a line attractor (**Fig. 1h**, “non-normal, *line attractor*”; Mante, Susillo et al., 2013). The Wong and Wang hypothesis focuses on bistability, and in cFINDR this is captured well by the ϕ term with a large radius ρ (**Fig. 1f**, “bistable attractors”). The comparisons between FINDR and cFINDR thus indicate that the dynamics from FINDR describe the data better than the three pre-existing hypotheses. But they do not imply that the dynamics are not non-normal.

The paragraph describing cFINDR begins with

To perform a head-to-head comparison with the three hypotheses of **Fig. 1d-h**

We hope this adequately emphasizes cFINDR's purpose.

Nevertheless, in the previous version of the main text, after the first half of the paragraph discussed the comparison between cFINDR and FINDR, we then wrote specifically about the Mante, Sussillo, et al. hypothesis, in a second half of the paragraph that began:

While the previous hypothesis suggesting non-normal dynamics with a line attractor can generate curved trial-averaged trajectories apparently similar to those we see in the data...

Perhaps our phrasing created confusion and led the reviewer to believe we were referring generically to non-normal dynamics. We very much apologize if that was the case. To make sure this does not happen and that readers are aware that we are referring narrowly and specifically to the Mante, Sussillo et al. hypothesis, we have amended this phrasing to:

While **one of** the previous hypotheses (**Fig. 1h**, suggesting a **line attractor with non-normal dynamics**) can generate curved trial-averaged trajectories apparently similar to those we see in the data (Extended Data Fig. 7g), there is a key difference, which is that in this **particular** previous hypothesis,

I am surprised that the origin is an unstable fixed point. Did you fit to data before the clicks or only after onset of the clicks? (I can't find the methods anymore to check this - did they disappear? Am I blind?) I wonder if this is why the origin is unstable.

We are ourselves puzzled by the disappearance of the Methods section and apologize for that. We have confirmed it is present in the current submission.

In brief: Yes, we fit only data after the onset of the clicks. We very robustly find the origin to be unstable.

In more detail: Thank you for raising this important point. We fit the data only after the onset of the clicks. There is no *a priori* reason to expect that different phases of the task, which are very saliently different to the animal, will be governed by the exact same dynamics. Indeed, classic hypotheses such as Wong and Wang 2006 specifically propose a bifurcation at the time of stimulus onset: at that point in time, the Wong and Wang 2006 system undergoes a bifurcation from tristable dynamics (with a stable point which is neutral in the sense of favoring neither of the possible decisions; see their Fig. 4A) to bistable dynamics, in which the two stable points represent the two possible decisions, and neutral points lie on the unstable manifold (their Fig. 4B and, related, 5B,C). Similarly, Machens, Romo, Brody 2005 specifically proposed bifurcations that would distinguish different phases within each individual trial of a parametric working memory task (their Fig. 3). In sum, the idea that there could be bifurcations separating different phases of a trial is well established.

For our data, we make no assumptions as to whether dynamics before stimulus onset are the same or different to that after stimulus onset. For this reason, we fit only data from stimulus onset forwards. We found that, in all 27 sessions, FINDR inferred the origin to be an unstable point.

Given the reviewer's questions about non-normality, we reasoned that whether or not the origin is unstable is a particularly important point, and we should make extra efforts to confirm our conclusions. We therefore double-checked the stability/instability of the origin using a model-free approach. To obtain these model-free estimates, we used the fact that our pulsatile stimulus contains silences in between the pulses. During those silences, the observed dynamics will be dominated by the autonomous dynamics of the system. Taking timepoints only from silences between pulses (more specifically, only timepoints in which there was no pulse within 60 ms), we computed the sample temporal derivative of the firing rates

of the recorded data, and projected those samples onto the plane used by the FINDR method. (This approach does share that choice of plane in neural space with the FINDR model; we used the same plane as FINDR to facilitate comparison to FINDR. But instability of the origin should not be affected by the choice of plane, so this does not affect the conclusions.) The single-trial, single-moment velocity vectors thus taken from the data are of course noisy; to reduce noise we computed their averages for each of five pie-slice-shaped regions immediately surrounding the origin. The average in each of those slices then represents the model-free estimate of the autonomous dynamics vector for that slice. Noise would result in a randomly-oriented vector for each slice¹. The slices were chosen to surround the origin: if all five vectors point away from the origin, this indicates an unstable point at the origin, and would occur by chance 3.1% of the time. The estimates we obtained are shown, for each of the 27 sessions, in Extended Data Fig. 16e. In 20 out of 27 individual sessions, the model-free estimates had all 5 arrows pointing away from the origin, thus statistically significantly ($p < 0.032$) indicating an unstable origin in the overwhelming majority of individual sessions. In not a single one out of the 27 sessions did we find all 5 arrows pointing inwards (which would be the requirement for significantly indicating a stable origin). Thus, the model-free approach very compellingly supports the origin being an unstable point.

Given the very consistent results across FINDR, the model-free approach, and across sessions, we believe that the evidence that the origin is an unstable point is extremely strong.

How can this [non-normality] be addressed? Well, one possibility is to constrain FINDR to linear normal dynamics, but I am concerned it wouldn't fit the data well either. A much simpler approach is to check if the Jacobian at the origin is normal. You can do this by taking the Schur decomposition of the Jacobian at the origin (when $u=0$), and calculating the Frobenius norm of the upper triangular part (not including the eigenvalues on the diagonal). Maybe take this as a fraction of the Frobenius norm of the diagonal, to get a ratio of the non-normal part to the normal part? But I am not sure how to make a strong statement from this. You could compare that value to the same value of the Jacobian calculated at other parts of the space to show that the origin has strongly non-normal dynamics for input amplification?

Since much of our analysis is in a 2-dimensional latent space, we can also evaluate non-normality simply by computing the angle between the two eigenvectors. We have done this, and also computed the Frobenius norm ratio, as suggested by the reviewer, and now show these results in Extended Data Fig. 16b. Across all sessions, the linearized dynamics around the origin are both consistently non-normal, and consistently unstable.

As described above, because the origin is unstable, we do not feel that we should be making a strong statement about the non-normality. As before, in the absence of some constraint not expected here (such as symmetric weight matrices) non-normal dynamics are to be expected. Nevertheless, we agree that it is worth documenting, and we thank the reviewer for pointing us in this direction. A future paper could perhaps explore in more detail to what extent non-normality vs non-linearity contribute to the shape of the autonomous dynamics trajectories. (Some sessions, for example session T176_208_05_31 in Extended Data Fig. 17 had an angle of 67 degrees between their eigenvectors, indicating only weak non-normality, yet still had strongly curved trajectories. This suggests that non-normality is unlikely to fully account for the curvature.) But we see that exploration, while interesting and valuable, as somewhat out of the scope of the current manuscript, which is more focused on the FINDR-inferred reduction in input dynamics magnitude as neural trajectories change direction, and on the biological implications of

¹ We note that when computing the sample derivative, we use the difference between points at timepoint t and at timepoint $t+1$. For each slice, we used only samples where timepoint t was located within the slice; but there was no restriction on the location of timepoint $t+1$. Consequently, if trajectories were random, the shape of the slice would have no effect on the results, and one would expect a randomly-oriented vector as the result for each slice.

this in the form of decision commitment, predictions made about behavior (tested and confirmed in Fig. 3n), and further implications regarding previous observations (**Fig. 4**) and comparisons across brain regions (**Fig. 5**).

We have added a description of the non-normal dynamics with an unstable origin to the Discussion, which now has a section that reads:

A recent study (Daie 2023) described neural trajectories that were well-described by non-normal dynamics (Goldman 2009, Murphy 2009). Consistent with this, the two-dimensional FINDR-inferred autonomous dynamics around the origin are also non-normal (Extended Data Fig. 16b,c), although with a key difference with respect to (Daie 2023, Goldman 2009, Murphy 2009), which is that here the origin is unstable (Extended Data Fig. 16a,e).

Major concern 2: Can we trust the flow field found by FINDR?

It is notoriously difficult to infer dynamics from neural data. Latent dynamics cannot be verified in ground truth in data (not including the toy models). Work from Valerio Mante, Lea Duncker, Guillaume Hennequin, and others have shown that the wrong dynamics can be inferred (which they verify with perturbation studies and trial-to-trial variability). Furthermore, FINDR has yet to be published meaning that it has not passed peer review.

How can this be addressed? One possibility is if you have perturbation data which would kick the trajectories into different regions of the flow field to make sure those results are consistent with what you have found with FINDR. A second possibility is to look at the trial-to-trial variability to analyze if those trajectories are consistent with the flow fields at a single trial resolution. A third possibility is to apply other methods to identify low dimensional dynamics (recurrent switching linear dynamical systems from Scott Linderman, low rank RNNs from Srđjan Ostojic) to check that the inferred mechanism is robust.

We appreciate and agree with the reviewer's point that inferring dynamics from neural data is difficult. Indeed, that is why FINDR was developed. We also agree that it is important to validate the results of FINDR.

In brief: The reviewer suggested three approaches towards validation: a) analyzing trial-to-trial variability; b) perturbations; c) applying other dynamics inference methods. We will add a fourth: testing predictions that follow from the inferred dynamics. Below we describe how our manuscript tested and confirmed a key prediction that followed from the main FINDR conclusions; and we describe a new analysis of trial-to-trial variability that supports the FINDR findings. (We also describe why we did not emphasize the other two approaches suggested by the reviewer.)

In more detail:

1) Our manuscript tests and confirms a core FINDR-based prediction. A major approach to validating a model is to make a prediction based on it, and test that prediction. We would like to point out that this is precisely what our manuscript does: our main conclusions from FINDR were that a change in the direction of the neural trajectory was paired with a transition from input-dominated to autonomous-dominated dynamics, suggesting that change in neural trajectory as the moment of

commitment to a decision. This is the moment that we refer to as the “neurally-inferred time of commitment” (nT_c). One of our key results, which we found remarkable in linking two types of completely different data, is the *behavioral* finding that sensory evidence presented after the *neurally*-defined nT_c does not affect the rat’s choice behavior (Fig. 3n). This finding thus confirms nT_c as a marker for internal (i.e., not overt) decision commitment. We consider this a major finding and it follows from, and is predicted by, the FINDR-inferred flow fields.

2) Analysis of trial-to-trial variability supports the FINDR-based conclusions. We followed the reviewer’s suggestion of analyzing trial-to-trial variability to probe for consistency with the FINDR results, and found that the results of this analysis supported the FINDR findings. We used an approach similar to the model-free approach used to confirm that the origin is an unstable point. That is, we compared the FINDR-inferred dynamics to dynamics estimated from sample derivatives of single-trial latent trajectories, separately from any dynamical model. There were three key differences in the model-free approach used here with respect to the model-free approach used for the origin: (1) here, we did not focus only on five tiles around the origin, but instead carried out estimates for each of an 8×8 array of regularly-spaced cells that together tiled the whole of the latent space spanned by the neural trajectories; (2) given that here we are estimating $8 \times 8 = 64$ vectors per session (not 5 as when focusing on the origin), we are more susceptible to noise. Therefore, to maximize the number of sample points used, we did not separate autonomous vs input vector fields, but instead estimated the net vector field; (3) we further reduced noise by smoothing spike trains with a $\sigma = 100\text{ms}$ Gaussian instead of the $\sigma = 20\text{ms}$ Gaussian we used when focusing on the origin. The neural population state trajectories from the smoothed spike trains were projected onto the subspace spanned by the FINDR latent axes, to allow direct comparisons between the vector fields estimated using this model-free approach and vector fields from FINDR. We then computed the average velocity vector for sample points in each of the 8×8 grid of cells, using the single-trial trajectories in that cell. Finally, we took the cosine similarity between the velocity vector for each of the 64 cells from FINDR and the velocity vector for each of the corresponding cells from the model-free approach, and took the mean of these cosine similarities, S_c , as a quantitative measure of the similarity between FINDR and the model-free approach. In computing S_c , only cells that had the number of states greater than 1% of the total number of states were included. When the number of states used to estimate the velocity vector was less than 1%, we considered that cell to be outside the “sample zone”, analogous to the sample zone in Figure 2.

Despite the noise inherent in the model-free approach, we found that the FINDR vector fields and those estimated using the model-free approach were substantially similar to each other, with a median cosine similarity S_c of 0.73 (Extended Data Fig. 16d). This confirms that a model-free analysis of trial-to-trial variability supports the FINDR-based results.

3) Neural perturbations are outside the scope of our manuscript. As the reviewer is probably aware, neural perturbation experiments to test flow field predictions are a whole further experiment in itself. They are beyond the scope of an already not-short manuscript.

4) Other existing flow field inference methods are outperformed by FINDR. We appreciate the suggestion of applying other latent dynamics inference methods to our data. We have in fact done so, and compared FINDR to them. Fig. 3 in Kim et al., *bioRxiv* 2023 shows that FINDR outperforms

rSLDS, SLDS, and auto-LFADS², in the sense that the model-predicted peri-stimulus time histograms (PSTHs) for these other approaches did not match the observed (and cross-validated) PSTHs as well as the FINDR-predicted PSTHs.

In further detail: Since the reviewer mentioned rSLDS in particular, below we include a figure (Reviewer Fig. 1) that has an updated version of the comparison made in Fig. 3 of Kim et al., 2023. For model evaluation, we followed the co-smoothing approach used in the Neural Latents Benchmark (Macke et al., 2011; Pei et al., 2021). We partitioned the trials in the session into 5 different folds. We used 3 of these folds for training, 1 fold for validation (hyperparameter optimization), and the remaining 1 fold for testing. With this testing set, we infer latent trajectories using 80% of the neurons, and then measure how well the PSTHs of the remaining held-out 20% of the neurons match the data. We show that, for this session, the 5-fold cross-validated coefficients of determination (R^2) of the evidence-sign conditioned PSTHs for the held-out neurons were significantly lower for rSLDS than those of FINDR. We used the ssm package to fit rSLDS, and used the variational Laplace EM method (Zoltowski et al., 2020) with structured mean-field variational posterior (a default setting in the package). For rSLDS, we did a hyperparameter search over the number of discrete latent states (1 through 4).

Taken together, we believe all of these validate the FINDR-based conclusions, and most importantly, validate and support the main conclusions of the manuscript, which is about the neural signature of the moment of decision commitment.

We'd like to thank the reviewer for their thought-provoking comments. We think that the model-free analyses that the reviewer prodded us to, both regarding the origin and the overall flow field, have significantly buttressed our conclusions, and we are grateful for that.

Minor concerns:

I am surprised that the origin is an unstable fixed point. Did you fit to data before the clicks or only after onset of the clicks? (I can't find the methods anymore to check this - did they disappear? Am I blind?) I wonder if this is why the origin is unstable.

Thank you for this question. Because it is so closely related to the issue of whether non-normal dynamics should be emphasized as the underlying explanation for our observations, we have answered this question above, immediately after the answer about cFINDR.

Did the authors remove the flow fields inferred from other animals? ED Fig 6 is replaced with validation across folds and seeds. That's fine, but I think it is very important for transparency and interpretation to show the inferred flow fields across ALL animals.

Thank you, we now include the FINDR-inferred vector fields from all 27 sessions in Extended Data Fig. 17.

² We are also interested in applying to our data low-rank RNNs, together with Ostojic, with whom we collaborate closely on a separate project. However, to date low-rank RNNs have only been applied to trial-averaged data. This means that their successful application to single-trial data will require some development.

Finally I have the same minor comment regarding mathematical precision. I just quickly flipped through the supplementary information and caught a few odd statements

- Eq 17, this is not a Fokker-Planck equation. This is just an OU process, which is a type of sde. The Fokker-Planck equation is a pde that describes the evolution of the distribution of an OU process.

- Section 3.1.4, $C(dt) = \phi$, where C was defined as an ode above. I am not sure what this means, it is not standard terminology.

This was from a quick skim but I urge the authors to go through a bit more carefully. I think it is important to get these details clear so that the paper is replicable.

Thank you for bringing this up. We revised the statements that the reviewer pointed out, and went through the Methods and Supplementary Information to correct mathematically imprecise statements we could find.

Referee #3 (Remarks to the Author):

The authors did a great job of addressing my concerns. I understood that the change in dynamics mode is not a transition from decision to action. The results including all sessions (Fig. 2k, l) are exactly what I wanted to see. I agree that the quantitative comparison with the authors' FINDR and Mante's attractor was not easy because there is no established way to model Mante attractor, so I think the authors did the best they could do. Overall, I have no further comments.

Thank you for the encouraging remark.

Minor concerns:

Just one point about Fig. 3n. The green "predicted" psychophysical kernel is added in the revised manuscript, but is it a quantitative prediction of MMDDM? I am asking because I am not sure if MMDDM really predicts a flat kernel before nT_c . A decision bound can have an additional effect (since choice is directly determined by the bound crossing, there would be stronger correlation between input pulse and choice at the moment of bound crossing), so it would not be flat. If it is not a quantitative prediction, the authors may decide to drop it or take care of it in some way. But I do not need to check the revision again.

Your suggestion regarding Fig. 3n is important. Indeed, the actual quantitative prediction of the psychophysical kernel is not flat before nT_c , but rather ramps up until nT_c and then abruptly drops after nT_c . We now show the quantitative prediction, generated using the MMDDM parameters fit to the data and also the same stimuli statistics from the data. Remarkably, the prediction well matches the observed psychophysical kernel.

Appendix

Figure 3n

n

weight of clicks on choice
inferred using logistic regression:

Extended Data Figure 16a-c

Extended Data Figure 16. Further analyses and validation of the dynamics discovered by FINDR. **a**, When we computed the eigenvalues of the numerical Jacobian J obtained from the detected slow point around the origin, the real components of both eigenvalues were greater than zero for all sessions ($n=27$), indicating that the origin is not a stable point. **b**, To quantify how non-normal the dynamics are around the origin, we computed the angle between the two eigenvectors of J . 90° indicates that the dynamics are normal, and angle less than 90° indicates that the dynamics are non-normal. **c**, We further evaluated the non-normality of the discovered dynamics around the origin by taking the Schur decomposition $J = QTQ^*$ and computing the ratio between the non-normal part and the normal part of the dynamics, $\rho = \|T_{1,2}\| / \|[T_{1,1}; T_{2,2}]\|$. $\rho > 0$ indicates that the dynamics are non-normal, with higher values of ρ indicating stronger non-normality.

Extended Data Figure 16d

Extended Data Figure 16... **d**, Here we estimated the low-dimensional vector field for each session using a method that does not specify a dynamical model (“model-free” approach). We compared the vector fields estimated using this approach to the FINDR-inferred vector fields. To obtain the model-free vector field, we first estimated single-trial firing rates of individual neurons by binning the spike trains in $\Delta t = 10\text{ms}$ bins and convolving the spike trains with a Gaussian of $\sigma=100\text{ms}$. Then, we projected the estimated single-trial population firing rate trajectories onto the subspace spanned by the FINDR latent axes. This allows direct comparisons between vector fields. For each evaluation point (i, j) on a 8-by-8 grid of the latent state space z , we estimated the velocity arrow by taking the average of $\dot{z} \approx (z_t - z_{t-1}) / \Delta t$ for all t across all trajectories that fall inside the cell corresponding to the point (i, j) . To compare vector fields, we measured S_c , the mean of the cosine similarities between the vector arrows of the model-free approach and the vector arrows from FINDR inside the sample zone. The median of the S_c 's across all sessions was 0.73. Three example sessions from across the distribution are shown, with session 2 around the median S_c of the histogram. For both FINDR and the model-free approach, the colored trajectories were obtained by trial-averaging based on the evidence strength. To compare between a random vector field and the model-free vector field, for each session, we generated 1,000 random vector fields by randomizing the direction of each arrow in the 8-by-8 grid.

Extended Data Figure 16e

Extended Data Figure 16... e, We assessed the dynamical stability around the origin using a model-free approach similar to d. We estimated the autonomous velocity around the initial starting point (indicated as the center of the circle) of the model-free latent trajectories by taking the average of $\dot{z} \approx (z_t - z_{t-1}) / \Delta t$ for all t across all trajectories that fall inside each of the 5 pie slices. Here we excluded time points where clicks affect the dynamics $(z_t - z_{t-1}) / \Delta t$, and only considered the trajectories with #L clicks = #R clicks during the epoch when they are in the pie slice, when computing the estimate of the autonomous dynamics arrow. When computing the average of $(z_t - z_{t-1}) / \Delta t$ for one of the pie slices, we required z_{t-1} to be inside the pie slice. The circles have a radius of 0.2 (in the units of z). We found that all five arrows were pointing outwards ($p < 0.5^5 = 0.03125$) for 20 out of 27 sessions, consistent overall with the stability analysis in a.

Extended Data Figure 17

Extended Data Figure 17. FINDR-inferred vector fields for all recording sessions ($n=27$) with more than 30 neurons and 400 trials, and sessions where the animal performed with greater than 80% accuracy. These fits were used for the summary plots in **Fig. 2k-l**.

Reviewer Figure 1

Reviewer Figure 1. FINDR outperforms SLDS (Fox et al., 2008; Linderman et al., 2017), rSLDS (Linderman et al., 2017), autoLFADS (Keshtkaran et al., 2022; Sedler & Pandarinath, 2023), and GPFA (Yu et al., 2008) in reconstructing the neural population activity of an example session. **a**, 5-fold cross-validated median evidence-sign conditioned PSTH R^2 on 13 held-out neurons (~20% of the neurons in this session) across different latent dimensions for FINDR, SLDS, rSLDS, autoLFADS, and

GPFA. For SLDS and rSLDS, we consider only the best-performing model among models assuming the number of discrete latent states ($=\{1, 2, 3, 4\}$). For autoLFADS, L corresponds to the factor dimension, not the size of the generator RNN. AutoLFADS with $L = 1$ fails to train. The baseline model here is defined as a constant firing rate model with the constant being the mean of the observed neural activity. **b**, Comparisons between FINDR and other models showing evidence-sign conditioned PSTH R^2 for all held-out neurons. **c**, FINDR-predicted evidence-sign conditioned PSTHs on example neurons ($L = 6$). **d**, SLDS-predicted evidence-sign conditioned PSTHs on example neurons ($L = 6$). **e**, rSLDS-predicted evidence-sign conditioned PSTHs on example neurons ($L = 6$). **f**, autoLFADS-predicted evidence-sign conditioned PSTHs on example neurons ($L = 6$). **g**, GPFA-predicted evidence-sign conditioned PSTHs on example neurons ($L = 6$).

We thank the reviewer for their comments. We will use blue for reviewer comments and black for our responses.

Referee #1 (Remarks to the Author):

The reviewers did an outstanding job at responding to my concerns. I am now really excited about the findings of this manuscript and am looking forward to seeing it in print. (I only have one small comment - when I suggested comparison with other methods, I meant to check that the inferred vector fields are similar. This isn't shown in the PSTH R squared analysis. I think it would be interesting and useful for readers who want to perform a similar analysis on their own data. However I leave this up to the authors. Their model free analysis was sufficient to respond to my concern about robustness of the findings.)

Among the other methods tested, only rSLDS allows visualization of vector fields in two dimensions. However, rSLDS failed to capture neural data when $L=2$ or $L=3$ (Reviewer Fig. 1a and e in our previous Response to Reviewers), so we did not plot vector fields estimated using rSLDS. If there is a method that performs competitively against FINDR in terms of held-out neural activity prediction with low-dimensional latents, and if we can visualize vector fields with the method, it would be interesting to see how the inferred vector field from this method differs from that of FINDR (e.g., Hu et al., *NeurIPS*, 2024 do this for rSLDS and gpSLDS for data from Nair et al., *Cell*, 2023 in their Figure 3).